# First-principles model of optimal translation factors stoichiometry

**Jean-Benoît Lalanne[1,2], Gene-Wei Li[1]\***

[1]Department of Biology, Massachusetts Institute of Technology, Cambridge, United States; [2]Department of Physics, Massachusetts Institute of Technology, Cambridge, United States

**Abstract** Enzymatic pathways have evolved uniquely preferred protein expression stoichiometry in living cells, but our ability to predict the optimal abundances from basic properties remains underdeveloped. Here, we report a biophysical, first-principles model of growth optimization for core mRNA translation, a multi-enzyme system that involves proteins with a broadly conserved stoichiometry spanning two orders of magnitude. We show that predictions from maximization of ribosome usage in a parsimonious flux model constrained by proteome allocation agree with the conserved ratios of translation factors. The analytical solutions, without free parameters, provide an interpretable framework for the observed hierarchy of expression levels based on simple biophysical properties, such as diffusion constants and protein sizes. Our results provide an intuitive and quantitative understanding for the construction of a central process of life, as well as a path toward rational design of pathway-specific enzyme expression stoichiometry.

**\*For correspondence:**
gwli@mit.edu

**Competing interests:** The authors declare that no competing interests exist.

## Introduction

A universal challenge faced by both evolution and synthetic pathway creation is to optimize the cellular abundance of proteins. This abundance optimization problem is not only multidimensional – often involving several proteins participating in the same pathway – but also under systems-wide constraints, such as limited physical space (*Klumpp et al., 2013*) and finite nutrient inputs (*You et al., 2013*). The complexity of this problem has prevented rational design of protein expression for pathway engineering (*Jeschek et al., 2017*). Fundamentally, being able to predict the optimal and observed cellular protein abundances from their individual properties would reflect an ultimate understanding of molecular and systems biology.

Evolutionary comparison of gene expression across microorganisms suggests that basic principles governing the optimization problem may exist. We recently reported broad conservation of relative protein synthesis rates within individual pathways, even under circumstances in which the relative transcription and translation rates for the homologous enzymes have dramatically diverged across species (*Lalanne et al., 2018*). Moreover, distinct proteins that evolved convergently toward the same biological function also displayed the same stoichiometry of protein synthesis in their respective species. These results suggest that the determinants of optimal in-pathway protein stoichiometry are likely modular and independent of detailed biochemical or physiological properties that differ across clades. However, the precise nature of such determinants remains unknown.

Translation of mRNA into proteins is a central pathway required for cell growth and therefore serves as an entry point for establishing a quantitative model of growth-optimized in-pathway stoichiometry. As a group, the total amount of translation-related proteins per cell mass linearly increases with growth rate in most conditions (*Scott et al., 2010*; *Dai et al., 2016*; *Schaechter et al., 1958*), a relationship considered a bacterial 'growth law'. In addition to ribosomes which have well-coordinated synthesis of subunits (*Nomura et al., 1984*), the translation pathway is comprised of nearly 100 protein factors involved in facilitating ribosome assembly, translation

initiation, elongation, and termination (*Marintchev and Wagner, 2004*; *Dever and Green, 2012*; *Rodnina, 2018*). The intracellular abundances of these factors vary over 100-fold (*Pedersen et al., 1978*; *Li et al., 2014*), and their ratios are often maintained in different growth conditions and across different species (*Lalanne et al., 2018*). What dictates the observed stoichiometry among translation factors is less understood. Early studies predicted expression of the highly expressed elongation factor Tu (EF-Tu) relative to the ribosome (*Klumpp et al., 2013*; *Ehrenberg and Kurland, 1984*) by maximizing translational flux per unit proteome. More recently, expression of several other components involved in the elongation step (ribosomes, tRNA, mRNA, EF-Tu, and EF-Ts) was predicted by minimizing the total mass of the components at a fixed translational flux (*Hu et al., 2020*). The selective pressure on expression levels remains to be determined for most members of the translation machinery, including initiation and termination factors that are much more lowly expressed and often assumed to be non-limiting.

Here, we sought to derive an intuitive model to understand the quantitative abundance hierarchy (*Figure 1B*) among the core translation factors (tlFs), which have well-characterized functions (*Table 1*, schematic in *Figure 1A*). Our goal is not to exhaustively model the heterogeneous movement of ribosomes on the transcriptome (*Shaw et al., 2003*; *Reuveni et al., 2011*;

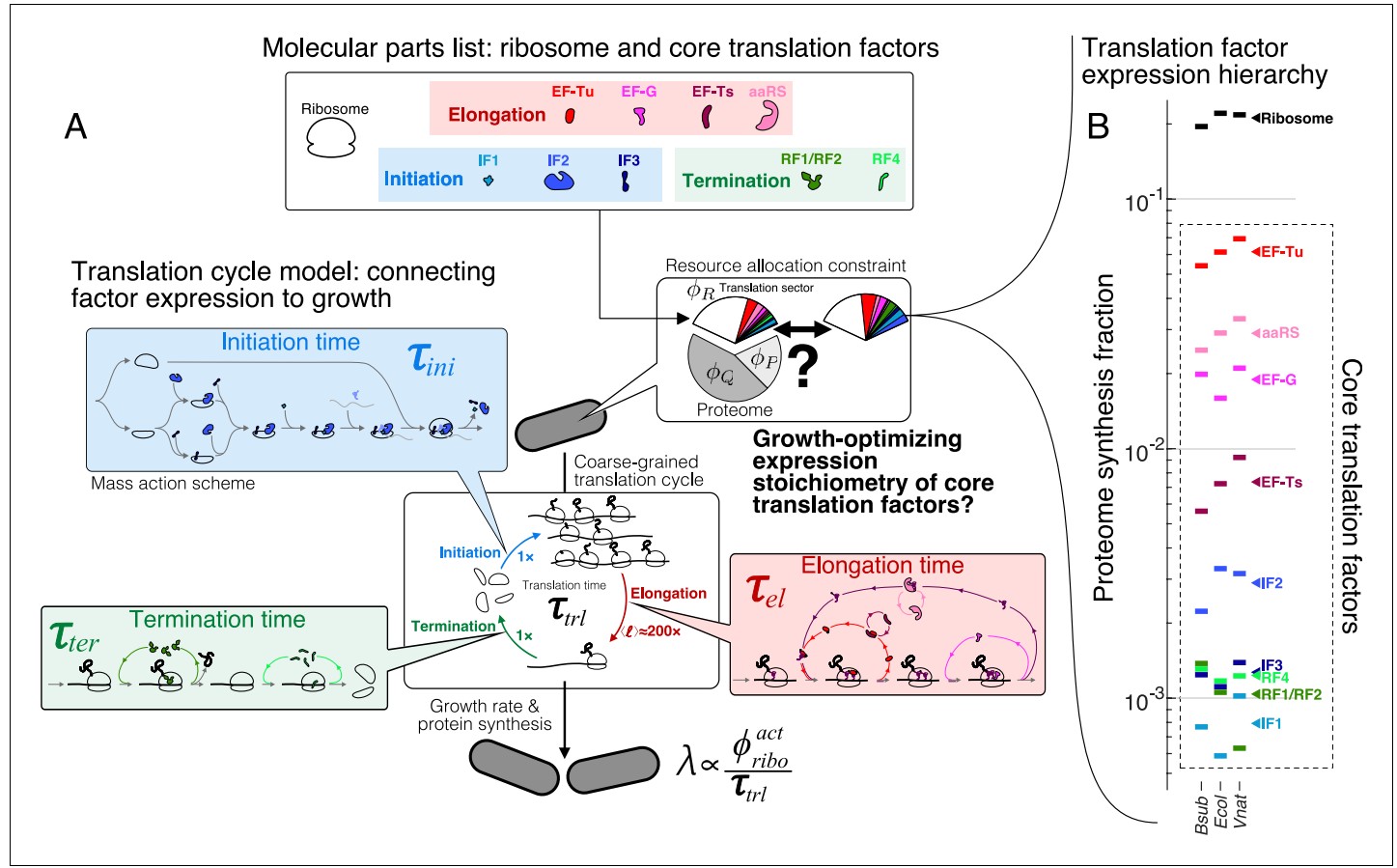

**Figure 1.** The hierarchy of mRNA translation factor expression stoichiometry. (A) Multiscale model relating translation factor expression to growth rate. The growth rate $\lambda$ is directly proportional to the active ribosome content ($\phi_{ribo}^{act}$) in the cell and inversely proportional to the average time to complete the translation cycle $\tau_{tl}$, consisting of the sum of the initiation ($\tau_{ini}$), elongation ($\tau_{el}$), and termination ($\tau_{ter}$) times. Each of these reaction times are determined by the translation factor abundances. On average, the elongation step is repeated around $\langle \ell \rangle \approx 200\times$ to complete a full protein, compared to $1\times$ for initiation and termination. Our framework of flux optimization under proteome allocation constraint addresses what ribosome and translation factor abundances maximize growth rate. (B) Measured expression hierarchy of bacterial mRNA translation factors, conserved across evolution. Horizontal bars mark the proteome synthesis fractions as measured by ribosome profiling (*Lalanne et al., 2018*) (equal to the proteome fraction by weight for a stable proteome) for key mRNA translation factors in *B. subtilis* (*Bsub*), *E. coli* (*Ecol*), and *V. natriegens* (*Vnat*) and are color-coded according to the protein (or group of proteins) specified. Triangles (◄) on the right indicate the mean synthesis fraction of the protein in the three species. See *Table 1* for a short description of the translation factors considered. Synthesis fractions in (B) can be found in *Supplementary file 1*.

**Table 1.** Brief description of the function of core translation factors considered.
For reviews of mRNA translation, see *Rodnina, 2018*; *Chen et al., 2016*.

| Step | Factor | Function |
|------|--------|----------|
| Initiation | IF1 | Initiation factor 1: binds to 30S ribosome subunits to facilitate initiator tRNA binding (*Laursen et al., 2005*; *Gualerzi and Pon, 2015*). |
| Initiation | IF2 | Initiation factor 2: ribosome-dependent GTPase interacting with 30 ribosome subunits, ensures correct binding of initiator tRNAs (*Laursen et al., 2005*; *Gualerzi and Pon, 2015*). |
| Initiation | IF3 | Initiation factor 3: prevents premature docking of 50S ribosomal subunits (*Laursen et al., 2005*; *Gualerzi and Pon, 2015*). |
| Elongation | EF-Tu | Elongation factor Tu: binds to charged tRNAs to form ternary complexes, brings charged tRNAs to empty ribosome A sites. (*Weijland et al., 1992*; *Agirrezabala and Frank, 2009*; *Andersen et al., 2003*) |
| Elongation | aaRS | tRNA synthetases: charge tRNAs with cognate amino acids (*Ibba and Soll, 2000*; *Pang et al., 2014*). |
| Elongation | EF-G | Elongation factor G: catalyzes translocation steps of the ribosome after peptide bond formation (*Andersen et al., 2003*; *Agirrezabala and Frank, 2009*). |
| Elongation | EF-Ts | Elongation factor Ts: nucleotide exchange factor for EF-Tu (*Agirrezabala and Frank, 2009*; *Andersen et al., 2003*). |
| Termination | RF1/RF2 | Peptide chain release factors 1 and 2: recognize stop codon and hydrolyze the completed protein. RF1 recognizes UAA, UAG, and RF2 UAA, UGA (*Bertram et al., 2001*). |
| Termination | RF4 | Ribosome recycling factor: catalyzes the dissociation of ribosome subunits following peptide chain release in translation termination (*Bertram et al., 2001*). |

*Subramaniam et al., 2014*; *Dykeman, 2020*) or to include as many details of the underlying molecular steps as possible (*Hu et al., 2020*; *Vieira et al., 2016*). Instead, we coarse-grained global translation into a cycle that consists of sequential steps with interconnected fluxes that depend on core tlFs concentrations. At steady-state cell growth, all individual fluxes are matched and the overall rate of ribosomes completing the full translation cycle is proportional to cell growth. By solving for the maximum flux under proteome allocation constraints, we obtained analytical solutions for the optimal factor concentrations, which agree well with the observed values. The ratios of optimal concentrations depend only on simple biophysical parameters that are broadly conserved across species. For instance, elongation factor EF-G is predicted to be more abundant than initiation and termination tlFs by a multiplicative factor of $\approx \sqrt{\text{average number of codons per protein}} \approx 14$, whereas EF-Tu is predicted to be more abundant than EF-G by a factor of $\approx \sqrt{\text{number of different amino acids}} \approx 4$. These results, arising from the optimization procedure and generic properties of the translation cycle, provide rationales for the order-of-magnitude expression of these important enzymes.

## Results

### Problem statement and model formulation

Our overall goal is to determine the growth-optimizing proteome allocation for the core translation factors. Conceptually, varying tlF concentrations has two opposing effects on cell proliferation. At the biochemical level, high tlF expression can facilitate growth by allowing more efficient usage of ribosomes. At the systems level, increased tlF expression can nonetheless limit growth by reducing the number of ribosomes and other proteins that can be produced. The tradeoffs between various tlFs and ribosomes create a multidimensional optimization problem.

We solve this multidimensional problem by treating translation as a dynamical system, in which ribosomes cycle through initiation, elongation, and termination. The resulting flux drives cell growth. During steady-state growth, every interlocked step of the translation cycle must have the same ribosome flux that is specified by the growth rate. We show that at the growth optimum, concentrations for distinct tlFs can be solved independently. The resulting analytical solutions can be expressed in terms of the growth rate and simple biophysical parameters.

### Cell growth driven by tlF-dependent ribosome flux

To describe the biochemical effects of tlF concentrations on cell growth, we first introduce a coarse-grained translation cycle time $\tau_{tl}$, or the time it takes for a ribosome to complete a typical cycle of protein synthesis (*Figure 1A*), which consists of three sequential steps: initiation ('*ini*'), elongation

('*el*'), and termination ('*ter*'). Each of these steps is catalyzed by multiple tlFs. The full translation cycle time is then sum of ribosome transit times at the three steps ($\tau_{tl} = \tau_{ini} + \tau_{el} + \tau_{ter}$), whose dependence on individual tlF concentrations can be quantitatively described through mass action kinetic schemes (schematically depicted in *Figure 1A*, see Appendices 2, 3, and 4 for details and examples below). We express tlF concentrations in units of proteome fractions (dry mass fraction of a specified protein to the full proteome), denoted by $\phi$ (*Scott et al., 2010*) (Materials and methods, section Conversion between concentration and proteome fraction). Using this notation, the translation cycle time $\tau_{tl}$ is a decreasing function of various tlFs concentrations ($\{\phi_{tlF,i}\}$).

In addition to its dependency on tlF concentrations, the translation cycle time provides a bridge between the cell growth rate and ribosome concentration. In steady-state growth (*Monod, 1949*; *Scott et al., 2010*; *Dai et al., 2016*), the growth rates of cells and of their protein content (total number of proteins) must be identical, denoted here as λ, as a result of the constant average cellular composition. The protein content grows at a rate determined by the flux of active ribosomes completing the translation cycle, that is $N_{ribo}^{act}/\tau_{tl}$, where $N_{ribo}^{act}$ is the number of active ribosomes per cell, divided by the total number of proteins $N_P$ per cell: $\lambda = N_{ribo}^{act}/\tau_{tl}N_P$. Active ribosomes are defined as those functionally engaged in, and cycling through, the initiation, elongation, and termination reactions of peptide synthesis. Rescaling to the total mass fraction (Materials and methods, section Conversion between concentration and proteome fraction) of proteome for active ribosomes ($\phi_{ribo}^{act}$) yields

$$\lambda = \frac{\phi_{ribo}^{act}}{\tau_{tl}} \frac{\langle\ell\rangle}{\ell_{ribo}}, \tag{1}$$

where $\ell_{ribo}$ is the number of amino acids in ribosomal proteins and $\langle\ell\rangle$ is the average number of codons per protein, weighted by expression levels (Materials and methods, section Average number of codons per protein: $\langle\ell\rangle$). The rescaling factor ($\ell_{ribo}/\langle\ell\rangle \approx 7300/200 = 36.5$) is approximately constant across growth conditions (Matrials and methods, section Average number of codons per protein: $\langle\ell\rangle$). This equation establishes how tlF concentrations affect the growth rate biochemically via $\tau_{tl}$.

We note that *Equation 1* is a generalized form of the bacterial growth law that relates the mass fraction of elongating ribosomes to growth rate ($\lambda = \frac{\phi_{ribo}^{el}}{\tau_{el}} \frac{\langle\ell\rangle}{\ell_{ribo}} = \gamma\phi_{ribo}^{el}$, where $\gamma$ is a rescaled translation elongation rate and $\phi_{ribo}^{el}$ is the proteome fraction of actively translating ribosomes [*Scott et al., 2010*; *Dai et al., 2016*; *Scott et al., 2014*]). This classic growth law was derived by considering the steady-state flux of peptide bond formation by elongating ribosomes, whereas our model focuses on the flux of ribosomes that traverse the entire translation cycle, thereby allowing us to consider the effects of translation factors and ribosomes engaged in additional steps (initiation, elongation, and termination). For each step, *Equation 1* can be extended to show that the growth rate is similarly proportional to the mass fraction of the corresponding ribosomes divided by the transit time at that step (Materials and methods, section Equality of ribosome flux in steady-state).

Steady-state growth thus imposes the requirement that the growth rate be inversely proportional to the translation cycle time and proportional to the number of active ribosomes engaged in the translation cycle (*Equation 1*). Inactive ribosomes, comprised of assembly intermediates, hibernating ribosomes, or otherwise non-functional ribosomes, have been found to constitute a small fraction ($\approx 5\%$) of the total ribosome pool for fast growth (*Lindahl, 1975*; *Dai et al., 2016*). Based on *Equation 1*, both increasing ribosome concentration and increasing tlF concentrations (which decreases $\tau_{tl}$) can accelerate growth. However, production of ribosomes and tlFs is subject to competition under a limited proteomic space, which we consider next.

## Optimization under proteome allocation constraint

To model the production cost tradeoff between tlFs and ribosomes, we integrate the flux-based formulation above with a proteomic constraint. Assuming that components of the translation machinery together accounts for a fixed fraction of proteome, that is, the 'translation sector' $\phi_{tl}$ (denoted $\phi_R$ in the context of growth laws [*Scott et al., 2010*]), the proteome fraction for active ribosomes is related to the proteome fraction for translation factors via

$$\phi_{ribo}^{act} = \phi_{tl} - \phi_{ribo}^{inact} - \sum_i \phi_{tlF,i}. \tag{2}$$

*Equations 1 and 2*, together with to the kinetic schemes for each step of the translation cycle, constitute the core of our model. Combining the biochemical effects (*Equation 1*) and the systems-level constraints (*Equation 2*) on tlFs, we arrive at a self-contained relationship between growth and tlF concentrations:

$$\lambda = \frac{\phi_{tl} - \phi_{ribo}^{inact} - \sum_i \phi_{tlF,i}}{\tau_{tl}(\{\phi_{tlF,i}\})} \frac{\langle \ell \rangle}{\ell_{ribo}}, \tag{3}$$

where we explicitly express $\tau_{tl}$ as a function of $\phi_{tlF,i}$ to reflect the dependence of ribosome transit times on translation factor abundances. The above relationship (*Equation 3*) allows us to ask: what is the stoichiometry of tlFs, or partitioning of the translation sector, that maximizes the growth rate (*Figure 1A*)?

The condition for the optimal TF abundances, that is, the set of $\phi_{tlF,i}$ that satisfies $(\partial \lambda / \partial \phi_{tlF,i})^* = 0$, can be obtained by considering the $\phi_{tlF,i}$ as independent variables and taking the derivative of *Equation 3* with respect to a specified tlF abundance. Under the assumptions that the translation sector ($\phi_{tl}$) and the proteome fraction for inactive ribosomes ($\phi_{ribo}^{inact}$) are both fixed in a given external nutrient condition, this yields

$$\left( \frac{\partial \tau_{tl}}{\partial \phi_{tlF,i}} \right)^* = -\frac{\langle \ell \rangle}{\ell_{ribo}} \frac{1}{\lambda^*}, \tag{4}$$

where the asterisk refers to the growth optimum within our model, that is, $(\partial \lambda / \partial \phi_{tlF,i})^* = 0$. Hence, under this framework, the tlF abundances are growth-optimized when the sensitivity of the translation cycle time to changing the considered tlF abundance ($\partial \tau_{tl} / \partial \phi_{tlF,i}$) reaches a value determined solely by the growth rate and protein size factors. We emphasize that the derivative above corresponds to a perturbation scenario in which the tlF abundance is changed while maintaining fixed the total proteomic resources to the translation sector, as prescribed by our optimization procedure. As such, it does not correspond an actual perturbation easily realizable experimentally.

Although *Equation 3* and the resulting optimization conditions (*Equation 4*, one for every tlF) corresponds to a coupled nonlinear system of multiple $\phi_{tlF,i}$, substantial decoupling occurs at the optimal growth rate. In this situation, most $\phi_{tlF,i}$ are only connected through the resulting growth rate. The optimization problem is then further simplified by the fact that the translation cycle consists of sequential and largely independent steps. The translation cycle time $\tau_{tl}$ corresponds to the sum of the coarse-grained initiation, elongation, and termination times, that is, $\tau_{tl} = \tau_{ini} + \tau_{el} + \tau_{ter}$. Given that each tlF is involved in a specific molecular step, the sensitivity matrix of these times to tlF concentration is sparse: $(\partial \tau_j / \partial \phi_{tlF,i})^* = 0$ for most combinations of $\tau_j$ and $\phi_{tlF,i}$. This lack of 'cross-reactivity' expresses that, for example, the initiation time $\tau_{ini}$ is unaffected by the tRNA synthetase concentration. This sparsity only occurs at the optimal expression levels, as the transit times typically depend on the growth rate (see an example in section Non binding-limited regime [one stop codon]) and $\partial \lambda / \partial \phi_{tlF,i} \neq 0$ away from the optimum. The optimum condition for factor $i$ then simplifies to:

$$\left( \frac{\partial \tau_j}{\partial \phi_{tlF,i}} \right)^* = -\frac{\langle \ell \rangle}{\ell_{ribo}} \frac{1}{\lambda^*}, \tag{5}$$

where $j$ above denotes the translation step(s) that tlF$_i$ participates in. This leads to simplifications that allow the system to be solved analytically in most cases: instead of solving the full system at once, individual reactions within the translation cycle can be considered in isolation. The resulting optimal concentrations are connected via the growth rate $\lambda^*$. Interestingly, the optimal stoichiometry among most tlFs is independent of $\lambda^*$ if the reactions are in the binding-limited regime, as we show below.

## Case study: Translation termination

We first illustrate the process of solving for the optimal tlF concentration for the relatively simple case of translation termination. The principles used here and the form of solutions provide conceptual guideposts for solving other steps of the translation cycle.

In bacteria, translation termination (*Bertram et al., 2001*) consists of two distinct, sequential steps: (1) stop codon recognition and peptidyl-tRNA hydrolysis catalyzed by class I peptide chain release factors RF1 and RF2, followed by (2) dissociation of ribosomal subunits from the mRNA, that is, ribosome recycling, catalyzed by RF4. We do not explicitly consider the additional factors (e.g. RF3 and EF-G) due to their lack of conservation or because they are non-limiting for this specific step (Appendix 2, section Omitted molecular details). RF1 and RF2 have the same molecular functions but recognize different stop codons (*Scolnick et al., 1968*): RF1 recognizes stops UAA and UAG, whereas RF2 recognizes UAA and UGA. For simplicity, we describe here a scenario where RF1 and RF2 have no specificity towards the three stop codons, which allows us to combine them in a single factor (denoted RFI). The model is readily generalized, with similar results, to the case of the two RFs with their specificity towards the three stop codons (Appendix 2, section Full three stop codons model).

Under a coarse-grained description, the total ribosome transit time at termination $\tau_{ter}$ can be decomposed into a sum of peptide release time and ribosome recycling time. In the treatment below, we consider a regime of binding-limited reactions for simplicity (rapid catalytic rate). A full model with catalytic components can also be solved analytically (Appendix 2, section Non binding-limited regime (one stop codon), *Figure 2A*). In the binding-limited regime ($k_{cat} \rightarrow \infty$), the peptide release time and ribosome recycling time are inversely proportional to the corresponding tlF concentrations:

$$\tau_{ter} = \frac{1}{k_{on}^{RFI}\phi_{RFI}} + \frac{1}{k_{on}^{RF4}\phi_{RF4}}, \tag{6}$$

where the association rate constants $k_{on}^{i}$ are rescaled by the factor's sizes in proteome fraction units (Materials and methods, section Conversion between concentration and proteome fraction). The above expression constitutes the solution of the mass action scheme for termination, connecting factor abundances to termination time.

The termination time (*Equation 6*) can then be directly substituted into the optimality condition (*Equation 5*) and solved in terms of $\lambda^*$:

$$\phi_{RFI}^* = \sqrt{\frac{\ell_{ribo}\lambda^*}{\langle\ell\rangle k_{on}^{RFI}}}, \qquad \phi_{RF4}^* = \sqrt{\frac{\ell_{ribo}\lambda^*}{\langle\ell\rangle k_{on}^{RF4}}}. \tag{7}$$

If the reactions are not binding-limited, an additional catalytic term $\propto \lambda^*/k_{cat}$ is added to the minimally required levels above (Appendix 2, section Non binding-limited regime [one stop codon]). The square-root dependence in the optimal RF concentrations emerges from the $\phi_i^{-1}$ dependence of $\tau_i$, for example, for ribosome recycling $\tau_{recyc} \propto \phi_{RF4}^{-1}$, which becomes $(\phi_i^*)^{-2}$ upon taking the derivative in the optimality condition (*Equation 5*). The square root is then obtained by solving for $\phi_i^*$. A similar square-root dependence has been noted in optimization of the ternary complex and tRNA abundances (*Ehrenberg and Kurland, 1984*; *Berg and Kurland, 1997*). Analysis of tlF expression across

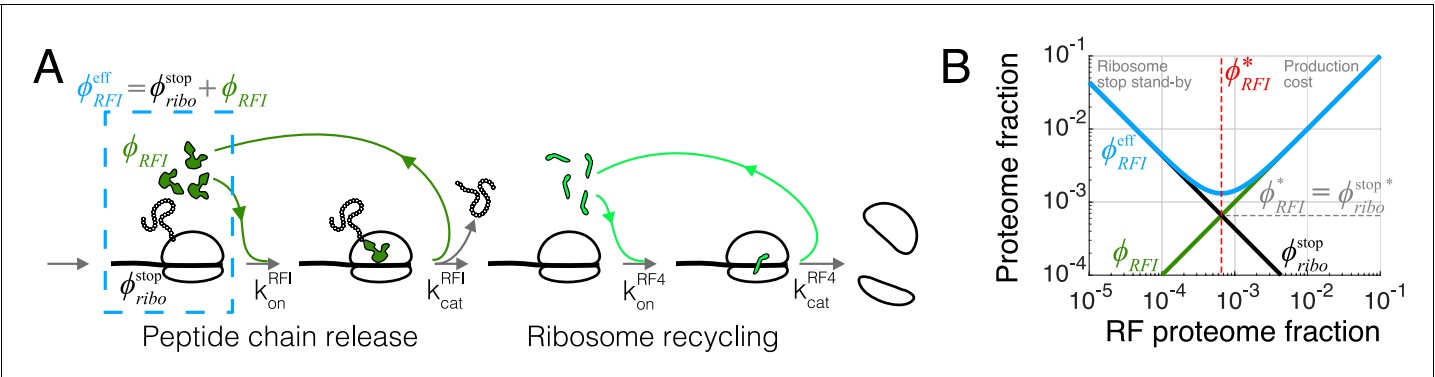

**Figure 2.** Case study with translation termination. (A) Coarse-grained translation termination scheme. (B) Illustration of the minimization of effective proteome fraction corresponding to peptide chain release factors, leading to the equipartition principle.

slower growth conditions supports the derived square root dependence (*Figure 4—figure supplement 2*). As a result of the square-root, the optimal RF concentrations are weakly affected by biophysical properties such as the association rate constants and protein sizes. In the binding-limited regime above, the ratio of the optimal concentrations between RFI and RF4 is independent of the growth rate and only depends on the kinetics of binding.

As a side note, the expression for termination time $\tau_{ter}$ in *Equation 6* must be modified in a regime where ribosomes are frequently queued upstream of stop codons. This would occur if the termination rate were slow and approached initiation rates on mRNAs (*Bergmann and Lodish, 1979*; *Lalanne et al., 2021*). In this regime, queues of ribosomes at stop codons would incur an additional time to terminate. In a general description, the resulting additional termination time can be absorbed in a queuing factor $\mathcal{Q} : \tau_{ter}^{full} := \tau_{ter} \ \mathcal{Q}(\tau_{ter})$ (Appendix 1 for derivation and discussion). The resulting nonlinearity would forbid the decoupling in the optimization procedure between RFI and RF4. Although absolute rates of termination are difficult to measure in vivo, translation on mRNAs is generally thought to be limited at the initiation step (*Laursen et al., 2005*), and consistently, ribosome queuing at stop codons in bacteria is not usually observed (except under severe perturbations, e.g. *Kavčič et al., 2020*; *Baggett et al., 2017*; *Mangano et al., 2020*; *Saito et al., 2020*; *Lalanne et al., 2021*). In the physiological regime of fast termination, the queuing factor converges to 1, yielding simple solutions that depend only on biophysical parameters (*Equations 7*).

## Equipartition between tlF and corresponding ribosomes

The optimal tlF concentrations (e.g. *Equation 7*) can also be intuitively derived from another viewpoint. For each reaction in the translation cycle, we can define an effective proteome fraction allocated to that process, combining the proteome fractions of the corresponding tlF and the ribosomes waiting at that specific step. As an example, for the case of peptide chain release factor (RFI) just treated, the effective proteome fraction includes the release factors and ribosomes with completed peptides waiting at stop codons (dashed box in *Figure 2A*), that is, $\phi_{RFI}^{eff} := \phi_{RFI} + \phi_{ribo}^{stop}$. This effective proteome fraction corresponds to the total proteomic space associated to a tlF in the context of the translation cycle.

During steady-state growth, the concentration of ribosomes waiting at any specific step of the translation cycle is equal to the total active ribosome concentration multiplied by the ratio of the transit time of that step to the full cycle: for example, here $\phi_{ribo}^{stop} = \frac{\tau_{stop}}{\tau_{tl}} \phi_{ribo}^{act}$, where $\tau_{stop} = 1/(k_{on}^{RFI} \phi_{RFI})$ is the time to arrival of RFI. Using *Equation 1* for $\phi_{ribo}^{act}$, the effective proteome fraction satisfies:

$$\phi_{RFI}^{eff} := \phi_{RFI} + \phi_{ribo}^{stop} \ = \phi_{RFI} + \frac{1}{\phi_{RFI}} \frac{\lambda}{k_{on}^{RFI}} \frac{\ell_{ribo}}{\langle \ell \rangle}$$
$$\geq 2\sqrt{\frac{\lambda}{k_{on}^{RFI}} \frac{\ell_{ribo}}{\langle \ell \rangle}}. \tag{8}$$

In the last line, we used the inequality of arithmetic and geometric means ($a + b \geq 2\sqrt{ab}$) to obtain the minimum of the effective proteome fraction. The equality holds when the two proteome fractions are equal ($\phi_{RFI} = \phi_{ribo}^{stop}$), which provides the solution for optimal $\phi_{RFI}$:

$$\phi_{RFI}^* = \sqrt{\frac{\ell_{ribo} \lambda^*}{\langle \ell \rangle k_{on}^{RFI}}}, \tag{9}$$

Hence, we recover *Equation 7* by minimizing the effective proteome fraction allocated to a given process in the translation cycle (the above argument applies to the optimal free concentration in the non-binding limited regime, see Appendix 2, section Non binding-limited regime (one stop codon) for an example). From this perspective, optimization of the translation apparatus balances the production cost of the enzyme of interest with the improved efficiency of a having less ribosomes idle at that step, *Figure 2B*. The optimal abundance in our model corresponds to a point of equipartition: the proteome fraction of free cognate factors equals the proteome fraction of ribosomes waiting at the corresponding step (*Figure 2B*).

### Case study: Ternary complex and tRNA cycle (EF-Tu and aaRS)

We next consider a more complex step of the translation cycle – elongation – and demonstrate that the optimality criterion (*Equation 5*) can similarly provide simple analytical solutions in the physiologically relevant regime. Translation elongation involves multiple interlocked cycles (one for each chemical species) and enzymes (EF-Tu, EF-G, EF-Ts, aminoacyl-tRNA synthetases (aaRS), and more). Our simplified kinetic scheme for translation elongation is shown in *Figure 3A*: charged tRNAs are brought to ribosomes through a ternary complex (TC), corresponding to a bound tRNA and EF-Tu. Following tRNA delivery and GTP hydrolysis, EF-Tu is released from the ribosome, and nucleotide exchange factor EF-Ts recycles EF-Tu back into the active pool, after which EF-Tu can bind a charged tRNA again and form another TC. At the ribosome, translocation to the next codon is

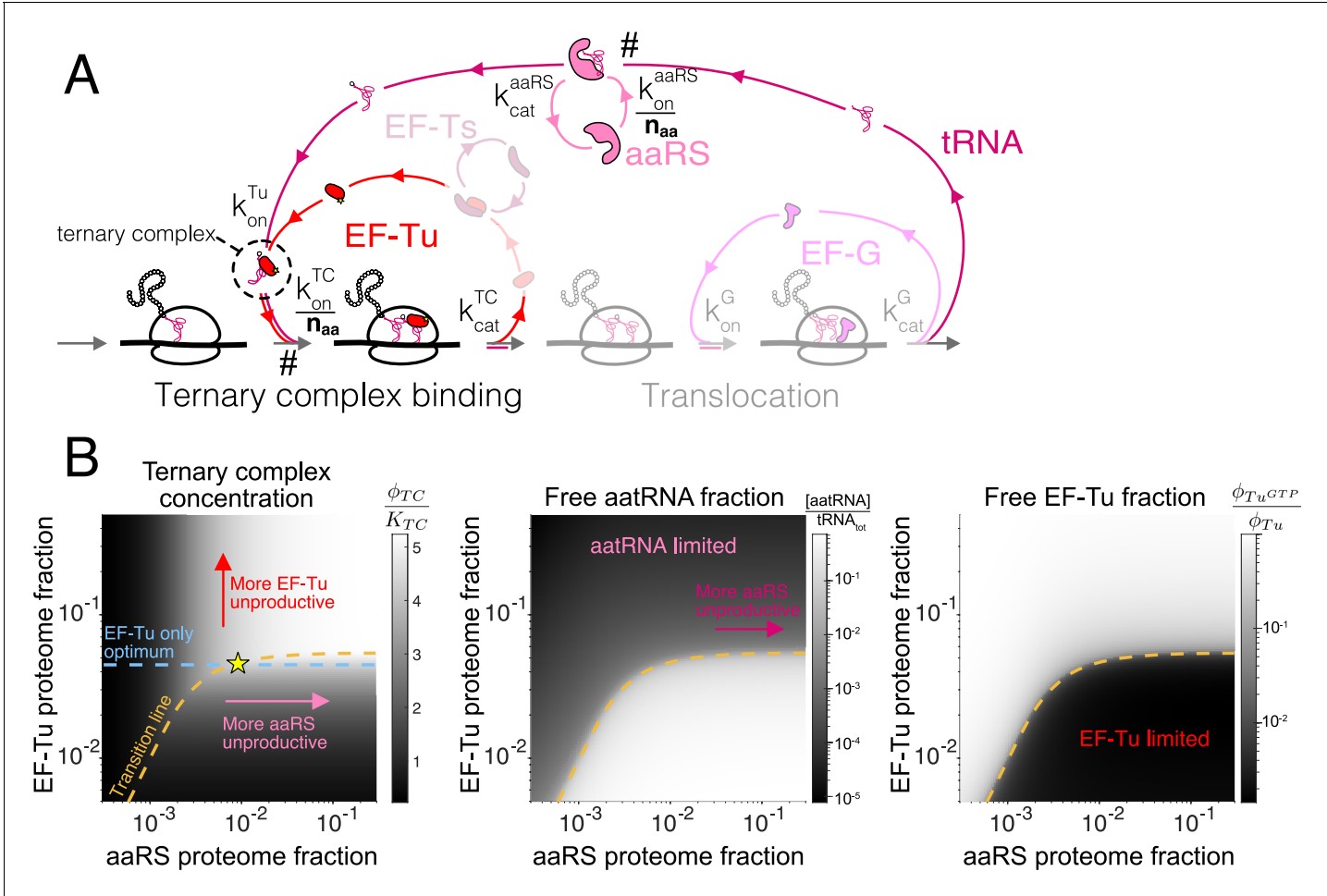

**Figure 3.** Case study with elongation factors (EF-Tu/aaRS). (**A**) Schematic of the translation elongation scheme, with the tRNA cycle, involving aminoacyl-tRNA synthetases (aaRS) and EF-Tu. Reactions with a # have their association rate constants rescaled by a factor of $n_{aa}^{-1} \approx 1/20$ through our coarse-graining to a single codon model. Greyed out cycles (EF-Ts and EF-G) can be solved in isolation (Appendix 3, sections Optimal EF-Ts abundance and Optimal EF-G abundance). (**B**) Exploration of the aaRS/EF-Tu expression space from numerical solution of the elongation model (Appendix 3, section Optimal EF-Tu and aaRS abundances). The transition line (orange) marks the boundary between the EF-Tu limited and aaRS limited regimes. Left panel shows the ternary complex concentration (which is closely related to the elongation rate, *Equation 10*). The ternary complex concentration is scaled by the dissociation constant $K_{TC}$ to the ribosome A site (see *Equation 39*). Middle panel shows the free charged tRNA fraction. Right panel shows the free EF-Tu fraction ($\phi_{Tu^{GTP}}$ denotes the proteome fraction of EF-Tu GTP that can bind to charged tRNAs to form the ternary complex). The star marks the optimal solution, as described in the text.

The online version of this article includes the following source data and figure supplement(s) for figure 3:

**Source code 1.** Source code to obtain panel (**B**) can be found in the associated scripts submitted with this work.

**Figure supplement 1.** Geometrical interpretation of the sharpness of the separation of the aaRS limited and EF-Tu limited regimes.

catalyzed by EF-G, followed by release of uncharged tRNAs. Aminoacyl-tRNA synthetases then charge tRNAs to complete the elongation cycle.

To reduce the complexity due to different tRNA isoacceptors and aaRSs, we self-consistently coarse-grained the translation elongation cycle to have a single codon (derived in Appendix 3, section Coarse-grained one-codon model). The resulting model harbors a single effective species for tRNA, aaRSs, and TCs, respectively. A rescaling factor ($1/n_{aa} \approx 1/20$, estimated in section Estimation of coarse-grained rates) arises in the procedure to decrease the rates of codon specific reactions and can be attached to either the respective rate constants or chemical species concentrations. In our formulation, we choose to rescale the association rate constants such that the coarse-grained abundance for each effective species corresponds to the sum over all individual codon-specific components. For example, $\phi_{aaRS}$ in our coarse-grained model corresponds to the summed proteome fraction of all aaRSs in the cell, and its association rate constant with the total tRNAs is rescaled by a factor of $1/n_{aa}$.

As a result of this choice of rescaling within our coarse-grained model, there are two classes of reactions in the elongation cycle that are distinguished by different kinetics: those that were codon specific (scaled by $1/n_{aa}$) and those that are not. Codon-specific reactions, for example, aaRS binding to cognate tRNAs and TC binding to cognate codons, are coarse-grained into one-codon reactions with reduced association rate constants (marked by # in *Figure 3A*). By contrast, codon-agnostic reactions do not incur such a rescaling and are thus much faster. We refer to this as a separation of timescale between the two classes of reactions (codon-specific vs. codon-agnostic), and note that this is not a reflection of slower underlying microscopic bimolecular reaction rates, but rather a result of our choice of variable in the coarse-graining.

Similar to translation termination, the factor-dependent ribosome transit time through a single codon ($\tau_{aa}$) is comprised of two steps, corresponding to binding of the TC and EF-G, respectively (formal derivation and non binding-limited regime in Appendix 3, section Coarse-grained translation elongation time):

$$\tau_{aa} = \frac{1}{\frac{k_{on}^{TC}}{n_{aa}}\phi_{TC}} + \frac{1}{k_{on}^{G}\phi_{G}}. \tag{10}$$

The coarse-grained factor-dependent portion of the total translation elongation time in our model is then given by the single codon time above multiplied by the average number of codons per protein, that is, $\langle \ell \rangle \tau_{aa}$. As discussed above, the rescaling of the TC association rate constant by $n_{aa}^{-1}$ arises as a result of our coarse-graining to a one-codon model (Appendix C, section C.1 Coarse-grained one-codon model). Note that the ternary complex concentration, $\phi_{TC}$, is a nonlinear function of the concentrations of all elongation factors (including $\phi_{G}$).

Despite the complexity of $\tau_{aa}$ as a function of the $\phi_{tlF,i}$, the fact that all fluxes are equal in steady-state allows several steps to be isolated and solved separately (EF-Ts and EF-G, greyed out in *Figure 3A*, respectively solved in Appendix C, sections C.3.3 Optimal EF-Ts abundance and C.3.4 Optimal EF-G abundance). For example, the approximate binding-limited solution for optimal EF-G concentration parallels that for termination factors:

$$\phi_{G}^{*} \approx \sqrt{\frac{\ell_{ribo}\lambda^{*}}{k_{on}^{G}}}. \tag{11}$$

Importantly, the optimum for EF-G is larger than the optimum for RFs by a factor $\sqrt{\langle \ell \rangle}$, reflecting that the typical translation cycle to produce a protein requires $\langle \ell \rangle$ steps catalyzed by EF-G and only one step for RFs (i.e. $\langle \ell \rangle \tau_{aa}$ enters the optimality condition, *Equation 5*, in contrast to $\tau_{ter}$ which is not multiplied by a scaling factor). The square root dependence arises here for the same reason as in the case of translation termination (derivative of $\phi^{-1}$).

In contrast to EF-G and EF-Ts, EF-Tu and aaRS cannot a priori be treated in isolation because the TC is composed of both EF-Tu and charged tRNAs. Still, the separation of timescales within our coarse-grained model (see Appendix C, section Interpretation of the sharp separation between aaRS and EF-Tu limited regimes) simplifies the solution considerably. Indeed, rapid binding of charged tRNAs to EF-Tu leads to either component being limiting for ternary complex concentration in most of the aaRS/EF-Tu expression space, leading to two clearly delineated regimes (*Figure 3B*). In one

regime, charged tRNAs are limiting (low aaRS), whereas EF-Tu is limiting in the other (low EF-Tu). These regimes are separated by a narrow transition region, whose sharpness is a reflection of the smallness of the rate rescaling parameter $n_{aa}^{-1}$ (see Appendix 3, section Interpretation of the sharp separation between aaRS and EF-Tu limited regimes). We term the focal region separating the two regimes in the aaRS/EF-Tu expression space the 'transition line' (see *1* for derivation and additional details).

The transition line corresponds to conditions in which EF-Tu and aaRS are co-limiting for TC concentration. In the EF-Tu limited region, increasing aaRS abundance does not increase ternary complex concentration: since all EF-Tu proteins are already bound to charged tRNAs, increasing tRNA charging cannot further increase TC concentration. Conversely, in the aatRNA limited region, increasing EF-Tu abundance does not increase TC concentration: since all charged tRNAs are already bound by EF-Tu, increasing EF-Tu concentration does not alleviate the requirement for more charged tRNAs. Given that the optimality condition requires non-zero increase in ternary complex concentration with increasing factor abundance (*Equation 5* using $\tau_{aa}$ from *Equation 10*), the optimal EF-Tu and aaRS abundances must be on the transition line.

Which point on the transition line corresponds to the optimum? Note that inside the EF-Tu limited region, the ternary complex concentration is entirely set by the total EF-Tu concentration: $\phi_{TC} \approx \phi_{Tu}$ (since most EF-Tu proteins are bound by charged tRNAs, *Figure 3—figure supplement 1*). As an approximation resulting from the narrow range of transition region (*Figure 3* and *Figure 3—figure supplement 1*), we assume that the EF-Tu limited regime solution $\phi_{TC} \approx \phi_{Tu}$ holds up to very close to the transition line. Replacing $\phi_{TC}$ by $\phi_{Tu}$ in the elongation time *Equation 10* and substituting in the optimality condition (*Equation 5*), the approximate optimal abundance for EF-Tu (the full solution includes additional terms from the EF-Ts cycle, section Optimal EF-Tu and aaRS abundances) can then be obtained in the same way as for translation termination factors:

$$\phi_{Tu}^* \approx \sqrt{\frac{\ell_{ribo} n_{aa} \lambda^*}{k_{on}^{TC}}}. \tag{12}$$

Importantly, compared to the solution for EF-G, the above is multiplied by an additional factor of $\sqrt{n_{aa}}$. This contribution arises from the rescaling of the association rate for the ternary complex to the ribosome in our coarse-grained one-codon model, increasing the requirement on EF-Tu abundance.

From the necessity for the combined EF-Tu and aaRS solution to fall on the transition line, the approximate solution for the optimal aminoacyl-tRNA synthetase abundance is then the intersection (yellow star in *Figure 3B*) of the transition line with the EF-Tu-only solution described above (dashed blue line in *Figure 3B*, derivation of solution in *Box 1*).

For the above derivation to be valid, the total number of tRNAs in the cell must be sufficient to accommodate all ribosomes (about two per ribosome, A- and P-sites) and binding to all EF-Tu (about $gt_4$ per ribosome based on endogenous expression stoichiometry [*Li et al., 2014*; *Lalanne et al., 2018*]). The number of tRNAs per ribosomes in the cell should thus be at least 6×. Remarkably, estimates of this ratio in the cell suggest that this is barely the case (between 6 and 7 tRNAs/ribosome at fast growth [*Dong et al., 1996*]). Although our model treats the total tRNA abundance as a measured parameter and omits its selective pressure (see *Hu et al., 2020* which includes RNA mass in their optimization procedure), the abundance of three core components of the tRNA cycle appear to be at the special point where the transition line plateau, that is set by total tRNA abundance, just crosses the EF-Tu-only optimum (blue line in *Figure 3B*). At this point, all three components are co-limiting.

## Optimal stoichiometry of mRNA translation factors

Analogous to the case studies above, optimal concentrations for all core translation factors can be solved using the optimality condition (*Equation 5*) and their respective kinetics schemes (the case of translation initiation is solved in Appendix 4). The analytical forms of the optimal solutions are shown in *Table 1*. In the binding-limited regime, the ratios of growth-optimized tlF concentrations are independent of the growth rate (except for aaRS), and are dependent only on basic biophysical parameters, such as protein sizes and diffusion constants.

## Box 1. The EF-Tu and aaRS transition line.

Within our framework, optimality of translation factors is dictated by how coarse-grained ribosome transit times depend on factors' abundances (*Equation 4*). For elongation factors aaRS and EF-Tu, contribution to the ribosome elongation time ($\tau_{el} = \langle \ell \rangle \tau_{aa}$) is through the concentration of the ternary complex (*Equation 10*). Obtaining the optimal EF-Tu and aaRS abundance therefore requires solving for the ternary complex concentration as a function of these two variables.

The steady-state solution for the ternary complex concentration in the aaRS/EF-Tu expression displays two sharply separated regime (*Figure 3B*), separated by a narrow transition region (the 'transition line'). As described in the main text, the transition line plays a critical role for identifying the optimal EF-Tu and aaRS abundances within our model. Away from the line, there is an unproductive excess of either factors, viz. either $\partial \phi_{TC} / \partial \phi_{Tu} \approx 0$ or $\partial \phi_{TC} / \partial \phi_{aaRS} \approx 0$. Here, we derive the equation for the transition line. First, we leverage the constraint imposed by the conservation of tRNAs, which in our model is:

$$\mathrm{tRNA}_{tot} = [\mathrm{R}_\emptyset] + \underbrace{2[\mathrm{R}_{TC}] + 2[\mathrm{R}_{tRNA}] + 2[\mathrm{R}_G]}_{\propto \lambda / k_{el}^{max}} + [\mathrm{tRNA}] + [\mathrm{tRNA:aaRS}] + [\mathrm{aatRNA}] + [\mathrm{TC}].$$

Above, $\mathrm{tRNA}_{tot}$ corresponds to the total tRNA concentration in the cell. In addition: $\mathrm{R}_\emptyset$: elongating ribosomes with empty A-site, $\mathrm{R}_{TC}$: ribosomes with bound TC, $\mathrm{R}_{tRNA}$: ribosomes with filled A-site and no bound factor, $\mathrm{R}_G$: ribosomes with bound EF-G, tRNA: free uncharged tRNAs, tRNA:aaRS: tRNA and aaRS complex, aatRNA: free charged tRNAs, and TC: ternary complex. Here, we assume that the elongating ribosomes always have a tRNA in the P-site, and a negligible occupancy in the E-site.

Using the system of equations from the mass action scheme at steady-state (section Translation elongation: optimal solutions), variables in the tRNA conservation equation above can be solved for in terms of the total abundance of EF-Tu and aaRS, the growth rate, and the steady-state ternary complex concentration. We note that the three ribosome species with a filled A site ($\mathrm{R}_{TC}$, $\mathrm{R}_{tRNA}$, and $\mathrm{R}_G$) do not depend on EF-Tu concentration, and can be coarse-grained to a term proportional to $\lambda / k_{el}^{max}$, where $k_{el}^{max}$ is the maximal translation elongation rate (not including the TC diffusion contribution) (*Dai et al., 2016*). In the binding-limited regime, converting to proteome fraction units, and leaving out the EF-Ts contribution without loss of generality (see section Optimal EF-Tu and aaRS abundances for a full treatment), we have:

$$\psi_{tRNA} = \underbrace{\frac{\lambda(\phi_{TC})}{\frac{k_{on}^{TC}}{n_{aa}}\phi_{TC}} \phi_{TC}}_{\mathrm{R}_\emptyset} + \frac{2\lambda(\phi_{TC})}{k_{el}^{max}} + \underbrace{\frac{\lambda(\phi_{TC})}{\frac{k_{on}^{aaRS}}{n_{aa}}\phi_{aaRS}}}_{\text{free uncharged tRNA}} + \underbrace{\frac{\lambda(\phi_{TC})}{k_{on}^{Tu}\phi_{Tu^{\mathrm{GTP}}}}}_{\text{free aatRNA}} + \frac{\phi_{TC}}{\ell_{Tu}},$$

$$\text{where } \phi_{Tu^{\mathrm{GTP}}} := \phi_{Tu} - \phi_{TC}. \tag{13}$$

Above, $\psi_{tRNA}$ is a normalized tRNA concentration (see *Equation 28*). We have explicitly highlighted that the growth rate is dependent on EF-Tu and aaRS only through the ternary complex concentration $\phi_{TC}$. From the definition of of the elongation time (*Equation 10*), we have $\lambda(\phi_{TC}) \propto \phi_{TC}/(K_{TC} + \phi_{TC})$(*Klumpp et al., 2013*; *Dai et al., 2016*) (definition of $K_{TC}$ in terms of model parameters: supplement, *Equation 39*). *Equation 13* is closed and can be solved for $\phi_{TC}$ at given abundances of EF-Tu ($\phi_{Tu}$) and aaRS ($\phi_{aaRS}$).

Although *Equation 13* is non-linear and cannot be solved exactly in general, the separation of timescales in our coarse-grained description simplifies the problem considerably. Indeed, numerical solutions of *Equation 13* (*Figure 3B*, section Optimal EF-Tu and aaRS abundances) show that the behavior of TC concentration in the two-dimensional EF-Tu/aaRS expression space is split into two distinct regimes, sharply delineated by a transition line (orange line in *Figure 3B*, a geometric heuristic explaining the sharp separation between the regimes is presented in Appendix 3, section Interpretation of the sharp separation between aaRS and EF-Tu limited regimes, *Figure 3—figure supplement 1*). Since TC concentration only increases as a function of both aaRS and EF-Tu on the transition line, the optimal solutions for the two factors must fall on it.

An expression for the transition line can be derived. Conceptually, the region of transition between the two regimes has both a low concentration of free EF-Tu molecules ($\phi_{Tu^{GTP}}/\phi_{Tu} \approx 0$) and a low concentration of free charged tRNAs ([aatRNAs]/tRNA$_{tot} \approx 0$). Although no values in the aaRS/EF-Tu expression plane can formally satisfy these two conditions simultaneously, the transition line is specified by setting the free charged tRNA term to 0 and replacing $\phi_{TC}$ by $\phi_{Tu}$ (no free EF-Tu) in **Equation 13**. We denote by $(\bar{\phi}_{Tu}, \bar{\phi}_{aaRS})$ points satisfying the resulting requirement, namely (see **Equation 40** for non binding-limited case):

$$\text{Transition line :} \psi_{\text{tRNA}} - \frac{\lambda(\bar{\phi}_{\text{Tu}})\text{n}_{\text{aa}}}{\text{k}_{\text{on}}^{\text{TC}}\bar{\phi}_{\text{Tu}}} - \frac{2\lambda(\bar{\phi}_{\text{Tu}})}{\text{k}_{\text{el}}^{\text{max}}} - \frac{\bar{\phi}_{\text{Tu}}}{\ell_{\text{Tu}}} := \Delta_{\text{tRNA}}(\bar{\phi}_{\text{Tu}}) = \frac{\text{n}_{\text{aa}}\lambda(\bar{\phi}_{\text{Tu}})}{\text{k}_{\text{on}}^{\text{aaRS}}\bar{\phi}_{\text{aaRS}}}, \quad (14)$$

where we have defined the excess tRNA ($\Delta_{tRNA}$) above. In words, $\Delta_{tRNA}$ corresponds to the available tRNAs after the tRNAs sequestered on ribosomes and EF-Tu in the TC are subtracted from the total tRNA budget. At large aaRS concentrations, the transition line plateaus as a result of the finite total tRNA budget within the cell (**Figure 3B**, middle panel). The plateau is reached once all tRNAs aaRS are charged: the system is then no longer limited by aaRSs, but by the amount of tRNAs.

Using the requirement that the optimum must fall on the transition line and the approximate solution for the EF-Tu optimum, the approximate optimal solution for aaRS is, from **Equation 14** (section Optimal EF-Tu and aaRS abundances for non binding-limited solution):

$$\phi_{aaRS}^* \approx \frac{n_{aa}\lambda^*}{k_{on}^{aaRS}\Delta_{tRNA}^*}, \text{where :} \Delta_{tRNA}^* = \psi_{tRNA} - \frac{n_{aa}\lambda^*}{k_{on}^{TC}\phi_{Tu}^*} - \frac{2\lambda^*}{k_{el}^{max}} - \frac{\phi_{Tu}^*}{\ell_{Tu}} \quad (15)$$

Within our model, the optimal aaRS concentration is thus set by the excess tRNAs at the EF-Tu optimum ($\Delta_{tRNA}^*$).

To obtain the numerical values of association rate constants needed for calculating the optimal tIF stoichiometry (**Table 1**), we used the measured $\hat{k}_{on}^{TC}$ in vivo and estimated all other association rate constants using a biophysically motivated scaling ($\hat{k}$ denotes the raw association rate constant in units µM$^{-1}$s$^{-1}$, which is different from the rescaled $k$, see section Conversion between concentration and proteome fraction). To our knowledge, the binding between TC and ribosomes, $\hat{k}_{on}^{TC} = 6.4$ µM$^{-1}$s$^{-1}$ (**Dai et al., 2016**), is the only measured association rate constant for any tIFs in a physiological context. We estimate the association rate constants for other reactions by scaling $\hat{k}_{on}^{TC}$ by the respective diffusion coefficients of the chemical species, that is for reaction involving species $A$ and $B$: $\hat{k}_{on}^{AB}/\hat{k}_{on}^{TC} = (D_A + D_B)/(D_{TC} + D_{ribo})$, where $D_i$ is the diffusion constant for the molecular species $i$ (see **Appendix 5—table 2**). Diffusion constants for several tIFs have been measured experimentally (**Bakshi et al., 2012**; **Sanamrad et al., 2014**; **Plochowietz et al., 2017**; **Volkov et al., 2018**), and uncharacterized ones can be estimated using the cubic-root scaling with number of codons per protein from the Stokes-Einstein relation (**Nenninger et al., 2010**) (see **Appendix 5—table 1**). For simplicity, this approach assumes that reactive radii and orientational constraints are similar for the different reactions (see 3 Discussion for additional assumptions). These strong assumptions are necessary given the lack of in vivo biochemical parameter measurements, and can be relaxed as refined empirical determination for more physiological association rates become available in the future. Nonetheless, we note that the square-root dependence on these parameters (**Table 1**) for our predictions makes the numerical values less sensitive to possible tIF-specific effects.

The estimated optimal tIF concentrations show concordance with the observed ones, both in terms of the absolute levels and the stoichiometry among tIFs (**Figure 4** for fast growth, see **Supplementary file 1** for data and **Figure 4—figure supplement 1** for additional growth conditions). A hierarchy of expression levels emerges such that the factors involved in elongation are more abundant compared to initiation and termination factors. The separation of these two classes is driven by the scaling factor $\sqrt{\langle \ell \rangle} \approx 14$ in our analytical solutions, which reflects the fact that the flux for elongation factors is $\langle \ell \rangle \approx 200$ times higher than that for initiation and termination factors. Within

**Table 2.** Compilation of predicted optimal abundances for translation factors.

The optimal abundance is the sum of the terms in each row. Columns correspond to contributions of different nature (diffusion of factor itself, diffusion of other factors involved in the factor's cycle, catalytic term). Terms must be multiplied by the common factors indicated in each column's header ($\propto$). For RF1+RF2, $\delta := 2\sqrt{f_{UAG}f_{UGA}}$ (see section Optimal abundances for RF1/RF2).

| Factor | Diffusion (direct) $\propto \sqrt{\frac{\lambda^*}{P}}$ | Diffusion (other) $\propto \sqrt{\frac{\lambda^*}{P}}$ | Catalytic sequestration $\propto \lambda^*$ |
|---|---|---|---|
| IF1 | $\sqrt{\frac{\ell_{ribo}\ell_{IF1}}{\langle\ell\rangle\hat{k}_{on}^{IF1}}\left[1+\frac{\ell_{IF2}+\ell_{IF3}}{\ell_{ribo}}\right]}$ | $\frac{\ell_{IF1}}{\langle\ell\rangle}\sqrt{\frac{\langle\ell\rangle}{\hat{k}_{on}^{50S}}}$ | $\frac{\ell_{IF1}}{\langle\ell\rangle}\left(\frac{1}{k^{RNA}}+\frac{1}{k_{cat}^{ini}}\right)$ |
| IF2 | $\sqrt{\frac{3}{4}}\sqrt{\frac{\ell_{ribo}\ell_{IF2}}{\langle\ell\rangle\hat{k}_{on}^{IF2}}}$ | $\frac{\ell_{IF2}}{\langle\ell\rangle}\left(\sqrt{\frac{\ell_{ribo}\ell_{IF1}}{\langle\ell\rangle\hat{k}_{on}^{IF1}}}+\sqrt{\frac{\langle\ell\rangle}{\hat{k}_{on}^{50S}}}\right)$ | $\frac{\ell_{IF2}}{\langle\ell\rangle}\left(\frac{1}{k^{RNA}}+\frac{1}{k_{cat}^{ini}}\right)$ |
| IF3 | $\sqrt{\frac{3}{4}}\sqrt{\frac{\ell_{ribo}\ell_{IF3}}{\langle\ell\rangle\hat{k}_{on}^{IF3}}}$ | $\frac{\ell_{IF3}}{\langle\ell\rangle}\left(\sqrt{\frac{\ell_{ribo}\ell_{IF1}}{\langle\ell\rangle\hat{k}_{on}^{IF1}}}+\sqrt{\frac{\langle\ell\rangle}{\hat{k}_{on}^{50S}}}\right)$ | $\frac{\ell_{IF3}}{\langle\ell\rangle}\left(\frac{1}{k^{RNA}}+\frac{1}{k_{cat}^{ini}}\right)$ |
| EF-G | $\sqrt{\frac{\ell_{ribo}\ell_{G}}{\hat{k}_{on}^{G}}}$ | | $\frac{\ell_{G}}{k_{cat}^{G}}$ |
| EF-Ts | $\sqrt{\frac{\ell_{Tu}\ell_{Ts}}{\hat{k}_{on}^{Ts}}}$ | | $\frac{\ell_{Ts}}{k_{cat}^{Ts}}$ |
| EF-Tu | $\sqrt{\frac{\ell_{ribo}\ell_{Tu}n_{aa}}{\hat{k}_{on}^{TC}}}$ | $\sqrt{\frac{\ell_{Tu}\ell_{Ts}}{\hat{k}_{on}^{Ts}}}$ | $\ell_{Tu}\left(\frac{1}{k_{cat}^{TC}}+\frac{1}{k_{cat}^{Ts}}\right)$ |
| RF1+RF2 | $\sqrt{\frac{\ell_{ribo}\ell_{RFI}(1+\delta)}{\langle\ell\rangle\hat{k}_{on}^{RFI}}}$ | | $\frac{\ell_{RFI}}{\langle\ell\rangle k_{cat}^{RFI}}$ |
| RF4 | $\sqrt{\frac{\ell_{ribo}\ell_{RF4}}{\langle\ell\rangle\hat{k}_{on}^{RF4}}}$ | | $\frac{\ell_{RF4}}{\langle\ell\rangle k_{cat}^{RF4}}$ |

each class, the finer hierarchy of expression levels can also be further explained by simple parameters. For example, EF-Tu is predicted to be more abundant than EF-G by a factor of $\sqrt{n_{aa}\ell_{Tu}/\ell_{G}}\approx 3.3$ (observed $\phi_{Tu}/\phi_{G}$: *E. coli* 3.9, *B. subtilis* 2.7, *V. natriegens* 3.3). A higher abundance is required for EF-Tu because it is bound to the different tRNAs, which effectively decreases the concentration by a factor of $n_{aa}\approx 20$ (see section Estimation of coarse-grained rates for derivation and discussion of why the factor is not equal to the number of different tRNAs). Taken together, our model offers straightforward explanations for the observed tIF stoichiometry.

For a few tlFs, the observed concentrations are two- to fivefold higher than the predicted optimal levels (e.g. EF-Ts, RF4, and IF1 in *Figure 4*). A potential explanation is that the corresponding reactions may not be binding or diffusion-limited, which would lead to a non-negligible fraction of tlFs sequestered at the catalytic step and thereby require higher total concentrations. Indeed, recent detailed modeling of the EF-Ts (*Hu et al., 2020*) cycle estimated only a small fraction (6% to 48%) of its abundance was in the free form in the cell, consistent with the large deviation we observe for this factor from our diffusion only prediction. Our optimization model can also be solved analytically in the non-binding-limited regime (*Table 1*), with the finite catalytic rate leading to an additional contribution of the form $\propto \ell\lambda^*/k_{cat}$. However, the numerical values for these solutions are in general difficult to obtain because the estimates for catalytic rates are sparse and often inconsistent with estimates of kinetics in live cells. As an example, median estimated aaRS catalytic rates (*Jeske et al., 2019*) measured in vitro is $\approx 3\ \text{s}^{-1}$, well below the *minimal* value of 15 s$^{-1}$, required to sustain translation flux at the measured value (Appendix 5), suggesting substantial deviation between in vitro and in vivo kinetics. While technically demanding, the fraction of free vs. bound factors can in principle be determined through live cell microscopy of tagged factors by partitioning the diffusive states of the tagged enzyme. Using that approach, *Volkov et al., 2018* estimated that EF-Tu was in its bound state <10% of the time (consistent with our diffusion-limited prediction closed to the observed value for this factor).

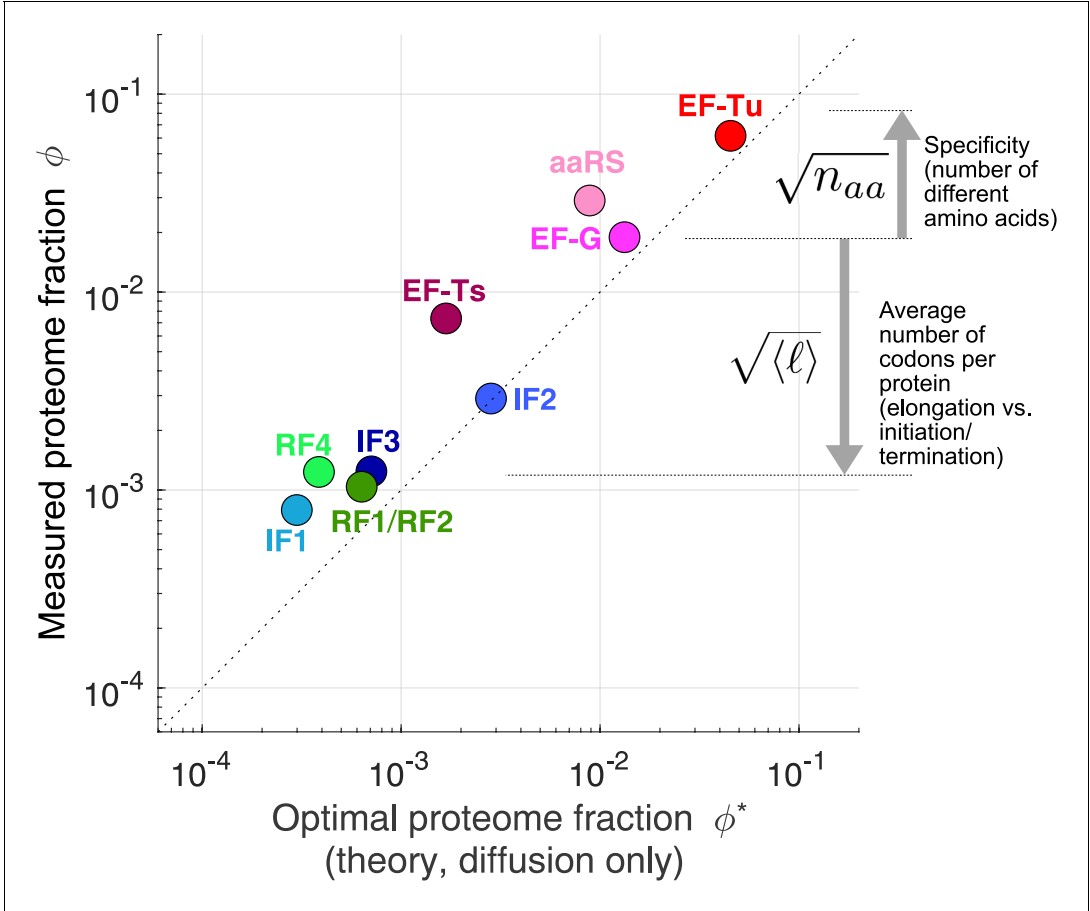

**Figure 4.** Predicted optimal abundance (no catalytic contribution, $k_{cat} \to \infty$) versus observed abundance. Measured proteome fractions are the average of *E. coli*, *B. subtilis*, *V. natriegens* (*Lalanne et al., 2018*). We note that given the sensitivity of the optimal aaRS abundance on the total tRNA/ribosome ratio (visually: yellow star's position in *Figure 3B* moves rapidly along x-axis upon changes in plateau of transition line), the prediction for aaRS should be interpreted with caution. Data and predicted values can be found in *Supplementary file 1* and *2*.
The online version of this article includes the following figure supplement(s) for figure 4:

**Figure supplement 1.** Measured and predicted proteome fraction for core translation factors in individual conditions.

**Figure supplement 2.** Expression stoichiometry of core translation factors in different species and at different growth rates.

Another potential explanation for the observed deviations from our predictions is that the selective pressure for these tIFs may be lower compared to the more highly expressed tIFs. This explanation is unlikely both because their stoichiometry are observed to be conserved (*Figure 1B*, *Figure 4—figure supplement 2*) and given that the expression of other lowly expressed tIFs (e.g. RF1, RF2, and individual aaRSs) has been shown to acutely affect cell growth (*Lalanne et al., 2021*; *Parker et al., 2020*). Nevertheless, the deviations from the predicted optimal levels suggest that a more refined model may be required than our first-principles derivation.

## Discussion

Despite the comprehensive characterization of their molecular mechanisms, the 'mixology' for the protein synthesis machineries inside living cells has remained elusive. Here, we establish a first-principles framework to provide analytical solutions for the growth-optimizing concentrations of translation factors. We find reasonable agreements between our parameter-free parsimonious predictions and the observed tIF stoichiometry (*Figure 4*). These results provide simple rationales for the hierarchy of expression levels, as well as insights into several construction principles for biological pathways.

An important implication from the agreement between observed stoichiometries and our predictions is that most tlFs are co-limiting for growth. Previous models have focused on expression optimization for the full translation sector, ribosomes (*Scott et al., 2010*; *Belliveau et al., 2021*), and the abundant elongation factors EF-Tu (*Ehrenberg and Kurland, 1984*; *Klumpp et al., 2013*). In a recent study, Hu and colleagues considered additional RNA components and EF-Ts in their optimization procedure (*Hu et al., 2020*). In line with the conclusions of these previous studies, our results demonstrate that multiple components of the translation machinery, regardless of their observed expression level, are simultaneously co-limiting for cell growth. By virtue of the interlocked translation cycles at steady state, the flux through every cycle must be matched. In our model, the optimality occurs when there are just enough tlFs to support the required flux in every cycle, such that the proteome fraction of free factors equals that of waiting ribosomes at that step (equipartition). If the concentration of any one tlF falls below the optimal point, it becomes the limiting factor for protein synthesis and growth. This result is supported by experimental evidence that slight knockdowns of individual RFs and aaRSs are detrimental to growth (*Parker et al., 2020*; *Lalanne et al., 2021*). Figuratively, the translation apparatus is analogous to a vulnerable supply chain, in which slowdown in any of the steps affects the full output.

In the binding-limited regime, the optimal tlF stoichiometry is independent of the specific growth rate (except for aaRS). This is consistent with the observation that relative tlF expression remains unchanged in *E. coli* in conditions with doubling times ranging from 20 min to 2 hr (*Lalanne et al., 2018*; *Li et al., 2014*; *Figure 4—figure supplement 2A*).

Our results are also consistent with the maintenance of the relative tlF expression across large phylogenetic distances even though the underlying regulation and cellular physiology has diverged (*Lalanne et al., 2018*; *Figure 1B*, and additional comparison to slow growing *C. crescentus* in *Figure 4—figure supplement 2A*). Under the assumption of diffusion-limited association to estimate parameters, the optimal tlF stoichiometry depends only on simple biophysical parameters, including protein sizes and diffusion constants, that are likely conserved in distant species. It remains to be determined if similar biophysical principles apply to the other pathways that also exhibit conserved enzyme expression stoichiometry.

In principle, our model can also make predictions on the growth defects at suboptimal tlF concentrations. However, experimentally testing these predictions will be difficult due to secondary effects of gene regulation that are not considered in our model near optimality. For example, we have recently shown that small changes in RF levels lead to idiosyncratic induction of the general stress response in *B. subtilis* due to a single ultrasensitive stop codon (*Lalanne et al., 2021*). As a result, the growth defect not only arises from reduced translation flux, but is in fact dictated by spurious regulatory connections that are normally not activated when tlF expression is at the optimum. We propose that tlF expression may be set at the optimal levels as our first-principles model suggests but entrenched by connections in the regulatory network. To predict the full expression-to-fitness landscape away from the optimum, a more comprehensive model may be required to take into account all the molecular interactions in the cell (*Karr et al., 2012*; *Macklin et al., 2020*).

Our coarse-graining approach has several limitations in its connection to detailed biochemical parameters. Foremost, coarse-grained association rate constants remain difficult to numerically estimate, and possibly neglect important features. In particular, given the sparsity of available in vivo rate constants, we estimate $\hat{k}_{on}$ for all tlFs reactions by scaling the measured TC association rate constant ($\hat{k}_{on}^{TC}$) by the respective diffusion coefficients. This approach generates more plausible values than the unrealistic overestimate from Smoluchowski theory (diffusion-limited rate for perfectly absorbing spheres, see Appendix 5). However, the simplifying assumptions that certain molecular properties of modeled reactions are similar (e.g. the size of the reactive surfaces, orientational constraints of the bimolecular interaction, and possible non-cognate binding events) may have to be modified for more detailed models. We also do not explicitly consider off-rates in our model. Instead, our parameters correspond to effective rate constants that account for possible sequential binding and unbinding events, that is, $\tilde{k}_{on} = k_{on}/n_{bind}$, with $n_{bind} = k_{cat}/(k_{cat} + k_{off})$. The effective association rate constants in our model thus contain information about catalytic and possible proofreading steps, which could be tlF-specific and are challenging to estimate. All these effects may contribute to the discrepancy between our predicted and observed tlF concentrations. As more physiological and molecular data become available, these tlF-specific features could be used to individually refine

our estimate for the association rates constants and our predictions. For example, elaborate calculations from structural data could account for rotational constraints (*Schlosshauer and Baker, 2004*), but are beyond the scope of the present work. Overall, we expect these tlF-specific corrections to be of limited influence on the final predictions due to the square-root dependence of the optimal expression (*Table 2*). We further note that a number of conclusions from our model, such as the factor of $\sqrt{\langle \ell \rangle}$ separating the optimal abundances of elongation from initiation/termination tlFs, are generic and do not depend on the specific association rates.

Taken together, our model provides the biophysical basis for the stoichiometry of translation factors in living cells. The first-principles approach complements more comprehensive models that include many biochemical parameters (*Hu et al., 2020*; *Vieira et al., 2016*), while providing intuitive rationales for the expression hierarchy. We anticipate that our approach will be generalizable to elucidate or design enzyme stoichiometry of other biological pathways, especially those whose activities are required for cell growth.

## Materials and methods

### Average number of codons per protein: $\langle \ell \rangle$
We calculate the average number of codons per protein, weighted by expression, as

$$\langle \ell \rangle := \frac{\sum_i e_i \ell_i}{\sum_i e_i}, \tag{16}$$

where $\ell_i$ is the number of codon for the protein product of gene $i$, and $e_i$ is the protein synthesis rate (as estimated from ribosome profiling [*Li et al., 2014*; *Lalanne et al., 2018*]) for gene $i$. For a stable proteome (in fast growing bacteria, the cell doubling time is shorter than the active degradation of most proteins [*Larrabee et al., 1980*]), the protein synthesis rate equals to the proteome mass fraction (*Li et al., 2014*). Changes in the expression of genes across growth conditions do not lead to substantial changes in $\langle \ell \rangle$. In *E. coli*, across growth conditions spanning $\approx$20 min doubling time to $\approx$120 min, $\langle \ell \rangle$ changes by about 20%. Specifically, we find $\langle \ell \rangle = 196, 210$, and $240$ in respectively MOPS complete ($\approx$20 min doubling time [*Li et al., 2014*]), MOPS minimal ($\approx$56 min doubling time [*Li et al., 2014*]), and NQ1390 forced glucose limitation ($\approx$120 min doubling time [*Mori et al., 2021*]), based on ribosome profiling data. Here for simplicity, we take $\langle \ell \rangle \approx 200$ throughout.

### Conversion between concentration and proteome fraction
Throughout, we use both units of concentration (molar), denoted as for example, $[A]$ for protein $A$, and proteome fraction, denoted by $\phi_A$ (*Scott et al., 2010*). The correspondence between the two is $\phi_A = [A]\ell_A/P$, where $\ell_A$ is the number of amino acid in protein $A$, and $P$ is the in-protein amino acid concentration in the cell. $P \approx 2.6 \times 10^6$ μM, and has a value approximately independent of growth rate (*Klumpp et al., 2013*; *Bremer and Dennis, 2008*). This change in units also relates to how association constants are defined in units of proteome fraction: $\hat{k}_{on}[A] := k_{on}\phi_A$, where the hat $\hat{}$ refers to the association constant in usual units of μM$^{-1}$ s$^{-1}$ (used to connect to empirical data). Hence, $k_{on} := \hat{k}_{on}P\ell^{-1}$ is the rescaled association rate in units of proteome fraction.

### Equality of ribosome flux in steady-state
In steady-state exponential growth, the ribosome flux in and out of each intermediate state is equal to the total flux. This results from the fact that no ribosome can accumulate in any intermediate state. Since the flux out of state $i$ is given by $\phi_{ribo}^i/\tau_i$, we must have:

$$\frac{\lambda \ell_{ribo}}{\langle \ell \rangle} = \frac{\phi_{ribo}^{act}}{\tau_{trl}} = \frac{\phi_{ribo}^{ini}}{\tau_{ini}} = \frac{\phi_{ribo}^{el}}{\tau_{el}} = \frac{\phi_{ribo}^{ter}}{\tau_{ter}}. \tag{17}$$

As a consequence, the proportion of ribosome in each state is equal to the proportion of time spent at that given step, for example for translation initiation:

$$\frac{\phi_{ribo}^{ini}}{\phi_{ribo}^{act}} = \frac{\tau_{ini}}{\tau_{ini} + \tau_{el} + \tau_{ter}}.$$

## Protein production flux and growth rate

In order to write the mass action kinetic scheme for more complex models, it is useful to recast our framework in terms of the protein number production flux $J$, defined as the number of full length proteins produced per cell volume per unit time. The production of each protein requires a ribosome to go through the full synthesis cycle, and as such $J$ provides a convenient quantity in mass action schemes formulated in molar units.

In steady-state of exponential growth (*Monod, 1949*; *Scott et al., 2010*; *Dai et al., 2016*), there is a direct relationship between the growth rate $\lambda$ (defined through $\mathrm{d}N/\mathrm{d}t = \lambda N$, where $N$ is the number of cells per unit volume of culture) and the protein production flux $J$. Explicitly, the protein mass accumulation rate is $\lambda M$, where $M$ is the total protein mass per unit volume of culture. If $V$ is the mean cell volume, then $\lambda M/V = N m_{aa} \langle \ell \rangle J$, where $m_{aa}$ is the mean amino acid mass. Defining $P := M/(m_{aa}NV)$, the in-protein amino acid concentration per cell (Materials and methods, section Conversion between concentration and proteome fraction), the connection between protein production flux $J$ and growth rate $\lambda$ is then $J = \frac{P\lambda}{\langle \ell \rangle}$. This relationship will be used to convert between molar and proteome fraction in some equations below.

## Summary of optimal solutions

Solutions for the factor predicted optimal abundances as a function of effective biochemical parameters and the growth rate at the optimum, are presented in *Table 1*. The table breaks down terms in each solution by categories: direct diffusion term (arising from diffusive search time), catalytic sequestration, and delay incurred by the diffusion of other proteins in part of the cycle of the factor of interest. Solutions are listed in terms of on-rate $\hat{k}_{on}$ (units of $\mu M^{-1} s^{-1}$). The aaRS solution follows a different form:

$$\phi_{aaRS}^* = \frac{n_{aa}\ell_{aaRS}\lambda^*}{\hat{k}_{on}^{aaRS}P\Delta_{tRNA}^*} + \frac{\ell_{aaRS}\lambda^*}{k_{cat}^{aaRS}},$$

$$\text{with } \Delta_{tRNA}^* := \frac{\text{tRNA}_{tot}}{P} - \frac{\lambda^*}{k_{on}^{TC}\phi_{TC}^*} - \frac{2\lambda^*}{k_{el}^{max}} - \frac{\phi_{TC}^*}{\ell_{Tu}} - \frac{\lambda^*}{k_{cat}^{aaRS}}, \quad \text{and} \quad \phi_{TC}^* := \sqrt{\frac{n_{aa}\ell_{ribo}\ell_{Tu}\lambda^*}{\hat{k}_{on}^{TC}P}}. \tag{18}$$

## Acknowledgements

We thank R Battaglia, J Cascino, M Gill, M Parker, D Parker, and G Schmidt for critical reading of the manuscript, and all members of the Li lab for discussion. This research was supported by NIH grant R35GM124732, the NSF CAREER Award, the Smith Odyssey Award, the Pew Biomedical Scholars Program, a Sloan Research Fellowship, the Searle Scholars Program, the Smith Family Award for Excellence in Biomedical Research; NSERC doctoral Fellowship and HHMI International Student Research Fellowship (to J-BL).

## Additional information

### Funding

| Funder | Grant reference number | Author |
| --- | --- | --- |
| National Institutes of Health | R35GM124732 | Gene-Wei Li |
| National Science Foundation | MCB 1844668 | Gene-Wei Li |
| Richard and Susan Smith Family Foundation | Smith Odyssey Award and Smith Family Award | Gene-Wei Li |
| Pew Charitable Trusts | Pew Scholar | Gene-Wei Li |
| Alfred P. Sloan Foundation | Sloan Research Fellowship | Gene-Wei Li |
| Kinship Foundation | Searle Scholar | Gene-Wei Li |

| National Research Council Canada | Doctoral fellowship | Jean-Benoît Lalanne |
| Howard Hughes Medical Institute | International Student Fellowship | Jean-Benoît Lalanne |

The funders had no role in study design, data collection and interpretation, or the decision to submit the work for publication.

### Author contributions
Jean-Benoît Lalanne, Conceptualization, Formal analysis, Visualization, Writing - original draft, Writing - review and editing; Gene-Wei Li, Conceptualization, Supervision, Writing - original draft, Project administration, Writing - review and editing

### Author ORCIDs
Jean-Benoît Lalanne https://orcid.org/0000-0001-8753-0669
Gene-Wei Li https://orcid.org/0000-0001-7036-8511

### Decision letter and Author response
Decision letter https://doi.org/10.7554/eLife.69222.sa1
Author response https://doi.org/10.7554/eLife.69222.sa2

## Additional files

### Supplementary files
• Supplementary file 1. Proteome synthesis fraction (in %) of core mRNA translation factors for species and growth conditions with fast growth estimated from ribosome profiling data (*Li et al., 2014*; *Lalanne et al., 2018*).

• Supplementary file 2. Diffusion-limited optima predicted for translation factors for fast-growth conditions.

• Supplementary file 3. Proteome synthesis fraction (in %) of core mRNA translation factors for species/conditions with slower growth estimated from ribosome profiling. Ribosome profiling data: *E. coli* (MOPS minimal [*Li et al., 2014*], M9 glucose [*Mori et al., 2021*], *C. crescentus* [*Schrader et al., 2014*], with synthesis rates estimated in *Lalanne et al., 2018*).

• Supplementary file 4. Diffusion-limited optima predicted for translation factors for slower growth conditions.

• Transparent reporting form

### Data availability
Already publicly available ribosome profiling datasets were used (GEO accessions GSE95211, GSE53767, and GSE139983). Computer scripts (Matlab) used for this study were submitted with the present work as Figure 3—source code 1. Supplementary files 1-4 contain the numerical data to reproduce figures.

The following previously published datasets were used:

| Author(s) | Year | Dataset title | Dataset URL | Database and Identifier |
|---|---|---|---|---|
| Lalanne JB, Taggart JC, Guo MS, Herzel L, Schieler A, Li GW | 2018 | Data from: Evolutionary Convergence of Pathway-specific Enzyme Expression Stoichiometry | https://www.ncbi.nlm.nih.gov/geo/query/acc.cgi?acc=GSE95211 | NCBI Gene Expression Omnibus, GSE95211 |
| Li G, Burkhardt D, Gross CA, Weissman JS | 2014 | Data from: Absolute quantification of protein production reveals principles underlying protein synthesis rates | https://www.ncbi.nlm.nih.gov/geo/query/acc.cgi?acc=GSE53767 | NCBI Gene Expression Omnibus, GSE53767 |

Mori M, Zhang Z, Banaei-Esfahani A, Lalanne JB, Okano H, Collins BC, Schmidt A, Schubert OT, Lee DS, Li GW, Aebersold R, Hwa T, Ludwig C | 2021 | Data from: From coarse to fine: The absolute *Escherichia coli* proteome under diverse growth conditions | https://www.ncbi.nlm.nih.gov/geo/query/acc.cgi?acc=GSE139983 | NCBI Gene Expression Omnibus, GSE139983

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

## Appendix 1

## Coarse-grained transition times: models of ribosome traffic

Our coarse-grained model of ribosome transitions between categories of initiation, elongation, and termination need to be distinguished from the individual molecular times of the respective steps in one important regard: ribosome traffic on mRNAs can lead to effective delays arising from transient queuing. For example, if translation termination is slow and ribosomes start to pile up and form queues upstream of stop codons on mRNAs, the molecular time of termination (time between ribosome arrival to the stop codon and its recycling to the free ribosome pool) will not be a correct reflection of the actual termination time of a ribosome, because of the additional wait time in the queue. A similar argument can be made for transient queuing forming in the body of genes for elongating ribosomes.

We connect these two (molecular and coarse-grained) levels of description by noting that our mass action schemes relating the translation factor abundance to the times of the specific steps can be used as input parameters in traffic models of ribosome movement along mRNAs taking into account possible many-body interactions (e.g. totally asymmetric exclusion processes [*Shaw et al., 2003*; *Kavčič et al., 2020*]). Solving these traffic models can then be used to obtain transition times in our coarse-grained translation cycle model. As we show below, corrections arising from transient queuing are small (for endogenous translation factor abundances) based on current estimates the absolute rates of initiation, elongation, and termination, on individual mRNAs, such that stochastic queuing does not play a dominant role in determining optimal translation factor expression levels.

As a first example, we relate the on-stop codon molecular termination time $\tau_{ter}$, which we obtain from solving our mass action scheme (see *Equation 6*), to the termination time in presence of queuing: $\tau_{ter}^{full}$. The difference between the two, as described above, being related to possible queues upstream of stop codons leading to further delays in the process of translation termination, and thus to a longer termination time than that of the molecular on-stop codon termination. The delay factor will be denoted $\mathcal{Q}(\tau_{ter})$, defined through:

$$\tau_{ter}^{full} := \tau_{ter}\,\mathcal{Q}(\tau_{ter}).$$

To derive the expression for the $\mathcal{Q}$ factor, note that in steady-state, ribosome numbers in a given state is directly proportional to the time to transition out of that state. Let $m_i$ be the mRNA concentration for gene $i$ in the cell, $n_{ter}(\alpha_i, \tau_{ter})$ the number of terminating ribosomes (including queues if present) on a transcript with per mRNA translation initiation rate (i.e. translation efficiency [*Li, 2015*]) $\alpha_i$, then:

$$\tau_{ter}^{full} \propto \sum_i m_i\, n_{ter}(\alpha_i, \tau_{ter}),$$

whereas

$$\tau_{ter} \propto \sum_i m_i\, n_{ter}^{\emptyset\mathcal{Q}}(\alpha_i, \tau_{ter}),$$

with $n_{ter}^{\emptyset\mathcal{Q}}(\alpha_i, \tau_{ter})$ the average number of terminating ribosomes on a transcript with translation efficiency $\alpha_i$, assuming no queue upstream of the stop codon. Note that $n_{ter}(\alpha_i, \tau_{ter}) \geq n_{ter}^{\emptyset\mathcal{Q}}(\alpha_i, \tau_{ter})$ (the differences being queued ribosomes). Hence, the queuing factor $\mathcal{Q}$ is:

$$\mathcal{Q}(\tau_{ter}) := \frac{\tau_{ter}^{full}}{\tau_{ter}} = \frac{\sum_i m_i\, n_{ter}(\alpha_i, \tau_{ter})}{\sum_i m_i\, n_{ter}^{\emptyset\mathcal{Q}}(\alpha_i, \tau_{ter})}.$$

Formally, $n_{ter}$ can be obtained by solving a TASEP model (*Shaw et al., 2003*), but a simplified queue model (*Bergmann and Lodish, 1979*; *Lalanne et al., 2021*) disregarding spatial information recapitulates the statistics of queue formation (as verified by full stochastic simulations, data not shown). The state space of the queue model is the number of ribosomes $N$ in the queue. Ribosomes arrive at a rate $\alpha$ (initiation rate on the transcript), and leave at the molecular termination rate $\tau_{ter}^{-1}$. The ribosome arrival rate at the queue is rigorously correct in steady-state, unless the queue

becomes large enough to affect the initiation process (fully jammed transcript), or RNA degradation. The stochastic process (away from the jammed state) is then described by: $N \rightarrow N+1$ at rate $\alpha$, and $N \rightarrow N-1$ at rate $\tau_{ter}^{-1}$ for $N>0$. The probability for the queue to have $N$ ribosomes, $P(N)$, can be obtained as the steady-state from the resulting master equation, leading to a geometric series: $P(N) = (\alpha\tau_{ter})^N(1-\alpha\tau_{ter})$. Hence, the prevalence of higher order queues scales as the ratio of the initiation to termination rate on the transcript. The average queue size, corresponding to $n_{ter}(\alpha_i, \tau_{ter})$, is:

$$n_{ter}(\alpha_i, \tau_{ter}) \approx \begin{cases} \dfrac{\tau_{ter}\alpha_i}{1-\tau_{ter}\alpha_i}, & \tau_{ter}^{-1} \geq \alpha_i(1+\ell_{footprint}\ell_i^{-1}), \\ \dfrac{\ell_i}{\ell_{footprint}}, & \tau_{ter}^{-1} < \alpha_i(1+\ell_{footprint}\ell_i^{-1}). \end{cases}$$

Above, the solution of the simple model is truncated at the value where the transcript becomes fully jammed with $\ell_i/\ell_{footprint}$ ribosomes ($\ell_i$ and $\ell_{footprint}$ being the size of gene $i$ and the size occupied by a ribosome respectively). The no queue ribosome number is simply equal to a model where queues with $N>1$ do not arise, hence $n_{ter}^{\emptyset\mathcal{Q}}(\alpha_i, \tau_{ter}) = \alpha_i\tau_{ter}$. Therefore, the queuing factor, under the stated assumptions (and assuming no transcript is in the jammed state), is

$$\mathcal{Q}(\tau_{ter}) \approx \frac{\sum_i m_i \frac{\alpha_i}{1-\tau_{ter}\alpha_i}}{\sum_i m_i \alpha_i}.$$

Expanding for fast termination gives $\mathcal{Q}-1 = \frac{\tau_{ter}\langle\alpha^2\rangle}{\langle\alpha\rangle}$ as the leading order correction, where the averages are weighted by mRNA levels. The above was derived assuming exponentially distributed initiation and termination times, but could be modified to account for more complex dynamics of the initiation and initiation steps.

The queuing factor can be estimated based on absolute measurements of the initiation and termination rates in cells. *Kennell and Riezman, 1977* estimate 3.2 s between initiation events on the *lacZ* mRNA (at 48 min per cell doubling). *Bremer and Dennis, 2008* estimate 1 s per ribosome initiation events at 20 min doubling time. Recent calibrated high-throughput measurements report a genome-wide median of 5.6 s per initiation events (*Gorochowski et al., 2019*). To our knowledge, estimation of absolute in vivo termination rates have not been performed, but we can estimate bounds. Indirect assessment based on steady-state protein production measurements place the fraction of actively elongating ribosome at about 95% (*Dai et al., 2016*). Assuming (upper bound) that the 5% of non elongating ribosomes are in the process of termination would give a termination time of $5\% \times 11.1s \approx 0.6s$ (fraction of ribosomes in a given state equal to the ratio of transition times), where we have used that the elongation time of an average protein is about 11.1 s ($200/18\,s^{-1}$) at fast growth (*Dai et al., 2016*). This upper bound is still much smaller than the reported median initiation time, suggesting that the queuing factor for termination is small. As additional support to the view that translation is far from being termination limited, small that queues at stop codons are only globally observed in ribosome profiling upon severe perturbations (*Kavčič et al., 2020*; *Baggett et al., 2017*; *Mangano et al., 2020*; *Saito et al., 2020*; *Lalanne et al., 2021*).

With regard to translation elongation, transient queuing in the body of gene can also lead to a difference between molecular and coarse-grained transition times in our model. However, the fraction of ribosomes transiently stalled due to this queuing scales as $\alpha\tau_{aa}$ in the low-density phase (defined by requirements $\alpha\tau_{ter} < 1$ and $\alpha\tau_{aa} < (1+\sqrt{\ell_{footprint}})^{-1} \approx 0.25$) of the TASEP model (*Shaw et al., 2003*). Since measured estimates place $\alpha\tau_{aa} \sim 0.01$ (*Dai et al., 2016*; *Gorochowski et al., 2019*), we do not consider the queuing effect for elongating ribosomes within our optimization framework for elongation factor abundances.

## Appendix 2

### Translation termination
#### Omitted molecular details

The kinetic scheme presented in *Figure 2A* does not include some known molecular details of translation termination. For example, GTPase RF3 has been shown to catalyze the release of RF1/RF2 post peptide hydrolysis and to effectively prevent rebinding to empty A site ribosome without peptide (*Pavlov et al., 1997*). RF3 is not included in our model given our desire for a parsimonious description and due to the absence of identifiable homologs in multiple bacteria (e.g. *B. subtilis*) (*Margus et al., 2007*). Our scheme aggregates the RF1/RF2 recycling rate with the catalytic rate, and further assume a unidirectional reaction without rebinding (consistent with a lower bound), effectively taking into account the action of RF3. In addition, translocation factor EF-G is known to be implicated in ribosome recycling via translocation post RF4 binding (*Zavialov et al., 2005*). We assume EF-G's abundance requirement toward the function of termination to be a minor fraction of its total requirement (non-sense to sense codons ≈ 0.5%) and to be non-limiting for this step. We thus coarse-grain EF-G's role in ribosome recycling through an effective catalytic rate for RF4, see *Borg et al., 2016* for details of EF-G's involvement in ribosome recycling. As another example of simplification in our coarse-graining, we also do not explicitly model RF1/RF2's post-translational modification by methyltransferase PrmC (*Mora et al., 2007*). Thus, the activity of the RFs within our description to correspond to the average within a possibly heterogeneous pool of modified and unmodified factors in the cell.

### Non binding-limited regime (one stop codon)

If translation termination is not diffusion limited, terms corresponding to the finite catalytic times must be included in addition to the diffusive contributions in the termination time (*Equation 6*). Under our simplified scheme (*Figure 2A*) and with a single stop codons (grouping RF1 and RF2), the molecular termination time is then sum of the four separate times corresponding to distinct events:

$$\tau_{ter} = \frac{1}{k_{on}^{RFI} \phi_{RFI}^{free}} + \frac{1}{k_{cat}^{RFI}} + \frac{1}{k_{on}^{RF4} \phi_{RF4}^{free}} + \frac{1}{k_{cat}^{RF4}}$$

The two novelties compared to the diffusion-limited regime (*Equation 6*) are: (1) addition of the catalytic times $k_{cat}^{-1}$ for the two steps, and importantly (2) the mass action diffusion terms now involve the free concentration of release factors. Generally, the free concentration of the tlFs can be obtained by solving the steady-state solutions of kinetic schemes under constraints imposed by conservation equations. The examples in e.g., sections B.3, C.3, and D.1 below provide the mathematical details associated with the procedure.

Here, the difference between the total and free concentration of release factor arises from the finite catalytic turnover of the enzymes, and corresponds to the concentration of ribosome bound release factors. Given the flux $J$ through the system in steady-state of growth, the concentration of ribosome bound release factor (e.g. for RF4) is $J/k_{cat}^{RF4}$, which becomes $\frac{\ell_{RF4}\lambda}{\langle\ell\rangle k_{cat}^{RF4}}$ upon converting to proteome fraction. This quantity sets the absolute minimum for the release factor abundance necessary to sustain growth $\lambda$ for a given $k_{cat}$. The free concentrations for the release factors are then:

$$\phi_{RFI}^{free} = \phi_{RFI} - \frac{\ell_{RFI}\lambda}{\langle\ell\rangle k_{cat}^{RFI}}, \quad \phi_{RF4}^{free} = \phi_{RF4} - \frac{\ell_{RF4}\lambda}{\langle\ell\rangle k_{cat}^{RF4}}. \tag{19}$$

Hence, the final solution for the steady-state termination time as a function of the total abundance of the release factors and growth rate is:

$$\tau_{ter} = \frac{1}{k_{on}^{RFI}\left(\phi_{RFI} - \frac{\ell_{RFI}\lambda}{\langle\ell\rangle k_{cat}^{RFI}}\right)} + \frac{1}{k_{cat}^{RFI}} + \frac{1}{k_{on}^{RF4}\left(\phi_{RF4} - \frac{\ell_{RF4}\lambda}{\langle\ell\rangle k_{cat}^{RF4}}\right)} + \frac{1}{k_{cat}^{RF4}}.$$

The relationship above, between termination time, total tlF abundance, and growth rate $\lambda$ closes the solution of the kinetic scheme. Substituting the above in the optimality condition (*Equation 5*) leads to the solution:

$$\phi_{RFI}^* = \sqrt{\frac{\ell_{ribo}\lambda^*}{\langle\ell\rangle k_{on}^{RFI}} + \frac{\ell_{RFI}\lambda^*}{\langle\ell\rangle k_{cat}^{RFI}}}, \quad \phi_{RF4}^* = \sqrt{\frac{\ell_{ribo}\lambda^*}{\langle\ell\rangle k_{on}^{RF4}} + \frac{\ell_{RF4}\lambda^*}{\langle\ell\rangle k_{cat}^{RF4}}}. \tag{20}$$

The additional terms $\propto \lambda^*$ correspond to the contribution to the optimal abundance arising from the finite catalytic rates, no present in the diffusion limited regime (*Equation 7*).

## Full three stop codons model

The full model with three different stop codons (UAA, UGA, UAG) and RF1/RF2 with different specificities (RF1: UAA, UAG; RF2: UAA, UGA) can also be solved exactly, leading to a small correction on the summed optimal abundance for RF1 and RF2 of $\sqrt{1 + 2\sqrt{f_{UAG}f_{UGA}}} < 1.05$ (fast growing species considered, where $f_{UAG}$ and $f_{UGA}$ are the fractional fluxes through the RF1 and RF2 stop codons, respectively) compared to the single stop codon optimum derived above ($\phi_{RFI}^*$, *Equation 20*). We provide details below. With three stop codons, the coarse-grained reaction scheme is shown in *Appendix 2—figure 1*. The relevant chemical species and parameters are listed in *Appendix 2—table 1*.

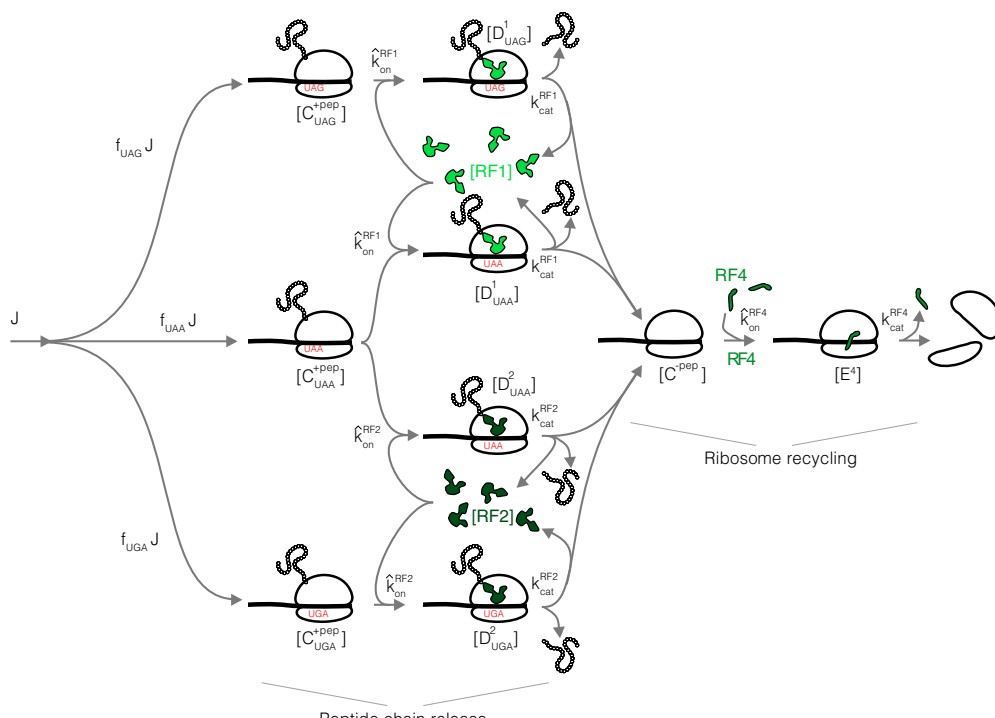

**Appendix 2—figure 1.** Coarse-grained translation termination scheme with three stop codons and RF1/RF2.

**Appendix 2—table 1.** Chemical species and parameters in three stop codons termination model.

| Variable | Description |
| --- | --- |
| $[C_{UAA}^{+pep}]$ | Ribosomes at UAA with peptide chain [μM] |
| $[C_{UAG}^{+pep}]$ | Ribosomes at UAG with peptide chain [μM] |
| $[C_{UGA}^{+pep}]$ | Ribosomes at UGA with peptide chain [μM] |
| $[D_{UAA}^1]$ | Ribosomes at UAA with peptide chain and RF1 bound [μM] |
| $[D_{UAG}^1]$ | Ribosomes at UAG with peptide chain and RF1 bound [μM] |
| $[D_{UAA}^2]$ | Ribosomes at UAA with peptide chain and RF2 bound [μM] |

*Continued on next page*

*Appendix 2—table 1 continued*

| Variable | Description |
|---|---|
| $[D^2_{UGA}]$ | Ribosomes at UGA with peptide chain and RF2 bound [µM] |
| $[C^{-pep}]$ | Ribosomes at all stops without peptide chain [µM] |
| $[E^4]$ | Ribosomes at all stops without peptide chain and RF4 bound [µM] |
| $[RF1]$ | Free RF1 [µM] |
| $[RF2]$ | Free RF2 [µM] |
| $[RF4]$ | Free RF4 [µM] |
| $J^{UAA} = f_{UAA}J$ | Ribosome flux through UAA [µM s$^{-1}$] |
| $J^{UAG} = f_{UAG}J$ | Ribosome flux through UAG [µM s$^{-1}$] |
| $J^{UGA} = f_{UGA}J$ | Ribosome flux through UGA [µM s$^{-1}$] |
| $\hat{k}^{RF1}_{on}$ | On-rate for RF1 [µM$^{-1}$ s$^{-1}$] |
| $\hat{k}^{RF2}_{on}$ | On-rate for RF2 [µM$^{-1}$ s$^{-1}$] |
| $\hat{k}^{RF4}_{on}$ | On-rate for RF4 [µM$^{-1}$ s$^{-1}$] |
| $k^{RF1}_{cat}$ | Catalytic rate for RF1 [s$^{-1}$] |
| $k^{RF2}_{cat}$ | Catalytic rate for RF2 [s$^{-1}$] |
| $k^{RF4}_{cat}$ | Catalytic rate for RF4 [s$^{-1}$] |
| $RF1_{tot}$ | Total RF1 [µM] |
| $RF2_{tot}$ | Total RF2 [µM] |
| $RF4_{tot}$ | Total RF4 [µM] |

The corresponding mass action system of equations for peptide release:

$$\frac{d[C^{+pep}_{UAA}]}{dt} = f_{UAA}J - [C^{+pep}_{UAA}]\left(\hat{k}^{RF1}_{on}[RF1] + \hat{k}^{RF2}_{on}[RF1]\right),$$

$$\frac{d[C^{+pep}_{UAG}]}{dt} = f_{UAG}J - \hat{k}^{RF1}_{on}[C^{+pep}_{UAG}][RF1],$$

$$\frac{d[C^{+pep}_{UGA}]}{dt} = f_{UGA}J - \hat{k}^{RF2}_{on}[C^{+pep}_{UGA}][RF1],$$

$$\frac{d[D^1_{UAA}]}{dt} = \hat{k}^{RF1}_{on}[RF1][C^{+pep}_{UAA}] - k^{RF1}_{cat}[D^1_{UAA}],$$

$$\frac{d[D^1_{UAG}]}{dt} = \hat{k}^{RF1}_{on}[RF1][C^{+pep}_{UAG}] - k^{RF1}_{cat}[D^1_{UAG}],$$

$$\frac{d[D^2_{UAA}]}{dt} = \hat{k}^{RF2}_{on}[RF2][C^{+pep}_{UAA}] - k^{RF1}_{cat}[D^2_{UAA}],$$

$$\frac{d[D^2_{UGA}]}{dt} = \hat{k}^{RF2}_{on}[RF2][C^{+pep}_{UGA}] - k^{RF1}_{cat}[D^2_{UGA}],$$

$$\frac{d[RF1]}{dt} = -\hat{k}^{RF1}_{on}[RF1]\left([C^{+pep}_{UAA}] + [C^{+pep}_{UAG}]\right) + k^{RF1}_{cat}\left([D^1_{UAA}] + [D^1_{UAG}]\right),$$

$$\frac{d[RF2]}{dt} = -\hat{k}^{RF2}_{on}[RF2]\left([C^{+pep}_{UAA}] + [C^{+pep}_{UGA}]\right) + k^{RF2}_{cat}\left([D^2_{UAA}] + [D^2_{UGA}]\right).$$

And for ribosome recycling:

$$\frac{d[C^{-pep}]}{dt} = k^{RF1}_{cat}\left([D^1_{UAA}] + [D^1_{UAG}]\right) + k^{RF2}_{cat}\left([D^2_{UAA}] + [D^2_{UGA}]\right) - \hat{k}^{RF4}_{on}[C^{-pep}][RF4],$$

$$\frac{d[E^4]}{dt} = \hat{k}^{RF4}_{on}[C^{-pep}][RF4] - k^{RF4}_{cat}[E^4],$$

$$\frac{d[RF4]}{dt} = -\hat{k}^{RF4}_{on}[C^{-pep}][RF4] + k^{RF4}_{cat}[E^4].$$

The conservation equations for RF1, RF2 and RF4 are:

$$
\begin{aligned}
RF1_{tot} &= [RF1] + [D^1_{UAA}] + [D^1_{UAG}],\\
RF2_{tot} &= [RF2] + [D^2_{UAA}] + [D^2_{UGA}],\\
RF4_{tot} &= [RF4] + [E^4].
\end{aligned}
$$

With a more complex scheme such as the one above, the optimization problem can be solved in three steps. First, we obtain the steady-state concentration of the chemical species. Second, we determine the effective coarse-grained termination time. Finally, the optimal abundance is found by substituting the termination time in the optimality condition (*Equation 5*), and solving the resulting system of equation.

## Steady-state concentrations for RFs

Note that the RF1/RF2 and RF4 completely decouple, and that the solution for RF4 is identical to the one stop codon case solved above (section Non binding-limited regime [one stop codon]). For peptide chain release, the steady-state of the system can be solved by expressing the all chemical species in terms of $[RF1]$, and $[RF2]$:

$$
\begin{aligned}
[C^{+pep}_{UAA}] &= \frac{f_{UAA}J}{\hat{k}^{RF1}_{on}[RF1] + \hat{k}^{RF2}_{on}[RF2]}\\
[D^1_{UAA}] &= f_{UAA}\frac{J}{k^{RF1}_{cat}}\left(\frac{\hat{k}^{RF1}_{on}[RF1]}{\hat{k}^{RF1}_{on}[RF1] + \hat{k}^{RF2}_{on}[RF2]}\right),\\
[D^2_{UAA}] &= f_{UAA}\frac{J}{k^{RF2}_{cat}}\left(\frac{\hat{k}^{RF2}_{on}[RF2]}{\hat{k}^{RF1}_{on}[RF1] + \hat{k}^{RF2}_{on}[RF2]}\right),\\
[C^{+pep}_{UAG}] &= \frac{f_{UAG}J}{\hat{k}^{RF1}_{on}[RF1]},\quad [C^{+pep}_{UGA}] = \frac{f_{UGA}J}{\hat{k}^{RF2}_{on}[RF2]},\quad [D^1_{UAG}] = f_{UAG}\frac{J}{k^{RF1}_{cat}},\quad [D^2_{UGA}] = f_{UGA}\frac{J}{k^{RF2}_{cat}}.
\end{aligned}
$$

(21)

Substituting these in the conservation equations for RF1 and RF2 leads to a closed system in terms of $[RF1]$ and $[RF2]$:

$$
\begin{aligned}
RF1_{tot} &= [RF1]\left[1 + f_{UAA}\frac{J}{k^{RF1}_{cat}}\left(\frac{\hat{k}^{RF1}_{on}}{\hat{k}^{RF1}_{on}[RF1] + \hat{k}^{RF2}_{on}[RF2]}\right)\right] + f_{UAG}\frac{J}{k^{RF1}_{cat}},\\
RF2_{tot} &= [RF2]\left[1 + f_{UAA}\frac{J}{k^{RF2}_{cat}}\left(\frac{\hat{k}^{RF2}_{on}}{\hat{k}^{RF1}_{on}[RF1] + \hat{k}^{RF2}_{on}[RF2]}\right)\right] + f_{UGA}\frac{J}{k^{RF2}_{cat}}.
\end{aligned}
$$

Under the assumption of identical biochemical properties for RF1 and RF2, namely $k^{RF1}_{cat} = k^{RF2}_{cat} := k^{RFI}_{cat}$ and $\hat{k}^{RF1}_{on} = \hat{k}^{RF2}_{on} := \hat{k}^{RFI}_{on}$, the total free concentration of RF1 and RF2 simplifies to: $[RF1] + [RF2] = RF1_{tot} + RF2_{tot} - \frac{J}{k^{RFI}_{cat}}$, where we used $f_{UAA} + f_{UAG} + f_{UGA} = 1$ (by definition). Using this relation to eliminate $[RF2]$ from the $[RF1]$ equation (and vice-versa), we obtain, upon conversion to proteome fraction:

$$
\begin{aligned}
\phi^{free}_{RF,tot} &:= \phi_{RF1} + \phi_{RF2} - \frac{\ell_{RFI}\lambda}{\langle\ell\rangle k^{RFI}_{cat}},\\
\phi^{free}_{RF1} &= \chi_{RF1}\phi^{free}_{RF,tot},\quad \phi^{free}_{RF2} = \chi_{RF2}\phi^{free}_{RF,tot},
\end{aligned}
$$

(22)

where

$$
\begin{aligned}
\chi_{RF1} &:= \frac{\phi_{RF1} - \frac{\ell_{RFI}\lambda}{\langle\ell\rangle k^{RFI}_{cat}}f_{UAG}}{\left(\phi_{RF1} - \frac{\ell_{RFI}\lambda}{\langle\ell\rangle k^{RFI}_{cat}}f_{UAG}\right) + \left(\phi_{RF2} - \frac{\ell_{RFI}\lambda}{\langle\ell\rangle k^{RFI}_{cat}}f_{UGA}\right)},\\
\chi_{RF2} &:= \frac{\phi_{RF2} - \frac{\ell_{RFI}\lambda}{\langle\ell\rangle k^{RFI}_{cat}}f_{UGA}}{\left(\phi_{RF1} - \frac{\ell_{RFI}\lambda}{\langle\ell\rangle k^{RFI}_{cat}}f_{UAG}\right) + \left(\phi_{RF2} - \frac{\ell_{RFI}\lambda}{\langle\ell\rangle k^{RFI}_{cat}}f_{UGA}\right)}.
\end{aligned}
$$

These constitute the steady-state solutions of the system of equation.

## Coarse-grained translation termination time

In order to obtain an expression for the termination time (peptide release portion), needed to determine the optimal RF abundance (i.e. to substitute in *Equation 5*), the peptide chain release contribution arises from the ribosome containing species listed in *Equation 21*, which sum to (under the assumption of identical biochemical properties for RF1/RF2):

$$
\begin{aligned}
[R_{ter}^{pep}] &= [C_{UAA}^{+pep}] + [C_{UAG}^{+pep}] + [C_{UGA}^{+pep}] + [D_{UAA}^1] + [D_{UAG}^1] + [D_{UAA}^2] + [D_{UGA}^2], \\
[R_{ter}^{pep}] &= J\left( \frac{f_{UAG}}{\hat{k}_{on}^{RFI}[RF1]} + \frac{f_{UGA}}{\hat{k}_{on}^{RFI}[RF2]} + \frac{f_{UAA}}{\hat{k}_{on}^{RFI}([RF1]+[RF2])} + \frac{1}{k_{cat}^{RFI}} \right).
\end{aligned}
$$

Upon conversion to proteome fraction, the above becomes:

$$
\phi_{ribo}^{pep} = \frac{\ell_{ribo}}{\langle\ell\rangle}\lambda\left( \frac{f_{UAG}}{k_{on}^{RFI}\phi_{RF1}^{free}} + \frac{f_{UGA}}{k_{on}^{RFI}\phi_{RF2}^{free}} + \frac{f_{UAA}}{k_{on}^{RFI}\left(\phi_{RF1}^{free} + \phi_{RF2}^{free}\right)} + \frac{1}{k_{cat}^{RFI}} \right) := \frac{\ell_{ribo}}{\langle\ell\rangle}\lambda\,\tau_{pep}.
$$

The bracketed term corresponds to the coarse-grained time associated with peptide chain release $\tau_{pep}$, and the free concentrations are given by *Equations 22*.

## Optimal abundances for RF1/RF2

The solved concentrations in steady-state (as a function of proteome fractions) and coarse-grained times allow us to determine the optimal RF1 and RF2 solutions (within our model). The optimality condition (*Equation 5*) is now:

$$
\left(\frac{\partial\tau_{pep}}{\partial\phi_{RF1}}\right)^* = -\frac{\langle\ell\rangle}{\ell_{ribo}\lambda^*}, \quad \left(\frac{\partial\tau_{pep}}{\partial\phi_{RF2}}\right)^* = -\frac{\langle\ell\rangle}{\ell_{ribo}\lambda^*}.
$$

Solving the above system leads to optima $\phi_{RF1}^*$ and $\phi_{RF2}^*$:

$$
\phi_{RF1}^* + \phi_{RF2}^* = \sqrt{\frac{\ell_{ribo}\lambda^*(1+\delta)}{\langle\ell\rangle k_{on}^{RFI}}} + \frac{\ell_{RFI}\lambda^*}{\langle\ell\rangle k_{cat}^{RFI}}, \tag{23}
$$

$$
\frac{\phi_{RF1}^* - \dfrac{f_{UAG}\ell_{RFI}\lambda^*}{\langle\ell\rangle k_{cat}^{RFI}}}{\phi_{RF2}^* - \dfrac{f_{UGA}\ell_{RFI}\lambda^*}{\langle\ell\rangle k_{cat}^{RFI}}} = \sqrt{\frac{f_{UAG}}{f_{UGA}}}. \tag{24}
$$

where the new factor $\delta := 2\sqrt{f_{UAG}f_{UGA}}$.

The relative flux through each stop codon ($f_{UAA}, f_{UAG}, f_{UGA}$) can be estimated in a variety of bacteria from ribosome profiling data (*Lalanne et al., 2018*) as the total synthesis fraction of genes with the respective stop codon. For fast growing species considered in the current study, $f_{UAA} \approx 0.9$, and the correction term to the optimal solution for the summed abundance of RF1 and RF2 ($\sqrt{1+\delta}$) is consequently small (*E. coli*: $f_{UAA} = 0.888$, $f_{UAG} = 0.015$, $f_{UGA} = 0.097$, $\sqrt{1+\delta} = 1.04$; *B. subtilis*: $f_{UAA} = 0.888$, $f_{UAG} = 0.064$, $f_{UGA} = 0.049$, $\sqrt{1+\delta} = 1.05$; *V. natriegens*: $f_{UAA} = 0.929$, $f_{UAG} = 0.041$, $f_{UGA} = 0.031$, $\sqrt{1+\delta} = 1.04$)

# Appendix 3

## Translation elongation
### Coarse-grained one-codon model

Translation elongation is a more complicated process than termination, involving multiple factors to bring the charged tRNA to the ribosome (EF-Tu), charge the tRNAs (aaRS), translocate the ribosome (EF-G), and perform nucleotide exchange on EF-Tu to drive the process (EF-Ts), in addition to others not included here. Our simplified kinetic scheme is illustrated in *Appendix 3—figure 1*. In anticipation coarse-graining procedure detailed below, rates rescaled in the conversion to a one-codon model are marked by *.

To simplify our model, we coarse-grain the elongation cycle by considering a single codon type (section Estimation of coarse-grained rates below or details of the coarse-graining procedure), effectively grouping the tRNA's, tRNA synthetases, and different ternary complexes to single entities. Importantly, as a result, the on-rates associated with these processes are rescaled by a factor close to $n_{aa}^{-1}$, where $n_{aa} = 20$.

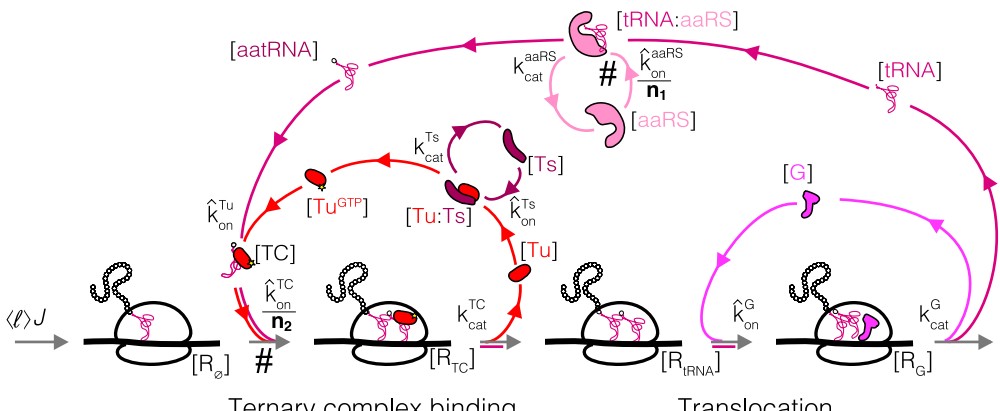

**Appendix 3—figure 1.** Coarse-grained reaction scheme for a single step (amino acid incorporation) of translation elongation. Tu: EF-Tu, Ts: EF-Ts, G: EF-G, aaRS: aminoacyl tRNA synthetases. Steps with slower rates as a result of the coarse-graining to one effective codon are marked by #.

An important distinction for elongation compared to initiation and termination is that multiple elongation steps (average $\langle \ell \rangle \approx 200$) are required to generate a protein. Hence, the flux into the through the elongation cycle is $\langle \ell \rangle$ larger than that through the initiation and termination steps (there is one initiation and termination event for each protein made, but about 200 elongation steps on average).

The mass action reaction scheme for translation elongation:

$$
\begin{aligned}
&\xrightarrow{\langle \ell \rangle J} R_\emptyset, \\
\mathrm{tRNA} + \mathrm{aaRS} &\xrightarrow{\hat{k}_{on}^{aaRS}/n_1} \mathrm{tRNAaaRS}, \\
\mathrm{tRNAaaRS} &\xrightarrow{k_{cat}^{aaRS}} \mathrm{aatRNA} + \mathrm{aaRS} \\
\mathrm{Tu} + \mathrm{Ts} &\xrightarrow{\hat{k}_{on}^{Ts}} \mathrm{TuTs}, \\
\mathrm{TuTs} &\xrightarrow{k_{cat}^{Ts}} \mathrm{Tu}^{GTP} + \mathrm{Ts}, \\
\mathrm{Tu}^{GTP} + \mathrm{aatRNA} &\xrightarrow{\hat{k}_{on}^{Tu}} \mathrm{TC}, \\
\mathrm{TC} + R_\emptyset &\xrightarrow{\hat{k}_{on}^{TC}/n_2} R_{TC}, \\
R_{TC} &\xrightarrow{k_{cat}^{TC}} R_{tRNA}, \\
R_{tRNA} + \mathrm{G} &\xrightarrow{\hat{k}_{on}^{G}} R_G, \\
R_G &\xrightarrow{k_{cat}^{G}} \mathrm{G} + \mathrm{tRNA}.
\end{aligned}
$$

(25)

To arrive at the above, we started with a full model of translation (not shown), will all possible codons, tRNA species, and ribosomes with different codons. To coarse-grain the model, we introduced the following effective variables, which correspond to the total concentration of each type of species involved, summed over the of the codon/amino acid specificity:

$$[\text{tRNA}] := \sum_i [\text{tRNA}_i], \quad [\text{aatRNA}] := \sum_i [\text{aatRNA}_i], \quad [\text{aaRS}] := \sum_{\hat{i}} [\text{aaRS}_i], \quad [\text{TC}] := \sum_i [\text{TC}_i]$$

$$[\text{R}_\theta] := \sum_{i,\nu,\mu} [\text{R}^i_{\nu\mu}], \quad [\text{R}_{TC}] := \sum_{i,j,\nu,\mu} [\text{R}^{i\,TC_j}_{\nu\mu}], \quad [\text{R}_{tRNA}] := \sum_{i,j,\nu,\mu} [\text{R}^{ij}_{\nu\mu}], \quad [\text{R}_G] := \sum_{i,j,\nu,\mu} [\text{R}^{ij}_{\nu\mu} :: G].$$

In the above, Greek indices correspond to different codons on mRNAs, and Roman indices to different tRNAs. Roman indices with a hat ($\hat{i}$) correspond to tRNA synthetases recognizing specific tRNAs (multiple amino acids have more than one tRNA isoacceptor). In defining these coarse-grained species (our approach is analogous to that of *Dai et al., 2016*), we redefined the two following kinetic parameters:

$$\frac{\hat{k}^{aaRS}_{on}}{n_1} := \hat{k}^{aaRS}_{on} \sum_i \frac{[\text{tRNA}_i][\text{aaRS}_i]}{[\text{tRNA}][\text{aaRS}]}, \quad \text{and} \quad \frac{\hat{k}^{TC}_{on}}{n_2} := \hat{k}^{TC}_{on} \sum_{\mu,\nu,i,j} \frac{[\text{R}^i_{\mu\nu}]S_{\nu,j}[\text{TC}_j]}{[\text{R}_\emptyset][\text{TC}]}. \tag{26}$$

$\hat{k}^{aaRS}_{on}$ and $\hat{k}^{TC}_{on}$ correspond to the microscopic bimolecular rates (assumed equal for the different chemical species). $S_{\nu,j}$ is the tRNA isoacceptor/codon specificity matrix (one if tRNA $i$ can recognize codon $\nu$, 0 otherwise) (*Björk and Hagervall, 2014*). Rescaling terms $n_1$ and $n_2$ are estimated below.

## Estimation of coarse-grained rates

The definition of coarse-grained parameters (*Equations 26*) involves sums:

$$\frac{1}{n_1} := \sum_i \frac{[\text{tRNA}_i][\text{aaRS}_i]}{[\text{tRNA}][\text{aaRS}]} \quad \text{and} \quad \frac{1}{n_2} := \sum_{\mu,\nu,i,j} \frac{[\text{R}^i_{\mu\nu}]S_{\nu,j}[\text{TC}_j]}{[\text{R}_\emptyset][\text{TC}]}.$$

These can be estimated from tRNA abundances, codon usage and individual synthetases' levels obtained from ribosome profiling data in *E. coli* (*Li et al., 2014*).

We first consider $n_1$. Note that the fraction of free tRNA of type $i$ to the total number of free tRNA (not bound to any protein) is not readily measurable. Assuming similarities between types of tRNA's, we approximate this fraction with the fraction of total tRNA of type $i$ to the total tRNA concentration, or

$$\frac{[\text{tRNA}_i]}{[\text{tRNA}]} \approx \frac{\text{tRNA}^{tot}_i}{\text{tRNA}_{tot}}.$$

The total tRNA concentration has been measured at fast growth for *E. coli* (*Dong et al., 1996*). The relative concentration of each tRNA synthetases (appropriately corrected for stoichiometry for the different classes) can be computed from the ribosome profiling data (*Li et al., 2014*), and we obtain

$$\frac{1}{n_1} := \sum_i \left( \frac{[\text{tRNA}_i]}{[\text{tRNA}]} \frac{[\text{aaRS}_i]}{[\text{aaRS}]} \right) \approx \sum_i \left( \frac{\text{tRNA}^{tot}_i}{\text{tRNA}_{tot}} \frac{[\text{aaRS}_i]}{[\text{aaRS}]} \right) \approx 0.056 \quad \Rightarrow n_1 \approx 17.8$$

This was to be expected since the synthetases in *E. coli* show little variability around their mean, and in the case of equal synthetase concentration, $n_1 = 20$ would strictly hold.

For the second sum ($n_2$), we use distribution of ribosome footprint reads across the transcriptome to estimate ribosome occupancies at different codons. We first make the following approximation for one of the sub-sum:

$$\sum_{\mu,i} \frac{[\text{R}^i_{\mu\nu}]}{[\text{R}_\emptyset]} \approx \sum_\mu \frac{N^{FP}_{\mu\nu}}{N^{FP}_{tot}},$$

where $N^{FP}_{\mu\nu}$ is the total number of ribosome footprint reads at codon pairs $\mu, \nu$ and $N^{FP}_{tot}$ is the total number of footprint reads mapping to coding sequences. The nature of the approximation is that

we are taking relative fraction of ribosome footprints (representing ribosomes across the elongation cycle at that codon pair) at a given codon pair to be equal to the relative fraction of ribosomes waiting for the ternary complex to derliver a tRNA to the A site. The modest differences in elongation rates at different codons seen in ribosome profiling data (*Mohammad et al., 2019*) justify this approximation.

From our data (not shown), we have that

$$\sum_\mu \frac{N_{\mu\nu}^{FP}}{N_{tot}^{FP}} \approx \sum_\mu \frac{N_{\nu\mu}^{FP}}{N_{tot}^{FP}} = \frac{N_\nu^{FP}}{N_{tot}^{FP}} := f_\nu$$

holds to better than 0.5% for each codon. $f_\nu$ above is the (expression weighted) codon usage. As before with the free tRNA concentrations, we can approximate the relative ternary complexes concentrations by the corresponding total tRNA concentrations:

$$\frac{1}{n_2} := \sum_{\mu,\nu,i,j} \frac{[R_{\mu\nu}^i]S_{\nu,j}[TC_j]}{[R][TC]} \approx \sum_{\nu,j} \frac{f_\nu S_{\nu,j}\, \text{tRNA}_j^{tot}}{\text{tRNA}_{tot}} \approx 0.048 \quad \Rightarrow n_2 \approx 20.8 \tag{27}$$

We used the same dataset as before for the total tRNA concentration in *E. coli* (*Dong et al., 1996*). The codon usage was determined directly from ribosome profiling data (*Li et al., 2014*). The sum of these products is graphically represented in *Appendix 3—figure 2*. The above sum of product of tRNA fraction and codon usage provides an effective number of different ternary complexes. A priori, that might have been expected to equal to the number of tRNAs ($\approx$40). However, as is apparent in *Appendix 3—figure 2*, certain tRNA-codon pairs are much more prevalent than others (even for amino acid with multiple codons and/or tRNA isoacceptors), which leads to a decrease in the effective concentration. The exact value depends on the detailed codon usage and tRNA abundance.

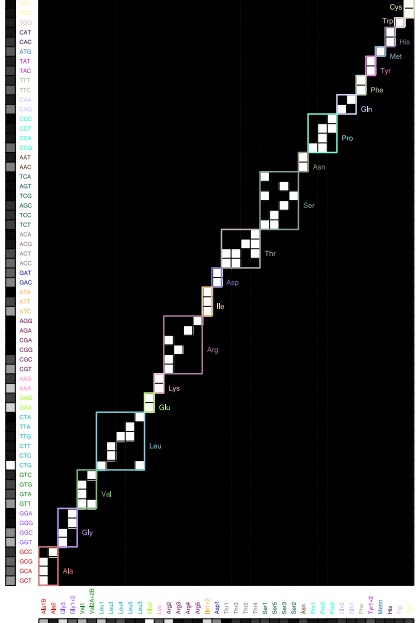 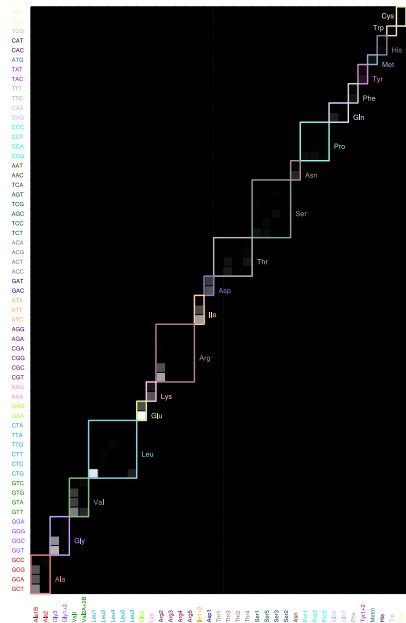

**Appendix 3—figure 2.** Graphical illustration of the sum (*Equation 27*). Left: codon usage (vertical, from analysis of ribosome profiling data from *Li et al., 2014*), tRNA-codon specificity (matrix, from *Björk and Hagervall, 2014*, with different amino acids outlined with different colors), and tRNA abundance (horizontal, from *Dong et al., 1996*) organized by amino acid. Right: product matrix.

Given the results above, we take for simplicity $n_1 = n_2 = n_{aa} = 20$.

## Translation elongation: optimal solutions

The mass action reactions corresponding to the one codon elongation cycle model are (*Equations 25*):

$$
\begin{aligned}
\frac{\mathrm{d}[\mathrm{R}_\emptyset]}{\mathrm{d}t} &= \langle \ell \rangle J - \frac{\hat{k}_{on}^{TC}}{n_{aa}}[\mathrm{TC}][\mathrm{R}_\emptyset], \\
\frac{\mathrm{d}[\mathrm{R}_{TC}]}{\mathrm{d}t} &= \frac{\hat{k}_{on}^{TC}}{n_{aa}}[\mathrm{TC}][\mathrm{R}_\emptyset] - k_{cat}^{TC}[\mathrm{R}_{TC}], \\
\frac{\mathrm{d}[\mathrm{Tu}]}{\mathrm{d}t} &= k_{cat}^{TC}[\mathrm{R}_{TC}] - \hat{k}_{on}^{Ts}[\mathrm{Tu}][\mathrm{Ts}], \\
\frac{\mathrm{d}[\mathrm{tRNA}]}{\mathrm{d}t} &= -\frac{\hat{k}_{on}^{aaRS}}{n_{aa}}[\mathrm{tRNA}][\mathrm{aaRS}] + k_{cat}^{G}[\mathrm{R}_G], \\
\frac{\mathrm{d}[\mathrm{tRNA::aaRS}]}{\mathrm{d}t} &= \frac{\hat{k}_{on}^{aaRS}}{n_{aa}}[\mathrm{tRNA}][\mathrm{aaRS}] - k_{cat}^{aaRS}[\mathrm{tRNA::aaRS}] = -\frac{\mathrm{d}[\mathrm{aaRS}]}{\mathrm{d}t}, \\
\frac{\mathrm{d}[\mathrm{aatRNA}]}{\mathrm{d}t} &= k_{cat}^{aaRS}[\mathrm{tRNA::aaRS}] - \hat{k}_{on}^{Tu}[\mathrm{aatRNA}][\mathrm{Tu}^{\mathrm{GTP}}], \\
\frac{\mathrm{d}[\mathrm{Tu}^{\mathrm{GTP}}]}{\mathrm{d}t} &= k_{cat}^{Ts}[\mathrm{TuTs}] - \hat{k}_{on}^{Tu}[\mathrm{aatRNA}][\mathrm{Tu}^{\mathrm{GTP}}], \\
\frac{\mathrm{d}[\mathrm{TuTs}]}{\mathrm{d}t} &= -k_{cat}^{Ts}[\mathrm{TuTs}] + \hat{k}_{on}^{Ts}[\mathrm{Tu}][\mathrm{Ts}] = -\frac{\mathrm{d}[\mathrm{Ts}]}{\mathrm{d}t}, \\
\frac{\mathrm{d}[\mathrm{TC}]}{\mathrm{d}t} &= \hat{k}_{on}^{Tu}[\mathrm{aatRNA}][\mathrm{Tu}^{\mathrm{GTP}}] - \frac{\hat{k}_{on}^{TC}}{n_{aa}}[\mathrm{TC}][\mathrm{R}_\emptyset], \\
\frac{\mathrm{d}[\mathrm{R}_{tRNA}]}{\mathrm{d}t} &= k_{cat}^{TC}[\mathrm{R}_{TC}] - \hat{k}_{on}^{G}[\mathrm{R}_{tRNA}][\mathrm{G}], \\
\frac{\mathrm{d}[\mathrm{R}_G]}{\mathrm{d}t} &= \hat{k}_{on}^{G}[\mathrm{R}_{tRNA}][\mathrm{G}] - k_{cat}^{G}[\mathrm{R}_G] = -\frac{\mathrm{d}[\mathrm{G}]}{\mathrm{d}t}.
\end{aligned}
$$

Conservation equations close the system:

$$
\begin{aligned}
\mathrm{Ts}_{tot} &= [\mathrm{Ts}] + [\mathrm{TuTs}], \\
\mathrm{Tu}_{tot} &= [\mathrm{Tu}] + [\mathrm{Tu}^{\mathrm{GTP}}] + [\mathrm{TuTs}] + [\mathrm{TC}] + [\mathrm{R}_{TC}], \\
\mathrm{tRNA}_{tot} &= [\mathrm{R}_\emptyset] + 2[\mathrm{R}_{TC}] + 2[\mathrm{R}_{tRNA}] + 2[\mathrm{R}_G] + [\mathrm{tRNA}] + [\mathrm{tRNAaaRS}] + [\mathrm{aatRNA}] + [\mathrm{TC}], \\
\mathrm{aaRS}_{tot} &= [\mathrm{tRNAaaRS}] + [\mathrm{aaRS}], \\
\mathrm{G}_{tot} &= [\mathrm{G}] + [\mathrm{R}_G].
\end{aligned}
$$

The ternary complex concentration and free EF-G concentration enter the translation elongation time (*Equation 10*, which is the diffusion limited and factor dependent contribution to the elongation time) and are required to infer optimal abundances of elongation factors. Both can to be obtained by solving the system of non-linear equations above.

First, catalytic steps must equal to the flux through in the system in steady-state and thus:

$$
[\mathrm{R}_G] = \frac{\langle \ell \rangle J}{k_{cat}^{G}}, \quad [\mathrm{R}_{TC}] = \frac{\langle \ell \rangle J}{k_{cat}^{TC}}, \quad [\mathrm{tRNA::aaRS}] = \frac{\langle \ell \rangle J}{k_{cat}^{aaRS}}, \quad [\mathrm{Tu::Ts}] = \frac{\langle \ell \rangle J}{k_{cat}^{Ts}}.
$$

Together with the conservation equations, these allow for immediate solutions for the free concentrations $[\mathrm{Ts}]$, $[\mathrm{aaRS}]$, and $[\mathrm{G}]$:

$$
\begin{aligned}
[\mathrm{Ts}] &= \mathrm{Ts}_{tot} - \frac{\langle \ell \rangle J}{k_{cat}^{Ts}}, \\
[\mathrm{aaRS}] &= \mathrm{aaRS}_{tot} - \frac{\langle \ell \rangle J}{k_{cat}^{aaRS}}, \\
[\mathrm{G}] &= \mathrm{G}_{tot} - \frac{\langle \ell \rangle J}{k_{cat}^{G}}.
\end{aligned}
$$

The solution for other species can then also be obtained in terms $[\mathrm{Tu}^{\mathrm{GTP}}]$, and $[\mathrm{TC}]$:

$$
[\mathrm{R}_{tRNA}] = \frac{\langle \ell \rangle J}{\hat{k}_{on}^{G}\left(\mathrm{G}_{tot} - \frac{\langle \ell \rangle J}{k_{cat}^{G}}\right)}, \ [\mathrm{R}_{\emptyset}] = \frac{\langle \ell \rangle n_{aa} J}{\hat{k}_{on}^{TC}[\mathrm{TC}]}
$$

$$
[\mathrm{tRNA}] = \frac{\langle \ell \rangle n_{aa} J}{\hat{k}_{on}^{aaRS}\left(\mathrm{aaRS}_{tot} - \frac{\langle \ell \rangle J}{k_{cat}^{aaRS}}\right)}, \ [\mathrm{aatRNA}] = \frac{\langle \ell \rangle J}{\hat{k}_{on}^{Tu}[\mathrm{Tu}^{\mathrm{GTP}}]},
$$

$$
[\mathrm{Tu}] = \frac{\langle \ell \rangle J}{\hat{k}_{on}^{Ts}\left(\mathrm{Ts}_{tot} - \frac{\langle \ell \rangle J}{k_{cat}^{Ts}}\right)}.
$$

Substituting these in the conservation equations for tRNAs and EF-Tu lead to the final system to solve (converting to proteome fraction):

$$
\frac{\mathrm{tRNA}_{tot}}{P} := \psi_{tRNA} = \frac{\lambda n_{aa}}{k_{on}^{TC}\phi_{TC}} + \frac{2\lambda}{k_{cat}^{TC}} + \frac{2\lambda}{k_{on}^{G}\left(\phi_{G} - \frac{\ell_{G}\lambda}{k_{cat}^{G}}\right)} + \frac{2\lambda}{k_{cat}^{G}} + \dots \tag{28}
$$

$$
\frac{\lambda n_{aa}}{k_{on}^{aaRS}\left(\phi_{aaRS} - \frac{\ell_{aaRS}\lambda}{k_{cat}^{aaRS}}\right)} + \frac{\lambda}{k_{cat}^{aaRS}} + \frac{\lambda}{k_{on}^{Tu}\phi_{Tu^{\mathrm{GTP}}}} + \frac{\phi_{TC}}{\ell_{Tu}},
$$

$$
\text{where } \phi_{Tu^{\mathrm{GTP}}} := \phi_{Tu} - \frac{\ell_{Tu}\lambda}{k_{on}^{Ts}\left(\phi_{Ts} - \frac{\ell_{Ts}\lambda}{k_{cat}^{Ts}}\right)} - \frac{\ell_{Tu}\lambda}{k_{cat}^{Ts}} - \phi_{TC} - \frac{\ell_{Tu}\lambda}{k_{cat}^{TC}}. \tag{29}
$$

where the solution for $\phi_{Tu^{\mathrm{GTP}}}$ in terms of the ternary concentration was obtained from the conservation equation for EF-Tu. *Equations 28 and 29* are closed, and the only variables to solve for is $\phi_{TC}$ in terms of the tlF abundances: $\phi_{Tu}, \phi_{Ts}, \phi_{G}, \phi_{aaRS}$, tRNA abundances, kinetic parameters, and the growth rate $\lambda$.

## Coarse-grained translation elongation time

In order to obtain the coarse-grained translation elongation time, we proceed as for translation termination (section Coarse-grained translation termination time). The summed concentration of the ribosome containing species for translation elongation in our model is:

$$
\begin{aligned}
[\mathrm{R}_{el}] &= [\mathrm{R}_{\emptyset}] + [\mathrm{R}_{TC}] + [\mathrm{R}_{tRNA}] + [\mathrm{R}_{G}], \\
&= \frac{\langle \ell \rangle n_{aa} J}{\hat{k}_{on}^{TC}[\mathrm{TC}]} + \frac{\langle \ell \rangle J}{k_{cat}^{TC}} + \frac{\langle \ell \rangle J}{\hat{k}_{on}^{G}\left(\mathrm{G}_{tot} - \frac{\langle \ell \rangle J}{k_{cat}^{G}}\right)} + \frac{\langle \ell \rangle J}{k_{cat}^{G}}.
\end{aligned}
$$

Converting to proteome fraction:

$$
\frac{1}{\ell_{ribo}}\phi_{ribo}^{el} = \lambda \left( \frac{n_{aa}}{k_{on}^{TC}\phi_{TC}} + \frac{1}{k_{cat}^{TC}} + \frac{1}{k_{on}^{G}\left(\phi_{G} - \frac{\ell_{G}\lambda}{k_{cat}^{G}}\right)} + \frac{1}{k_{cat}^{G}} \right).
$$

From the coarse-grained flux relations through the different categories (*Equation 17*), which defines the coarse-grained transition times, we thus have:

$$
\tau_{el} = \langle \ell \rangle \tau_{aa}, \text{ where } \tau_{aa} = \frac{n_{aa}}{k_{on}^{TC}\phi_{TC}} + \frac{1}{k_{cat}^{TC}} + \frac{1}{k_{on}^{G}\left(\phi_{G} - \frac{\ell_{G}\lambda}{k_{cat}^{G}}\right)} + \frac{1}{k_{cat}^{G}}. \tag{30}
$$

Above, $\tau_{aa}$ is the effective time for a single step (by one codon) of translation elongation, and $\tau_{ind}$ corresponds to the summed time of factor independent transitions in each elongation step (not explicitly included in the kinetic scheme).

## Optimality conditions for translation elongation factors

The optimality condition (*Equation 5*) applied to translation elongation factors leads to:

$$
\left(\frac{\partial \tau_{taa}}{\partial \phi_{G}}\right)^{*} = \left(\frac{\partial \tau_{taa}}{\partial \phi_{Tu}}\right)^{*} = \left(\frac{\partial \tau_{taa}}{\partial \phi_{Ts}}\right)^{*} = \left(\frac{\partial \tau_{taa}}{\partial \phi_{aaRS}}\right)^{*} = -\frac{1}{\ell_{ribo}\lambda^{*}}. \tag{31}
$$

where **Equation 30** was used for $\tau_{aa}$. Since the free EF-G concentration does not depend on EF-Tu, EF-Ts, or aaRS concentration, the conditions for EF-Tu, EF-Ts and aaRS simplify to:

$$\frac{\partial}{\partial \phi_{Tu}}\left(\frac{n_{aa}}{k_{on}^{TC}\phi_{TC}}\right)^* = \frac{\partial}{\partial \phi_{Ts}}\left(\frac{n_{aa}}{k_{on}^{TC}\phi_{TC}}\right)^* = \frac{\partial}{\partial \phi_{aaRS}}\left(\frac{n_{aa}}{k_{on}^{TC}\phi_{TC}}\right)^* = -\frac{1}{\ell_{ribo}\lambda^*}. \tag{32}$$

Carrying through the differentiation also leads to conditions on the derivatives of the ternary complex concentration at the optimum:

$$\left(\frac{\partial \phi_{TC}}{\partial \phi_{Tu}}\right)^* = \left(\frac{\partial \phi_{TC}}{\partial \phi_{Ts}}\right)^* = \left(\frac{\partial \phi_{TC}}{\partial \phi_{aaRS}}\right)^* = \frac{k_{on}^{TC}\left(\phi_{TC}^*\right)^2}{\ell_{ribo}n_{aa}\lambda^*}. \tag{33}$$

These relationships will be useful to solve for the some elongation factor optimal abundances below.

## Optimal EF-Ts abundance

Differentiating **Equation 28** with respect to $\phi_{Tu}$ and $\phi_{Ts}$, we get at the optimum:

$$\frac{1}{\ell_{ribo}} + \frac{\lambda^*}{k_{on}^{Tu}\left(\phi_{Tu^{\mathrm{GTP}}}^*\right)^2}\left(\frac{\partial \phi_{Tu^{\mathrm{GTP}}}}{\partial \phi_{Tu}}\right)^* = \frac{1}{\ell_{Tu}}\left(\frac{\partial \phi_{TC}}{\partial \phi_{Tu}}\right)^*,$$

$$\frac{1}{\ell_{ribo}} + \frac{\lambda^*}{k_{on}^{Tu}\left(\phi_{Tu^{\mathrm{GTP}}}^*\right)^2}\left(\frac{\partial \phi_{Tu^{\mathrm{GTP}}}}{\partial \phi_{Ts}}\right)^* = \frac{1}{\ell_{Tu}}\left(\frac{\partial \phi_{TC}}{\partial \phi_{Ts}}\right)^*.$$

By **Equation 33**, the above leads to the additional condition at the optimum:

$$\left(\frac{\partial \phi_{Tu^{\mathrm{GTP}}}}{\partial \phi_{Tu}}\right)^* = \left(\frac{\partial \phi_{Tu^{\mathrm{GTP}}}}{\partial \phi_{Ts}}\right)^*.$$

Directly differentiating **Equation 29**, and using **Equation 33**, leads to:

$$\left(\frac{\partial \phi_{Tu^{\mathrm{GTP}}}}{\partial \phi_{Tu}}\right)^* = 1 - \frac{k_{on}^{TC}\left(\phi_{TC}^*\right)^2}{\ell_{ribo}n_{aa}\lambda^*} = \left(\frac{\partial \phi_{Tu^{\mathrm{GTP}}}}{\partial \phi_{Ts}}\right)^* = \frac{\ell_{Tu}\lambda^*}{k_{on}^{Ts}\left(\phi_{Ts}^* - \frac{\ell_{Ts}\lambda}{k_{cat}^{Ts}}\right)^2} - \frac{k_{on}^{TC}\left(\phi_{TC}^*\right)^2}{\ell_{ribo}n_{aa}\lambda^*}.$$

Therefore, the optimal abundance for EF-Ts is:

$$\phi_{Ts}^* = \sqrt{\frac{\ell_{Tu}\lambda^*}{k_{on}^{Ts}}} + \frac{\ell_{Ts}\lambda^*}{k_{cat}^{Ts}}. \tag{34}$$

## Optimal EF-G abundance

The optimality condition for EF-G is complicated by the fact that EF-G free concentration appears in the solution for the steady-state ternary complex through the tRNA conservation **Equation 28**. Differentiating the conservation tRNA equation, and using the optimality condition 31 (replacing a number of terms with the elongation time $\tau_{aa}$, **Equation 30**):

$$0 = -\frac{2}{\ell_{ribo}} + \frac{\lambda^* n_{aa}}{k_{on}^{TC}\left(\phi_{Tu}^*\right)^2}\left(\frac{\partial \phi_{TC}}{\partial \phi_{G}}\right)^* + \frac{1}{\ell_{Tu}}\left(\frac{\partial \phi_{TC}}{\partial \phi_{G}}\right)^* - \frac{\lambda^*}{k_{on}^{Tu}\left(\phi_{Tu^{\mathrm{GTP}}}^*\right)^2}\left(\frac{\partial \phi_{Tu^{\mathrm{GTP}}}}{\partial \phi_{G}}\right)^*. \tag{35}$$

Above, the right-hand portion corresponds to the additional constraint coming from the implication of EF-G in the steady-state concentration of the ternary complex. From the equation for $\phi_{Tu^{\mathrm{GTP}}}$ (**Equation 29**), we have directly:

$$\left(\frac{\partial \phi_{Tu^{\mathrm{GTP}}}}{\partial \phi_{G}}\right)^* = -\left(\frac{\partial \phi_{TC}}{\partial \phi_{G}}\right)^*.$$

Substituting this in **Equation 35**:

$$\frac{2}{\ell_{ribo}} = \left[ \frac{1}{\ell_{Tu}} + \frac{\lambda^*}{k_{on}^{Tu}(\phi_{Tu^{\mathrm{GTP}}}^*)^2} + \frac{\lambda^* n_{aa}}{k_{on}^{TC}(\phi_{TC}^*)^2} \right] \left( \frac{\partial \phi_{TC}}{\partial \phi_G} \right)^*. \tag{36}$$

The derivative of the ternary complex with respect to EF-G at the optimum can be obtained from the original optimality condition 31, by carrying through the differentiation:

$$\left( \frac{\partial \phi_{TC}}{\partial \phi_G} \right)^* = \frac{k_{on}^{TC}}{n_{aa}} (\phi_{TC}^*)^2 \left[ \frac{1}{\ell_{ribo}\lambda^*} - \frac{1}{k_{on}^G \left( \phi_G^* - \frac{\ell_G \lambda^*}{k_{cat}^G} \right)^2} \right].$$

Substituting in *Equation 36*, we arrive at a final equation for EF-G in terms of the concentration of other elongation factor and the optimal growth rate:

$$\frac{2}{\ell_{ribo}} = \lambda^* \left[ 1 + \frac{k_{on}^{TC}(\phi_{TC}^*)^2}{n_{aa}\ell_{Tu}\lambda^*} + \frac{k_{on}^{TC}(\phi_{TC}^*)^2}{n_{aa}k_{on}^{Tu}(\phi_{Tu^{\mathrm{GTP}}}^*)^2} \right] \left( \frac{1}{\ell_{ribo}\lambda^*} - \frac{1}{k_{on}^G \left( \phi_G^* - \frac{\ell_G \lambda^*}{k_{cat}^G} \right)^2} \right).$$

The optimal solution for EF-G is thus:

$$\phi_G^* = \sqrt{\frac{\ell_{ribo}\lambda^*}{k_{on}^G} \left( \frac{\Delta + 1}{\Delta - 1} \right)} + \frac{\ell_G \lambda^*}{k_{cat}^G} \geq \sqrt{\frac{\ell_{ribo}\lambda^*}{k_{on}^G}} + \frac{\ell_G \lambda^*}{k_{cat}^G},$$

$$\text{where: } \Delta := \frac{k_{on}^{TC}(\phi_{TC}^*)^2}{n_{aa}\ell_{Tu}\lambda^*} + \frac{k_{on}^{TC}(\phi_{TC}^*)^2}{n_{aa}k_{on}^{Tu}(\phi_{Tu^{\mathrm{GTP}}}^*)^2}. \tag{37}$$

Note that given that the term $\Delta$ involves $\phi_{TC}^*$ and $\phi_{Tu^{\mathrm{GTP}}}^*$, and so the solution above is not a priori complete. However, using the approximate ternary complex concentration at the optimum (*Equation 12*, derived in details in section Optimal EF-Tu and aaRS abundances), we have:

$$\Delta > \frac{k_{on}^{TC}(\phi_{TC}^*)^2}{n_{aa}\ell_{Tu}\lambda^*} \approx \frac{\ell_{ribo}}{\ell_{Tu}} \approx 18.5 \gg 1$$

This means that the lower bound for $\phi_G^*$ above (*Equation 37*) is a good approximation: in the physiological regime, we can approximately neglect the indirect dependence of the ternary complex concentration on EF-G via the tRNA conservation equation. Hence, the approximate solution for the EF-G optimal abundance is (same for had we initially assumed that $\phi_{TC}$ was independent of $\phi_G$, in which case the solution for EF-G can be obtained identically as that of release factors):

$$\phi_G^* \approx \sqrt{\frac{\ell_{ribo}\lambda^*}{k_{on}^G}} + \frac{\ell_G \lambda^*}{k_{cat}^G}.$$

## Optimal EF-Tu and aaRS abundances

While simplifying relations were possible with EF-Ts and EF-G, allowing their solution (approximately) independently from the rest of the cycle, EF-Tu and aaRS are intricately connected through the tRNA cycle. We thus return to the tRNA conservation equation, *Equation 28*. For notational simplicity, we group the catalytic step of the TC, EF-G binding, and EF-G catalytic action (translocation) in parameter $k_{el}^{max}$ (these do not depend on $\phi_{Tu}$ and $\phi_{aaRS}$) which we take to the be experimentally determined value of 22 s$^{-1}$ (*Dai et al., 2016*). Further dropping the EF-Ts related and catalytic terms (will be added back at the end, they only contribute a fixed term at the optimum) in the equation for the free EF-Tu, we get:

$$\frac{\text{tRNA}_{tot}}{P\lambda} = \frac{n_{aa}}{k_{on}^{TC}\phi_{TC}} + \frac{2}{k_{el}^{max}} + \cdots$$

$$\frac{n_{aa}}{k_{on}^{aaRS}\left(\phi_{aaRS} - \frac{\ell_{aaRS}\lambda}{k_{cat}^{aaRS}}\right)} + \frac{1}{k_{cat}^{aaRS}} + \frac{1}{k_{on}^{Tu}\phi_{Tu^{\text{GTP}}}} + \frac{\phi_{TC}}{\ell_{Tu}\lambda}, \tag{38}$$

where $\phi_{Tu^{\text{GTP}}} = \phi_{Tu} - \phi_{TC}$ is the free EF-Tu concentration.

This system is first solved numerically (**Figure 3B**). To close the equation in terms of uniquely $\phi_{TC}$, we use our relationship for $\lambda$ (**Equation 1**), with:

$$\tau_{trl} = \langle \ell \rangle \left( \frac{n_{aa}}{k_{on}^{TC}\phi_{TC}} + \frac{1}{k_{el}^{max}} \right) + \tau_{ini} + \tau_{ter},$$

where as before $k_{el}^{max}$ is the maximum rate of translation elongation (from reactions other than ternary complex diffusion) estimated from in vivo kinetic measurements ($\approx 22$ s$^{-1}$[**Dai et al., 2016**]), and $\tau_{ini} + \tau_{ter} \approx 0.5$ s the estimated time for the initiation and termination step ($\approx 5 - 10\%$ of the full translation cycle translation time), taken as fixed parameters here. Using this relationship for the translation time leads to the explicit relationship between growth and ternary complex concentration:

$$\lambda(\phi_{TC}) = \frac{\phi_{ribo}}{\ell_{ribo}}\left(\frac{k_{trl}\phi_{TC}}{\phi_{TC} + K_{TC}}\right), \text{ with } k_{trl} := \frac{\langle \ell \rangle k_{el}^{max}}{\langle \ell \rangle + k_{el}^{max}(\tau_{ini} + \tau_{ter})} \text{ and } K_{TC} := \frac{k_{trl}n_{aa}}{k_{on}^{TC}} \tag{39}$$

which is the same relationship as the one derived in **Klumpp et al., 2013**, with the addition of the terms corresponding to the rest translation cycle. Substituting the explicit relationship between growth and ternary complex concentration above (**Equation 39**) in the aaRS/EF-Tu tRNA cycle relationship (**Equation 38**) closes the system for $\phi_{TC}$. Numerical solution for this equation is presented in **Figure 3B** (see section Estimation of optimal abundances for other parameters).

The main conclusion from numerically solving the reduced system (**Equations 38 and 39)** is that the EF-Tu/aaRS space is partitioned in two regimes, resulting from the separation of scale of reactions in the coarse-grained model. Specifically, $k_{on}^{Tu} \gg \frac{k_{on}^{TC}}{n_{aa}}$, so that any imbalance between the constituents of the ternary complex (charged tRNAs, free EF-Tu), results in stoichiometric unproductive excess of the component in surplus.

We can derive a relation for the "transition line' in the aaRS/EF-Tu space where both free charged tRNAs and free EF-Tu are at low concentrations. This corresponds to setting the (formally impossible) requirement $\phi_{Tu^{\text{GTP}}} \approx 0 \Rightarrow \phi_{TC} \approx \phi_{Tu}$ and $[\text{aatRNA}] \propto \frac{1}{k_{on}^{Tu}\phi_{Tu^{\text{GTP}}}} \approx 0$, that is,

$$\frac{\text{tRNA}_{tot}}{P\lambda(\bar\phi_{Tu})} - \frac{n_{aa}}{k_{on}^{TC}\bar\phi_{Tu}} - \frac{2}{k_{el}^{max}} - \frac{\bar\phi_{Tu}}{\ell_{Tu}\lambda(\bar\phi_{Tu})} = \frac{n_{aa}}{k_{on}^{aaRS}\left(\bar\phi_{aaRS} - \frac{\ell_{aaRS}\lambda(\bar\phi_{Tu})}{k_{cat}^{aaRS}}\right)} + \frac{1}{k_{cat}^{aaRS}}. \tag{40}$$

The $\bar{\cdot}$ signifies the transition line relationship between $\bar\phi_{Tu}$ and $\bar\phi_{aaRS}$, which is displayed in **Figure 3B**.

The heuristic to estimate the optimal EF-Tu concentration described in the main text can be extended to include the EF-Ts cycle. In particular, in the EF-Tu limited regime, with $\phi_{Tu^{\text{GTP}}} \approx 0$, we have (from **Equation 29**):

$$\phi_{TC} \approx \phi_{Tu} - \frac{\ell_{Tu}\lambda}{k_{on}^{Ts}\left(\phi_{Ts} - \frac{\ell_{Ts}\lambda}{k_{cat}^{Ts}}\right)} - \frac{\ell_{Tu}\lambda}{k_{cat}^{Ts}} - \frac{\ell_{Tu}\lambda}{k_{cat}^{TC}}.$$

Substituting the above expression for $\phi_{TC}$ in the optimality condition (**Equation 32**) for $\phi_{Tu}$, we arrive at (using the optimal solution for EF-Ts, **Equation 34**):

$$\phi_{Tu}^* \approx \sqrt{\frac{\ell_{ribo}n_{aa}\lambda^*}{k_{on}^{TC}}} + \sqrt{\frac{\ell_{Tu}\lambda^*}{k_{on}^{Ts}}} + \frac{\ell_{Tu}\lambda^*}{k_{cat}^{Ts}} + \frac{\ell_{Tu}\lambda^*}{k_{cat}^{TC}}.$$

Above, the last three terms (not appearing in **Equation 12**) correspond to the additional diffusion of the EF-Ts cycle, and catalytic contributions.

Following the argument (see main text) that the optimal aaRS abundance should lie on the transition line (*Equation 40*), we obtain:

$$\phi^*_{aaRS} \approx \frac{n_{aa}\lambda^*}{k^{aaRS}_{on}\Delta^*_{tRNA}} + \frac{\ell_{aaRS}\lambda^*}{k^{aaRS}_{cat}},$$

with $\Delta_t$ related to the excess tRNA (tRNAs remaining after subtracting tRNAs sequestered on the ribosome and TC from the total tRNA budget):

$$\Delta^*_{tRNA} := \frac{\text{tRNA}_{tot}}{P} - \frac{n_{aa}\lambda^*}{k^{TC}_{on}\phi^*_{TC}} - \frac{2\lambda^*}{k^{max}_{el}} - \frac{\phi^*_{TC}}{\ell_{Tu}} - \frac{\lambda^*}{k^{aaRS}_{cat}}, \text{ where } \phi^*_{TC} = \sqrt{\frac{n_{aa}\ell_{ribo}\lambda^*}{k^{TC}_{on}}}.$$

## Interpretation of the sharp separation between aaRS and EF-Tu limited regimes

The sharp separation of the solution for $\phi_{TC}$ in two distinct regimes (EF-Tu limited, and aaRS limited, illustrated in *Figure 3B*), can be intuitively understood from a geometrical viewpoint.

For the simplicity of the argument (not strictly necessary), neglecting the short initiation and termination times in *Equation 39*, and using $\text{tRNA}_{tot} = \frac{t\phi_{ribo}P}{\ell_{ribo}}$ (with $t$ the tRNA to ribosome molar ratio). The tRNA conservation condition, *Equation 38*, can then be rewritten as (binding-limited regime):

$$\underbrace{(t-1)\frac{\phi_{ribo}}{\ell_{ribo}}}_{\text{tRNA budget}} - \underbrace{\frac{\phi_{TC}}{\ell_{Tu}}}_{\text{ternary complex}} - \underbrace{\frac{\lambda(\phi_{TC})}{k^{max}_{el}}}_{\text{A-site tRNA}} = \lambda(\phi_{TC})[\underbrace{\frac{n_{aa}}{k^{aaRS}_{on}\phi_{aaRS}}}_{\text{uncharged tRNA}} + \underbrace{\frac{1}{k^{Tu}_{on}(\phi_{Tu}-\phi_{TC})}}_{\text{free charged tRNA}}]$$

At given abundance of EF-Tu ($\phi_{Tu}$) and aaRS ($\phi_{aaRS}$), the solution for $\phi_{TC}$ is obtained when equality in the above equation is reached. The behavior of the various terms with $\phi_{TC}$ is illustrated for different values of $\phi_{aaRS}$ and $\phi_{Tu}$ in *Figure 3—figure supplement 1*: the number of uncharged tRNAs (pink line in *Figure 3—figure supplement 1*) is a decreasing function of aaRS, and free charged tRNA (red line in *Figure 3—figure supplement 1*) are dependent on $\phi_{Tu}$. Specifically, the free charged tRNA contribution, due to the rapid association rate $k^{Tu}_{on}$ (codon agnostic) between charged tRNAs and EF-Tu (red line), is negligible except for a very narrow range where $\phi_{TC} \approx \phi_{Tu}$, at which point a sharp divergence occurs. This rapid divergence bounds the solution for $\phi_{TC}$ at the total EF-Tu concentration.

The aaRS limited regime corresponds to conditions in which the uncharged tRNA contribution (pink line) intersects the available tRNA budget (full black line), lower left in *Figure 3—figure supplement 1*. In contrast, the EF-Tu limited regime corresponds to conditions in which the free charged tRNA (red line) intersects the tRNA budget, upper right in *Figure 3—figure supplement 1*. The sharpness of the transition between the two regime arises from the near vertical divergence of the free charged tRNA contribution.

## Appendix 4

### Translation initiation

Translation initiation is also relatively complex compared to translation termination. In contrast with other steps of the translation cycle, binding of factors necessary for the process (IF1, IF2, IF3, initiator tRNA) do not occur in a strict sequential order, leading to a 'heterogeneous assembly landscape' (*Gualerzi and Pon, 2015*; *Chen et al., 2016*) more complex to model. However, one assembly pathway is kinetically favored (*Milón et al., 2012*). We take this favored assembly pathway as our kinetic scheme (*Appendix 4—figure 1*, note that binding of tRNA/mRNA are coarse-grained to a single even without loss of generality). We provide some evidence below that taking a more complex assembly pathway would minimally affect the predicted optimal initiation factor abundances.

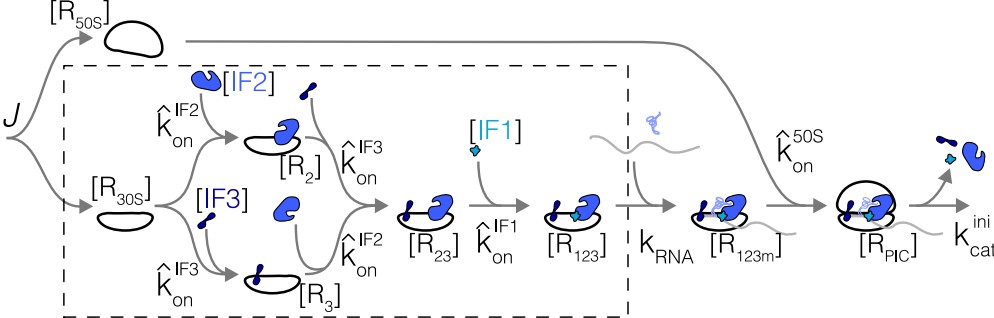

**Appendix 4—figure 1.** Simplified kinetic scheme for translation initiation. Reactions in dashed box correspond to sub-system solved in detail first (section Sub-pathway without subunits joining). Variables are labeled on the scheme.

The reactions in our simplified schemes are:

$$
\begin{aligned}
&\xrightarrow{J} R_{30S} + R_{50S}, \\
R_{30S} + IF3 &\xrightarrow{\hat{k}_{on}^{IF3}} R_3, \\
R_{30S} + IF2 &\xrightarrow{\hat{k}_{on}^{IF2}} R_2, \\
R_3 + IF2 &\xrightarrow{\hat{k}_{on}^{IF2}} R_{23}, \\
R_2 + IF3 &\xrightarrow{\hat{k}_{on}^{IF3}} R_{23}, \\
R_{23} + IF1 &\xrightarrow{\hat{k}_{on}^{IF1}} R_{123}, \\
R_{123} &\xrightarrow{k_{RNA}} R_{123m}, \\
R_{123m} + R_{50S} &\xrightarrow{\hat{k}_{on}^{50S}} R_{PIC}, \\
R_{PIC} &\xrightarrow{k_{cat}^{ini}} IF1 + IF2 + IF3,
\end{aligned}
$$

with corresponding mass action equations:

$$\frac{d[R_{30S}]}{dt} = J - \hat{k}_{on}^{IF2}[R_{30S}][IF2] - \hat{k}_{on}^{IF3}[R_{30S}][IF3],$$

$$\frac{d[R_2]}{dt} = \hat{k}_{on}^{IF2}[R_{30S}][IF2] - \hat{k}_{on}^{IF3}[R_2][IF3],$$

$$\frac{d[R_3]}{dt} = \hat{k}_{on}^{IF3}[R_{30S}][IF3] - \hat{k}_{on}^{IF2}[R_3][IF2],$$

$$\frac{d[R_{23}]}{dt} = \hat{k}_{on}^{IF2}[R_3][IF2] + \hat{k}_{on}^{IF3}[R_2][IF3] - \hat{k}_{on}^{IF1}[R_{23}][IF1],$$

$$\frac{d[R_{123}]}{dt} = \hat{k}_{on}^{IF1}[R_{23}][IF1] - k_{RNA}[R_{123}],$$

$$\frac{d[R_{123m}]}{dt} = k_{RNA}[R_{123}] - \hat{k}_{on}^{50S}[R_{123m}][R_{50S}],$$

$$\frac{d[R_{PIC}]}{dt} = \hat{k}_{on}^{50S}[R_{123m}][R_{50S}] - k_{cat}^{ini}[R_{PIC}],$$

$$\frac{d[R_{50S}]}{dt} = J - \hat{k}_{on}^{50S}[R_{123m}][R_{50S}],$$

$$\frac{d[IF1]}{dt} = -\hat{k}_{on}^{IF1}[R_{23}][IF1] + k_{cat}^{ini}[PIC],$$

$$\frac{d[IF2]}{dt} = -\hat{k}_{on}^{IF2}([R_{30S}] + [R_3])[IF2] + k_{cat}^{ini}[PIC],$$

$$\frac{d[IF3]}{dt} = -\hat{k}_{on}^{IF3}([R_{30S}] + [R_2])[IF3] + k_{cat}^{ini}[PIC],$$

and conservation equations:

$$IF1_{tot} = [IF1] + [R_{123}] + [R_{123m}] + [R_{PIC}],$$

$$IF2_{tot} = [IF2] + [R_2] + [R_{23}] + [R_{123}] + [R_{123m}] + [R_{PIC}],$$

$$IF3_{tot} = [IF3] + [R_3] + [R_{23}] + [R_{123}] + [R_{123m}] + [R_{PIC}],$$

$$[R_{50S}] = [R_{30S}] + [R_2] + [R_3] + [R_{23}] + [R_{123}] + [R_{123m}].$$

We assume the steady-state concentrations of small and large ribosomal subunits to be equal.

## Sub-pathway without subunits joining

The system of equation is complicated by the second branch of the pathway corresponding to 50S subunit binding. However, in the regime $\sqrt{\frac{\ell_{IF}}{\ell_{ribo}}\frac{\hat{k}_{on}^{50S}}{\hat{k}_{on}^{IF}}} \ll 1$ (which is realized because of the large size of the ribosome and slower association rate constant for the large subunit compared to the initiation factors again due to size), the effect of this branch is to add a term to the optimal abundance equal to the concentration of species $R_{123m}$ (see derivation in section Pathway including subunits joining). We focus here on the solution of the part of the reaction scheme boxed in *Appendix 4—figure 1*. This sub-scheme corresponds to:

$$\xrightarrow{J} R_{30S},$$
$$R_{30S} + IF3 \xrightarrow{\hat{k}_{on}^{IF3}} R_3,$$
$$R_{30S} + IF2 \xrightarrow{\hat{k}_{on}^{IF2}} R_2,$$
$$R_3 + IF2 \xrightarrow{\hat{k}_{on}^{IF2}} R_{23},$$
$$R_2 + IF3 \xrightarrow{\hat{k}_{on}^{IF3}} R_{23},$$
$$R_{23} + IF1 \xrightarrow{\hat{k}_{on}^{IF1}} R_{123},$$
$$R_{123} \xrightarrow{k_{RNA}} R_{123m}.$$

$$\frac{\mathrm{d}[R_{30S}]}{\mathrm{dt}} = J - \hat{k}_{on}^{IF2}[R_{30S}][IF2] - \hat{k}_{on}^{IF3}[R_{30S}][IF3],$$

$$\frac{\mathrm{d}[R_2]}{\mathrm{dt}} = \hat{k}_{on}^{IF2}[R_{30S}][IF2] - \hat{k}_{on}^{IF3}[R_2][IF3],$$

$$\frac{\mathrm{d}[R_3]}{\mathrm{dt}} = \hat{k}_{on}^{IF3}[R_{30S}][IF3] - \hat{k}_{on}^{IF2}[R_3][IF2],$$

$$\frac{\mathrm{d}[R_{23}]}{\mathrm{dt}} = \hat{k}_{on}^{IF2}[R_3][IF2] + \hat{k}_{on}^{IF3}[R_2][IF3] - \hat{k}_{on}^{IF1}[R_{23}][IF1],$$

$$\frac{\mathrm{d}[R_{123}]}{\mathrm{dt}} = \hat{k}_{on}^{IF1}[R_{23}][IF1] - k_{RNA}[R_{123}],$$

$$\frac{\mathrm{d}[IF1]}{\mathrm{dt}} = -\hat{k}_{on}^{1}[R_{23}][IF1] + k_{RNA}[R_{123}],$$

$$\frac{\mathrm{d}[IF2]}{\mathrm{dt}} = -\hat{k}_{on}^{IF2}([R_{30S}] + [R_3])[IF2] + k_{RNA}[R_{123}],$$

$$\frac{\mathrm{d}[IF3]}{\mathrm{dt}} = -\hat{k}_{on}^{IF3}([R_{30S}] + [R_2])[IF3] + k_{RNA}[R_{123}],$$

with conservation equations:

$$IF1_{tot} = [IF1] + [R_{123}],$$
$$IF2_{tot} = [IF2] + [R_2] + [R_{23}] + [R_{123}],$$
$$IF3_{tot} = [IF3] + [R_3] + [R_{23}] + [R_{123}],$$

This system can be solved as with the previous schemes. In steady-state, we find for concentrations in terms of the free concentrations $[IF2]$ and $[IF3]$:

$$[R_{123}] = \frac{J}{k_{RNA}}, \quad [IF1] = IF1_{tot} - \frac{J}{k_{RNA}}, \quad [R_{23}] = \frac{J}{\hat{k}_{on}^{IF1}[IF1]}, \quad [R_{30S}] = \frac{J}{\hat{k}_{on}^{IF2}[IF2] + \hat{k}_{on}^{IF3}[IF3]},$$

$$[R_2] = \frac{\hat{k}_{on}^{IF2}[IF2]}{\hat{k}_{on}^{IF3}[IF3]}\left(\frac{J}{\hat{k}_{on}^{IF2}[IF2] + \hat{k}_{on}^{IF3}[IF3]}\right), \quad [R_3] = \frac{\hat{k}_{on}^{IF3}[IF3]}{\hat{k}_{on}^{IF2}[IF2]}\left(\frac{J}{\hat{k}_{on}^{IF2}[IF2] + \hat{k}_{on}^{IF3}[IF3]}\right),$$

and the coupled equations for $[IF2]$ and $[IF3]$ that need to be solved:

$$IF2_{tot} = [IF2] + \frac{\hat{k}_{on}^{IF2}[IF2]}{\hat{k}_{on}^{IF3}[IF3]}\left(\frac{J}{\hat{k}_{on}^{IF2}[IF2] + \hat{k}_{on}^{IF3}[IF3]}\right) + \frac{J}{\hat{k}_{on}^{IF1}[IF1]} + \frac{J}{k_{RNA}},$$

$$IF3_{tot} = [IF3] + \frac{\hat{k}_{on}^{IF3}[IF3]}{\hat{k}_{on}^{IF2}[IF2]}\left(\frac{J}{\hat{k}_{on}^{IF2}[IF2] + \hat{k}_{on}^{IF3}[IF3]}\right) + \frac{J}{\hat{k}_{on}^{IF1}[IF1]} + \frac{J}{k_{RNA}}. \tag{41}$$

As for translation termination (section Coarse-grained translation termination time) and elongation (section Coarse-grained translation elongation time), summing the ribosome containing species:

$$[R_{ini}] = [R_{30S}] + [R_2] + [R_3] + [R_{23}] + [R_{123}],$$

$$= J\left(\frac{1}{\hat{k}_{on}^{IF2}[IF2]} + \frac{1}{\hat{k}_{on}^{IF3}[IF3]} - \frac{1}{\hat{k}_{on}^{IF2}[IF2] + \hat{k}_{on}^{IF3}[IF3]} + \frac{1}{\hat{k}_{on}^{IF1}[IF1]} + \frac{1}{k_{RNA}}\right),$$

allows us to read the initiation time directly (recast in proteome fraction units):

$$\tau_{ini} = \frac{1}{k_{on}^{IF2}\phi_{IF2}^{free}} + \frac{1}{k_{on}^{IF3}\phi_{IF3}^{free}} - \frac{1}{k_{on}^{IF2}\phi_{IF2}^{free} + k_{on}^{IF3}\phi_{IF3}^{free}} + \frac{1}{k_{on}^{IF1}\phi_{IF1}^{free}} + \frac{1}{k_{RNA}}. \tag{42}$$

The above is the time can be used in the optimality condition (*Equation 5*). Note that the parallel nature of the reactions with IF2 and IF3 leads to a reduction compared to a purely sequential pathway (negative term above decreasing the total initiation time, as expected if multiple reactions can occur in parallel).

Given that binding of IF1 occurs last in this scheme, its free concentration takes a simple form ($\phi_{IF1}^{free} = \phi_{IF1} - \frac{\ell_{IF1}\lambda}{\langle \ell \rangle k_{RNA}}$). In contrast, computing the free IF2 and IF3 concentrations requires solving the non-linear coupled system, *Equations 41*. Recasting these in units of proteome fraction:

$$\tilde{\phi}_{IF2} = \phi_{IF2}^{free} + \frac{\lambda \ell_{IF2}}{\langle \ell \rangle k_{on}^{IF3} \phi_{IF3}^{free}} \left( \frac{k_{on}^{IF2} \phi_{IF2}^{free}}{k_{on}^{IF2} \phi_{IF2}^{free} + k_{on}^{IF3} \phi_{IF3}^{free}} \right),$$

$$\tilde{\phi}_{IF3} = \phi_{IF3}^{free} + \frac{\lambda \ell_{IF3}}{\langle \ell \rangle k_{on}^{IF2} \phi_{IF2}^{free}} \left( \frac{k_{on}^{IF3} \phi_{IF3}^{free}}{k_{on}^{IF2} \phi_{IF2}^{free} + k_{on}^{IF3} \phi_{IF3}^{free}} \right),$$

with $\tilde{\phi}_{IF2} := \phi_{IF2} - \frac{\ell_{IF2}\lambda}{\langle \ell \rangle k_{RNA}} - \frac{\ell_{IF2}\lambda}{\langle \ell \rangle k_{on}^{IF1} \phi_{IF1}^{free}}$, and similarly for $\tilde{\phi}_{IF3}$. We show now that the terms coupling the two equations for $\phi_{IF2}^{free}$ and $\phi_{IF2}^{free}$ (bracketed above) are small at the optimum. Indeed, based on results in simpler schemes (self-consistency confirmed below), we expect at the optimum:

$$\phi_{IF2}^{free,*} \sim \sqrt{\frac{\ell_{ribo}\lambda^*}{\langle \ell \rangle k_{on}^{IF2}}} \quad \text{and} \quad \phi_{IF3}^{free,*} \sim \sqrt{\frac{\ell_{ribo}\lambda^*}{\langle \ell \rangle k_{on}^{IF3}}}.$$

Hence, we expect the two terms at the optimum in the coupled equations above to compare as (e.g. in the free IF2 equation):

$$\frac{\phi_{IF2}^{free,*}}{\left( \frac{\lambda^* \ell_{IF2}}{\langle \ell \rangle k_{on}^{IF3} \phi_{IF3}^{free,*}} \right)} \sim \frac{\ell_{ribo}}{\ell_{IF2}} \sqrt{\frac{k_{on}^{IF3}}{k_{on}^{IF2}}} \gg 1,$$

coming from the large size of the ribosome compared to the initiation factors. In addition, the derivative of the coupling terms, which appear in the optimality condition and therefore in identifying the optimal abundances, are all of the form $\frac{\lambda^* \ell_{IF}}{\langle \ell \rangle k_{on}^{IF} (\phi_{IF}^{free})^2}$ compared to the main term. This scales scales as $\ell_{IF} \ell_{ribo}^{-1} \ll 1$ at the self-consistent solution. Hence, neglecting the coupling is justified as an approximate solutions near the optimum, and we obtain for the free concentrations of IFs:

$$\phi_{IF1}^{free} = \phi_{IF1} - \frac{\ell_{IF1}\lambda}{\langle \ell \rangle k_{RNA}},$$

$$\phi_{IF2}^{free} \approx \phi_{IF2} - \frac{\ell_{IF2}\lambda}{\langle \ell \rangle k_{RNA}} - \frac{\ell_{IF2}\lambda}{\langle \ell \rangle k_{on}^{IF1} \phi_{IF1}^{free}},$$

$$\phi_{IF3}^{free} \approx \phi_{IF3} - \frac{\ell_{IF3}\lambda}{\langle \ell \rangle k_{RNA}} - \frac{\ell_{IF3}\lambda}{\langle \ell \rangle k_{on}^{IF1} \phi_{IF1}^{free}}.$$

Substituting these in the expression for the initiation time, *Equation 42*, and using the optimality condition (*Equation 5*, we find that no simple solution exist for the non symmetric case of $k_{on}^{IF2} \neq k_{on}^{IF3}$). Since the on-rates should be similar for IF2 and IF3 (difference in size should only lead to modest difference in on-rates coefficient, by roughly $(\ell_{IF2}/\ell_{IF3})^{1/3} \approx 1.7$ assuming Stokes scaling), the symmetric case is approximately correct. We report the symmetric solution for simplicity. The final optimal solutions for the three factors for the sub-scheme solved here is:

$$\phi_{IF1}^* \approx \sqrt{\frac{\ell_{ribo}\lambda^*}{\langle \ell \rangle k_{on}^{IF1}} \left[ 1 + \frac{\ell_{IF2} + \ell_{IF3}}{\ell_{ribo}} \right]} + \frac{\ell_{IF1}\lambda^*}{\langle \ell \rangle k^{ini}},$$

$$\phi_{IF2}^* \approx \sqrt{\frac{3}{4}} \sqrt{\frac{\ell_{ribo}\lambda^*}{\langle \ell \rangle k_{on}^{IF2}}} + \frac{\ell_{IF2}}{\langle \ell \rangle} \sqrt{\frac{\ell_{ribo}\lambda^*}{\langle \ell \rangle k_{on}^{IF1}}} + \frac{\ell_{IF2}\lambda^*}{\langle \ell \rangle k^{ini}}, \tag{43}$$

$$\phi_{IF3}^* \approx \sqrt{\frac{3}{4}} \sqrt{\frac{\ell_{ribo}\lambda^*}{\langle \ell \rangle k_{on}^{IF3}}} + \frac{\ell_{IF3}}{\langle \ell \rangle} \sqrt{\frac{\ell_{ribo}\lambda^*}{\langle \ell \rangle k_{on}^{IF1}}} + \frac{\ell_{IF3}\lambda^*}{\langle \ell \rangle k^{ini}}.$$

The form of the solution is again similar to that derived for the simpler translation termination scheme (c.f., *Equation 20*), with three differences, each of which has an intuitive interpretation. First, the factor $\left[ 1 + \frac{\ell_{IF2} + \ell_{IF3}}{\ell_{ribo}} \right]$ in the IF1 solution arises as a result of IF1 binding being last in our initiation pathway. Indeed, IF1 concentration also influences free IF2 and IF3 concentration, leading to additional selective pressure to increase its abundance. In effect, the molecular species waiting for IF1 to diffuse to its target is not only the ribosome, but the ribosome with IF2 and IF3 bound, and a total

amino acid weight $\ell_{ribo} \rightarrow \ell_{ribo} + \ell_{IF2} + \ell_{IF3}$. Second, the factor of $\sqrt{3/4} \approx 0.87 < 1$ for IF2 and IF3 (corresponding to the symmetric case), arising from the parallel pathway for IF2 and IF3 rendering the process more efficient. We therefore see that the correction from having multiple reactions in parallel is modest (0.87 vs. 1). The third difference to the simpler case of translation termination are the second terms for IF2 and IF3, corresponding to the additional delay incurred by binding of IF1. These come from the assumed sequential nature of our initiation scheme (**Appendix 4—figure 1**). In such cases, factors binding earlier have to be present at higher abundances to account for their wait times for later binding events. The exact form of this correction term would be different for more complex assembly pathways (but would be captured by average delays from other factor binding).

## Pathway including subunits joining

The solutions above (**Equations 43**) are for the reduced scheme (boxed in **Appendix 4—figure 1**). The full solutions includes the delay arising from 50S subunit binding. Including subunit joining requires the solution of an additional equation for the steady-state concentration of species with all three initiation factors, mRNA and initiator tRNA waiting for subunit joining (species $R_{123m}$ in **Appendix 4—figure 1**, denoted $\phi_{123m}$ in units of proteome fraction). The equation to solve for $\phi_{123m}$ can be obtained from the 50S ribosome subunit conservation equation:

$$\frac{\lambda}{k_{on}^{50S}\phi_{123m}} = \frac{\lambda}{k_{on}^{IF2}\phi_{IF2}^{free}} + \frac{\lambda}{k_{on}^{IF3}\phi_{IF3}^{free}} - \frac{\lambda}{k_{on}^{IF2}\phi_{IF2}^{free} + k_{on}^{IF3}\phi_{IF3}^{free}} + \frac{\lambda}{k_{on}^{IF1}\phi_{IF1}^{free}} + \frac{\lambda}{k_{RNA}} + \frac{\langle\ell\rangle\phi_{123m}}{\ell_{30S}}.$$

$\phi_{123m}$ appears in the equations for the free concentration of the initiation factors (from the conservation equations), and also leads to the appearance of a new term in the expression for the initiation time $\tau_{ini}$ (**Equation 42**) corresponding to this step: $\frac{\langle\ell\rangle\phi_{123m}}{\ell_{30S}\lambda}$.

These two additions, resulting from the parallel branch of 50S joining, can be simplified due to a separation of scales between the various terms. For large initiation factor concentrations, the corresponding mass action terms in the equation for $\phi_{123m}$ negligibly contribute to the solution. In this regime, the new term involving $\phi_{123m}$ in the initiation time $\tau_{ini}$ does not alter the form the optimal abundances of IF1, IF2, and IF3 beyond adding a constant term. Hence, in the regime of high free IF concentration, the optimality condition has the same form as derived in the previous section. We can therefore obtain $\phi_{123m}$ assuming large IF concentration, denoted $\phi_{123m}^{\infty}$:

$$\phi_{123m}^{\infty} = \frac{\ell_{30S}}{\langle\ell\rangle}\left(-\frac{\lambda}{2k_{RNA}} + \sqrt{\frac{1}{4}\left(\frac{\lambda}{k_{RNA}}\right)^2 + \frac{\langle\ell\rangle\lambda}{\ell_{30S}k_{on}^{50S}}}\right)$$

This solution will be self-consistent provided (for all initiation factors):

$$\frac{\lambda^*}{k_{on}^{IF}\phi_{IF}^{free,*}} \ll \frac{\lambda^*}{k_{RNA}} + \frac{\langle\ell\rangle\phi_{123m}^{\infty}}{\ell_{30S}} = \frac{\lambda^*}{2k_{RNA}} + \sqrt{\frac{1}{4}\left(\frac{\lambda^*}{k_{RNA}}\right)^2 + \frac{\langle\ell\rangle\lambda^*}{\ell_{30S}k_{on}^{50S}}},$$

It therefore suffices to show:

$$\frac{\lambda^*}{k_{on}^{IF}\phi_{IF}^{free,*}} \ll \sqrt{\frac{\langle\ell\rangle\lambda^*}{\ell_{30S}k_{on}^{50S}}}.$$

Using our optimality condition on $\phi_{IF}^{free,*}$ (**Equation 43**) assuming no contribution from $\phi_{123m}$ (self-consistency), and converting association rates in units $\mu M^{-1}s^{-1}$, the above condition reduces to:

$$\sqrt{\frac{\ell_{IF}}{\ell_{ribo}}\frac{\hat{k}_{on}^{50S}}{\hat{k}_{on}^{IF}}} \ll 1.$$

The self-consistency condition is met both because initiation factors are smaller than ribosomes $\ell_{IF} \ll \ell_{ribo}$, and because the on-rate for subunit joining is lower than initiation factor binding ($\hat{k}_{on}^{50S} \ll \hat{k}_{on}^{IF}$), given again the size differences. The solution, including the contribution from ribosome subunits joining is then:

$$\phi_{IF1}^* \approx \sqrt{\frac{\ell_{ribo}\lambda^*}{\langle\ell\rangle k_{on}^{IF1}}\left[1+\frac{\ell_{IF2}+\ell_{IF3}}{\ell_{ribo}}\right]} + \frac{\ell_{IF1}}{\ell_{30S}}\phi_{123m}^\infty + \frac{\ell_{IF1}\lambda^*}{\langle\ell\rangle}\left(\frac{1}{k_{RNA}}+\frac{1}{k_{cat}^{ini}}\right),$$

$$\phi_{IF2}^* \approx \sqrt{\frac{3}{4}}\sqrt{\frac{\ell_{ribo}\lambda^*}{\langle\ell\rangle k_{on}^{IF2}}} + \frac{\ell_{IF2}}{\langle\ell\rangle}\sqrt{\frac{\ell_{ribo}\lambda^*}{\langle\ell\rangle k_{on}^{IF1}}} + \frac{\ell_{IF2}}{\ell_{30S}}\phi_{123m}^\infty + \frac{\ell_{IF2}\lambda^*}{\langle\ell\rangle}\left(\frac{1}{k_{RNA}}+\frac{1}{k_{cat}^{ini}}\right),$$

$$\phi_{IF3}^* \approx \sqrt{\frac{3}{4}}\sqrt{\frac{\ell_{ribo}\lambda^*}{\langle\ell\rangle k_{on}^{IF3}}} + \frac{\ell_{IF3}}{\langle\ell\rangle}\sqrt{\frac{\ell_{ribo}\lambda^*}{\langle\ell\rangle k_{on}^{IF1}}} + \frac{\ell_{IF3}}{\ell_{30S}}\phi_{123m}^\infty + \frac{\ell_{IF3}\lambda^*}{\langle\ell\rangle}\left(\frac{1}{k_{RNA}}+\frac{1}{k_{cat}^{ini}}\right),$$

where for $k_{RNA}$ much faster than the association between the subunits, $\phi_{123m}^\infty \approx \sqrt{\frac{\ell_{30S}\lambda^*}{\langle\ell\rangle k_{on}^{50S}}}$.

## Appendix 5

### Estimation of optimal abundances

To compare prediction from our parsimonious framework (*Table 1*) requires specific values of kinetic parameters. We use empirical measurements together with scaling relations to estimate these kinetic parameters.

Catalytic rates for many enzymes have been measured in vitro, but the obtained values can be sharply incompatible with kinetic parameters that have been measured in the cell. An example is the class tRNA synthetases. Tallying the measured $k_{cat}$ for all wild-type *E. coli* aaRSs (*Jeske et al., 2019*), we find a median value of $k_{cat}^{aaRS} \approx 3$ s$^{-1}$, and 80% of reported value below 6 s$^{-1}$. The total molar concentration of aaRSs in the cell is comparable to the total number of ribosomes, and the per-step elongation speed of ribosome is above 15 s$^{-1}$ (*Dai et al., 2016*; *Johnson et al., 2020*). Hence, the absolute minimum catalytic rate to sustain the translation elongation flux needs to obey $k_{cat}^{aaRS} > 15$ s$^{-1}$, which is much higher than most in vitro measured values. To avoid the difficulties in estimating catalytic parameters, and to derive a lower bound on factor abundance from our model, we focus on the diffusive contributions (related to the associate rate) in our predictions, assuming large catalytic rates ($k_{cat} \rightarrow \infty$).

To estimate diffusion-limited association rate constants $\hat{k}_{on}$, we scaled the measured in vivo association rate constant for the ternary complex, $\hat{k}_{on}^{TC} = 6.4$ M$^{-1}$s$^{-1}$ (*Dai et al., 2016*) by diffusion of the respective components, that is, $\hat{k}_{on}^{AB}/\hat{k}_{on}^{TC} = (D_A + D_B)/(D_{TC} + D_{ribo})$, where $D_i$ is the diffusion coefficients for the molecular species $i$. While the in vivo diffusion coefficient for a number of component of the translation apparatus exist (*Bakshi et al., 2012*; *Sanamrad et al., 2014*; *Volkov et al., 2018*; *Plochowietz et al., 2017*), several factors do not have measured diffusion coefficients. For these, we used the cubic root scaling from the Stokes-Einstein relation (*Nenninger et al., 2010*), see *Appendix 5—table 1*.

We note that an alternative estimate for $\hat{k}_{on}$ using the Smoluchowski relation ($\hat{k}_{on}^{Smol} = 4\pi DR$, where $D$ is the relative diffusion coefficients of the two reactants and $R$ the capture radius) is overly simplistic as it assumes perfectly absorbing spheres. The actual diffusion-limited association rate constant could be much lower due to orientation constraints and other factors. It is also difficult to measure the capture radius in physiological conditions. Indeed, the Smoluchowski $\hat{k}_{on}^{Smol}$ calculated using the diffusion coefficients of EF-Tu in vivo ($\approx 3$ μm$^2$s$^{-1}$, [*Volkov et al., 2018*]) and a previous estimate for the capture radius ($R \approx 2$ nm, [*Klumpp et al., 2013*]) yields $\hat{k}_{on}^{TC,Smol} \approx 45$ μM$^{-1}$s$^{-1}$, which is several fold greater than the in vivo estimate of $k_{on}^{TC}$ based on kinetic measurements of elongation ($\hat{k}_{on}^{TC} = 6.4$ μM$^{-1}$s$^{-1}$, [*Dai et al., 2016*]). This comparison illustrates that the idealized Smoluchowski formula is not applicable. That said, our scaling approach does come at the price of assuming similar molecular properties leading to decrease of the association rate constants for the other tIFs. These could be further refined via for example, structural modeling (*Schlosshauer and Baker, 2004*), or upon new in vivo rate constant measurements.

Additional measured quantities required to compute our estimates are: the measured growth rate λ* = 5.5 × 10$^{-4}$ s$^{-1}$ (for *Figure 4* taken to be the average of the fast-growing species considered, corresponding to a doubling time of 21 ± 1 min. Individual species values: *E. coli*: 21.5 ± 1 min, *B. subtilis*: 21 ± 1 min, *V. natriegens*: 19 ± 1 min. See below for slower growth conditions), the tRNA concentration (estimated from the tRNA to ribosome ratio of 6.5 (*Dong et al., 1996*) using: $\text{tRNA}_{tot} = (\text{tRNA/ribo})\phi_{ribo}P/\ell_{ribo}$), the maximum per-codon elongation rate, excluding ternary complex diffusion, $k_{el}^{max} = 22$ s$^{-1}$ (*Dai et al., 2016*) (used to estimate the number of tRNAs sequestered on ribosomes and therefore the excess tRNA number in the optimum for aaRS, see *Equations 18 and 38*), the in-protein amino acid concentration $P = 2.6$ M (*Klumpp et al., 2013*; *Bremer and Dennis, 2008*).

For the fast growth average, results are displayed in *Figure 4* listed in *Supplementary file 2*. Additional predictions in individual conditions are shown in *Figure 4—figure supplement 1*, with numerical values for measured and predicted values listed in *Supplementary files 1–4*. For predictions in different growth conditions/species, we used used the measured growth rates in the corresponding conditions (values listed in *Supplementary files 1* and *3*), and association rate constants estimated based on *E. coli* data (*Appendix 5—tables 1–3*), and the tRNA abundance (only needed

for the prediction of aaRS) at the corresponding growth rate in *E. coli* from *Dong et al., 1996*. As a result of the lack of quantitation of tRNA abundance in other species, these values were used for *B. subtilis, V. natriegens* and *C. crescentus*, and should be interpreted with caution given possible difference in cellular physiology for these species.

**Appendix 5—table 1.** Protein sizes (number of codons) and diffusion coefficients.

Unless otherwise noted, number of codons per protein are taken for *E. coli* (*Keseler et al., 2017*) (ribosome size taken from *Wittmann, 1982*). #For the ternary complex, the total mass of tRNA+EF-Tu was converted to an equivalent amino acid length for the diffusion constant scaling estimate. †For aaRS, the size for the summed aaRSs is, from the coarse graining, $\ell_{aaRS} = \sum_i \phi_{aaRS,i} / \sum_i (\phi_{aaRS,i}/\ell_{aaRS,i})$, here with proteome fractions estimated from ribosome profiling (*Li et al., 2014*) in *E. coli* and sizes accounting for varying complex stoichiometries. Measured diffusion coefficients are taken from: *Bakshi et al., 2012*; *Sanamrad et al., 2014* for the ribosome, from *Plochowietz et al., 2017*; *Volkov et al., 2018* for tRNAs, and from *Volkov et al., 2018* for the TC.

| Factor | Number of codon per protein | Diffusion coefficient (μm² s⁻¹) |
|---|---|---|
| Ribosome | $\ell_{ribo} = 7336$ | $D_{ribo} = 0.05 \pm 0.01$ |
| 30S subunit | $\ell_{30S} = 3108$ | $D_{subunits} = 0.2 \pm 0.1$ |
| TC | $\ell_{TC} = 630^{\#}$ | $D_{TC} = 3 \pm 0.5$ |
| tRNA | N/A | $D_{tRNA} = 8 \pm 1$ |
| IF1 | $\ell_{IF1} = 72$ | $D_{IF1} = D_{TC} \sqrt[3]{\frac{\ell_{TC}}{\ell_{IF1}}}$ |
| IF2 | $\ell_{IF2} = 890$ | $D_{IF2} = D_{TC} \sqrt[3]{\frac{\ell_{TC}}{\ell_{IF2}}}$ |
| IF3 | $\ell_{IF3} = 180$ | $D_{IF3} = D_{TC} \sqrt[3]{\frac{\ell_{TC}}{\ell_{IF3}}}$ |
| EF-G | $\ell_G = 704$ | $D_G = D_{TC} \sqrt[3]{\frac{\ell_{TC}}{\ell_G}}$ |
| EF-Ts | $\ell_{Ts} = 283$ | $D_{Ts} = D_{TC} \sqrt[3]{\frac{\ell_{TC}}{\ell_{Ts}}}$ |
| EF-Tu | $\ell_{Tu} = 394$ | $D_{Tu} = D_{TC} \sqrt[3]{\frac{\ell_{TC}}{\ell_{Tu}}}$ |
| aaRS | $\ell_{aaRS} = 987^{\dagger}$ | $D_{aaRS} = D_{TC} \sqrt[3]{\frac{\ell_{TC}}{\ell_{aaRS}}}$ |
| RF1/RF2 | $\ell_{RFI} = 362$ | $D_{RFI} = D_{TC} \sqrt[3]{\frac{\ell_{TC}}{\ell_{RFI}}}$ |
| RF4 | $\ell_{RF4} = 185$ | $D_{RF4} = D_{TC} \sqrt[3]{\frac{\ell_{TC}}{\ell_{RF4}}}$ |

**Appendix 5—table 2.** Expression used to estimate the association rate constants for our predictions (*Table 1*).

Diffusion coefficients are listed in *Appendix 5—table 1*.

| Factors involved in reaction | Variable | Used expression for association rate constant |
|---|---|---|
| Ternary complex and ribosome | $\hat{k}_{on}^{TC}$ | $6.4 \pm 0.6$ μM⁻¹s⁻¹ (*Dai et al., 2016*) |
| EF-G and ribosome | $\hat{k}_{on}^{G}$ | $\hat{k}_{on}^{TC}(D_G + D_{ribo})/(D_{TC} + D_{ribo})$ |
| aaRS And tRNAs | $\hat{k}_{on}^{aaRS}$ | $\hat{k}_{on}^{TC}(D_{tRNA} + D_{aaRS})/(D_{TC} + D_{ribo})$ |
| EF-Ts and ribosome | $\hat{k}_{on}^{Ts}$ | $\hat{k}_{on}^{TC}(D_{Ts} + D_{ribo})/(D_{TC} + D_{ribo})$ |
| EF-Tu and tRNAs | $\hat{k}_{on}^{Tu}$ | $\hat{k}_{on}^{TC}(D_{tRNA} + D_{Tu})/(D_{TC} + D_{ribo})$ |
| IF1 and 30S subunit | $\hat{k}_{on}^{IF1}$ | $\hat{k}_{on}^{TC}(D_{IF1} + D_{subunit})/(D_{TC} + D_{ribo})$ |
| IF2 and 30S subunit | $\hat{k}_{on}^{IF2}$ | $\hat{k}_{on}^{TC}(D_{IF2} + D_{subunit})/(D_{TC} + D_{ribo})$ |
| IF3 and 30S subunit | $\hat{k}_{on}^{IF3}$ | $\hat{k}_{on}^{TC}(D_{IF3} + D_{subunit})/(D_{TC} + D_{ribo})$ |

*Continued on next page*

*Appendix 5—table 2 continued*

| Factors involved in reaction | Variable | Used expression for association rate constant |
|---|---|---|
| 50S and 30S subunits | $\hat{k}_{on}^{50S}$ | $\hat{k}_{on}^{TC}(D_{subunit} + D_{subunit})/(D_{TC} + D_{ribo})$ |
| RF1/RF2 and ribosome | $\hat{k}_{on}^{RFI}$ | $\hat{k}_{on}^{TC}(D_{RFI} + D_{ribo})/(D_{TC} + D_{ribo})$ |
| RF4 and ribosome | $\hat{k}_{on}^{RF4}$ | $\hat{k}_{on}^{TC}(D_{RF4} + D_{ribo})/(D_{TC} + D_{ribo})$ |

**Appendix 5—table 3.** Additional parameters used to obtain numerical values for predictions. For the doubling times (growth rates) and tRNA to ribosome ratios used for in individual growth conditions considered, see *Supplementary files 2* and *4*. $P$ is taken from *Klumpp et al., 2013*, $k_{el}^{max}$ from *Dai et al., 2016*, and the tRNA/ribosome ratios from *Dong et al., 1996*.

| Parameter | Value | Description |
|---|---|---|
| $P$ | 2.6 ± 0.5 M | In-protein amino acid concentration in the cell. |
| $\lambda$ | $(5.5 \pm 0.6) \times 10^{-4}$ s$^{-1}$ | Average fast growth, see *Supplementary file 1*. |
| $\langle \ell \rangle$ | 200 ± 10 | Average number of codons per protein (*Equation 16*). |
| $n_{aa}$ | 20 ± 2 | Rescaling factor in elongation model (see *Equation 26*). |
| $k_{el}^{max}$ | 22 ± 2 s$^{-1}$ | Maximal translation elongation rate. |
| $\sqrt{1 + \delta}$ | 1.05 ± 0.01 | Factor in three stop codon model (see *Equation 23*) |
| $t := $ tRNA/ribosome | 6.5 to 11 | Values taken listed in *Supplementary files 2* and *4*. |
| tRNA$_{tot}$ | $t\phi_{ribo}P/\ell_{ribo}$ | Total tRNA abundance, estimated from tRNA/ribosome. |

