## [Decision Letter]

**Acceptance summary:**

This paper presents a theoretical analysis of the abundance of components of the translation machinery (ribosomes, initiation, elongation and release factors, tRNA synthetases) in bacteria. These proteins make up a large fraction of the total proteome and their abundance is closely linked to cell growth. That the stoichiometry of the different components is adjusted such as to maximize the growth rate has been postulated a long time ago, but was so far only studied in detail for ribosomes and EF-Tu, the most abundant elongation factor. Here, the authors extend these earlier works to an unprecedented level of detail and provide a complete analysis based on this idea and derive the optimal stoichiometry for all these factor, which they find to be in good agreement with the observed abundance in different bacteria. This provides new evidences new evidence supporting the idea of proteome optimization for maximal growth.

**Decision letter after peer review:**

Thank you for submitting your article "First-principles model of optimal translation factors stoichiometry" for consideration by *eLife*. Your article has been reviewed by 3 peer reviewers, and the evaluation has been overseen by a Reviewing Editor and Aleksandra Walczak as the Senior Editor. The following individuals involved in review of your submission have agreed to reveal their identity: Martin J Lercher (Reviewer #1); Srividya Iyer-Biswas (Reviewer #3).

Essential revisions:

The essential revisions are listed in the list of comments from the different reviewers.

– Discuss the limitations of using the properties of the "metabolic protein sector" corresponding to the optimal stoichiometry condition to derive the optimal conditions for the translation factor sector and the ribosome sector.

– Expand the discussion of the case of Ef-Tu by moving parts of the Supplementary in the main text (at the end of P.12) (comment from Reviewer #2).

– Discuss in more depth the results of Figure 4, and in particular the need for "fitting parameters".

– Provide a point-by-point answer all the reviewers comments.

*Reviewer #1 (Recommendations for the authors):*

To make the parameter choices transparent, there should be a table that lists all model parameters, how they were derived, and an estimate of the uncertainty of the value.

*Reviewer #2 (Recommendations for the authors):*

This is an highly interesting study and there are only a few points, where it could be improved in my opinion:

– While the case of the release factors is very well described in the main text, the case of EF-Tu and aaRS is much harder to understand based on only the main text and the many references to the Supplement make the reading difficult here. I would suggest to maybe expand the description in the main text (starting with the last paragraph on p. 12).

– I think one could distinguish between a binding-limitation of the reactions and a diffusion-limitation of the binding step of the reaction. The former is needed for the analysis, while the latter is only required in the final step, where quantitative ratios are determined and binding rates have to be approximated by a diffusion-limited rate.

– Figure 4 could in principle also be done for individual organisms (using the data shown in Figure 1b).

*Reviewer #3 (Recommendations for the authors):*

Addressing the concern raised through clear prose in the manuscript will suffice.

[Editors' note: further revisions were suggested prior to acceptance, as described below.]

Thank you for submitting your article "First-principles model of optimal translation factors stoichiometry" for consideration by *eLife*. Your article has been reviewed by 3 peer reviewers, and the evaluation has been overseen by a Reviewing Editor and Aleksandra Walczak as the Senior Editor. The following individual involved in review of your submission has agreed to reveal their identity: Srividya Iyer-Biswas (Reviewer #3).

Essential revisions:

The referees are overall satisfied with the response you provided to their comments, and with the modification you made to the paper. There is one exception however, which regards your response to point (6) of reviewer #1. His comment is copied below.

Could you please provide an answer to his new comment, and modify the paper accordingly so that a final decision regarding your manuscript can be made.

*Reviewer #1 (Recommendations for the authors):*

The authors' responses adequately address my previous concerns, with one exception, the answer to my previous point (6) about the uniform re-scaling of the association rates. The authors now write in the manuscript:

"Importantly, the absolute values of the optimal concentrations can be anchored by the association rate constant between TC and the ribosome obtained from translation elongation kinetic measurements in vivo (Dai et al., 2016). The latter was found to be several-fold smaller than the simplest and absolute upper bound of a Smoluchowski estimate of perfectly absorbing spheres (section Estimation of optimal abundances), and we assume that the rescaling factor is the same for all reactions."

In my interpretation, what this really means is that the presented theory based on diffusion limitations provides (for whatever reasons) good predictions for the relative levels of tlFs, but not for the absolute levels. In order to get a quantitative prediction, the authors need to reduce ("rescale") all estimated on-rates by almost an order of magnitude, based on the known discrepancy between their estimate and an experimental value for a single tlF.

It is not clear (i) what causes that discrepancy (other than that the estimates are very approximate), and (ii) if and why the magnitude of the discrepancy can be transferred to other factors, and (iii) why its necessity does not invalidate the authors' approach.

This is an important point and should be made much more explicit than is currently done: The impressive quantitative agreement between theory and observation rests on the assumption that it is appropriate to make this uniform rescaling. In my opinion, the analysis is very interesting – but based on the necessary re-scaling, it is currently not clear if the authors' approach really provides the correct description of reality.

*Reviewer #2 (Recommendations for the authors):*

In my opinion, the authors have done a nice job responding to the comments and revising the manuscript.

*Reviewer #3 (Recommendations for the authors):*

The authors have adequately addressed my previous concerns. I recommend the manuscript for publication.

---

## [Author Response]

Essential revisions:The essential revisions are listed in the list of comments from the different reviewers.– Discuss the limitations of using the properties of the "metabolic protein sector" corresponding to the optimal stoichiometry condition to derive the optimal conditions for the translation factor sector and the ribosome sector.

Indeed, our original formalism implicitly assumed that the metabolic sector responds to the growth rate in a pre-determined way. The model was thus not about optimization of the full proteome from scratch. In the revised manuscript, we restrict the scope of the optimization problem to be within the translation sector and not the full proteome. This yields the same optimality condition. Therefore, the model predictions remain the same under this alternative formulation which relies on few assumptions. The revised model formalism is presented in section “Optimization under proteome allocation constraint” (starting p. 5, line 136, also copied on p. 8 of the current response).

– Expand the discussion of the case of Ef-Tu by moving parts of the Supplementary in the main text (at the end of P.12) (comment from Reviewer #2).

We have added additional details related to the EF-Tu case study to the main text (Box 1, p. 12 in the revised main text).

– Discuss in more depth the results of Figure 4, and in particular the need for "fitting parameters".

There are no fitting parameters. The empirical parameters used in the model are now listed in Appendix 5 Tables 1-3 (p.47-48). We have also added more discussion related to the results of Figure 4.

– Provide a point-by-point answer all the reviewers comments.

Our responses are detailed below.

Reviewer #1 (Recommendations for the authors):To make the parameter choices transparent, there should be a table that lists all model parameters, how they were derived, and an estimate of the uncertainty of the value.

We thank the reviewer for raising this all-important point on parameters. The formulae for our predictions are described in detail in section “Summary of optimal solutions” (Table 2). In order to clarify parameter selection, we have added Appendix 5 Tables 2 to list the formulae used to obtain the association rate constants, and Appendix 5 Table 3 to highlight additional parameters originally described in the supplementary text of the manuscript.

In particular, Appendix 5 Tables 2 now makes it explicit that the association rate constants are estimated by anchoring to that inferred from in vivo kinetic measurements for the ternary complex, scaled by the diffusion coefficients of participating factors. If measured values of diffusion constants for a factor have not been reported (the majority), we estimate the diffusion constant from that of the ternary complex rescaled by the ℓ^1/3^ for the two factors (Stokes-Einstein relation (6)), as shown in Appendix 5 Table 1.

Appendix 5 Table 3 includes a number of additional miscellaneous parameters, many of which are needed for the aaRS prediction: the maximum translation elongation rate (k_el_^max^=22 s^-1^, taken directly from (3)), the tRNA abundance (calculated from the measured tRNA/ribosome ratio and the total ribosome abundance converted to molar), tRNA_tot_ = (tRNA/ribosome ratio) 𝜙_ribo_ P/ℓ_ribo_. The tRNA/ribosome ratio is taken from (7) at the corresponding growth rate, and 𝜙_ribo_ is the estimated ribosomal protein abundance from ribosome profiling.

Reviewer #2 (Recommendations for the authors):This is an highly interesting study and there are only a few points, where it could be improved in my opinion:– While the case of the release factors is very well described in the main text, the case of EF-Tu and aaRS is much harder to understand based on only the main text and the many references to the Supplement make the reading difficult here. I would suggest to maybe expand the description in the main text (starting with the last paragraph on p. 12).

We thank the reviewer for raising an issue concerning the clarity of the section relating to EF-Tu and tRNA synthetases. In order to facilitate following the argument without resorting to the supplementary material, we have added a technical section (“Box. 1: The EF-Tu and aaRS transition line”, starting on p. 12 line 362) providing condensed details of the solution of that portion of the elongation cycle. Box. 1 contains intermediate equations and a derivation of the equation specifying the “transition line” (orange line in Figure 3B), critical for identifying the optimal solution for tRNA synthetases.

Furthermore, a new supplementary section “Interpretation of the sharp separation between aaRS and EF-Tu limited regimes” (p. 39, line 1381) was added to provide an intuitive understanding of how the separation of scales between codon specific/agnostic reactions in our coarse-grained model leads to a sharp transition between the aaRS limited and EF-Tu limited regime. The geometrical argument is summarized in a new supplementary figure (Figure 3—figure supplement 1).

– I think one could distinguish between a binding-limitation of the reactions and a diffusion-limitation of the binding step of the reaction. The former is needed for the analysis, while the latter is only required in the final step, where quantitative ratios are determined and binding rates have to be approximated by a diffusion-limited rate.

Following the reviewer’s suggestion, we have changed reference from diffusion limitation to binding limitation in all but the final section in which we estimate parameters (specifically: changing “diffusion-limited” to “binding-limited”). We now explicitly mention that we are estimating parameters under the assumption of diffusion limitation (anchoring values based on quantities measured in vivo), p.11 line 431:

“To obtain the numerical values of association rates needed to estimate the optimal tlF stoichiometry (Table 2), we further assume that binding reactions are diffusion limited.”

– Figure 4 could in principle also be done for individual organisms (using the data shown in Figure 1b).

We thank the reviewer for this suggestion. We now include in the revised manuscript predictions for individual species (and new conditions including slower growth) as Figure 4—figure supplement 1.

Reviewer #3 (Recommendations for the authors):Addressing the concern raised through clear prose in the manuscript will suffice.

We thank the reviewer for pointing out this subtle but deep point about our model formulation. We agree that in our original derivation, the derivative with respect to translation factor abundance leading to the optimality condition implicitly assumed that the system obeyed the growth law scaling under such perturbation. This is an overly strict requirement that is not necessary to arrive at the optimality condition.

Following your suggestion as well as Reviewer 1’s suggestion, in this revised submission we now instead consider the more circumscribed problem of constraining the total proteomic sector to the translation machinery, and use that constraint to derive the optimality condition. Under this new formulation, the derivative with regards to translation factor abundance is then not to be understood as what happens to actual cells under perturbed expression, but rather as what would happen in an (imagined) scenario where the total translation sector is fixed (and therefore the expression of other sectors as well). While this constraint restricts the scope of our model, it focuses attention on the core problem we are addressing, namely identifying the selective pressures operating on the sub-partitioning of resources allocated to a given pathway.

We also include a clarification about the meaning of the derivative in our optimization condition (p. 5, line 158):

“We emphasize that the derivative above corresponds to a perturbation scenario in which the tlF abundance is changed while maintaining fixed the total proteomic resources to the translation sector, as prescribed by our optimization procedure. As such, it does not correspond an actual perturbation easily realizable experimentally.”

References:

1. L. Volkov, et al., tRNA tracking for direct measurements of protein synthesis kinetics in live cells. Nat. Chem. Biol. 14, 618–626 (2018).

2. D. Rodriguez-Correa, A. E. Dahlberg, Kinetic and thermodynamic studies of peptidyltransferase in ribosomes from the extreme thermophile Thermus thermophilus. Rna 14, 2314–2318 (2008).

3. X. Dai, et al., Reduction of translating ribosomes enables *Escherichia coli* to maintain elongation rates during slow growth. Nat. Microbiol. 2, 1–9 (2016).

4. P. Nissen, et al., Crystal structure of the ternary complex of Phe-tRNAPhe, EF-Tu, and a GTP analog. Science (80-. ). 270, 1464–1472 (1995).

5. S. Klumpp, M. Scott, S. Pedersen, T. Hwa, Molecular crowding limits translation and cell growth. Proc. Natl. Acad. Sci. U. S. A. 110, 16754–9 (2013).

6. A. Nenninger, G. Mastroianni, C. W. Mullineaux, Size dependence of protein diffusion in the cytoplasm of *Escherichia coli*. J. Bacteriol. 192, 4535–4540 (2010).

7. H. Dong, L. Nilsson, C. G. Kurland, Co-variation of tRNA Abundance and Codon Usage in *Escherichia coli* at Different Growth Rates. J. Mol. Biol. 260, 649–663 (1996).

[Editors' note: further revisions were suggested prior to acceptance, as described below.]

Essential revisions:The referees are overall satisfied with the response you provided to their comments, and with the modification you made to the paper. There is one exception however, which regards your response to point (6) of reviewer #1. His comment is copied below.Could you please provide an answer to his new comment, and modify the paper accordingly so that a final decision regarding your manuscript can be made.Reviewer #1 (Recommendations for the authors):The authors' responses adequately address my previous concerns, with one exception, the answer to my previous point (6) about the uniform re-scaling of the association rates. The authors now write in the manuscript:"Importantly, the absolute values of the optimal concentrations can be anchored by the association rate constant between TC and the ribosome obtained from translation elongation kinetic measurements in vivo (Dai et al., 2016). The latter was found to be several-fold smaller than the simplest and absolute upper bound of a Smoluchowski estimate of perfectly absorbing spheres (section Estimation of optimal abundances), and we assume that the rescaling factor is the same for all reactions."In my interpretation, what this really means is that the presented theory based on diffusion limitations provides (for whatever reasons) good predictions for the relative levels of tlFs, but not for the absolute levels. In order to get a quantitative prediction, the authors need to reduce ("rescale") all estimated on-rates by almost an order of magnitude, based on the known discrepancy between their estimate and an experimental value for a single tlF.It is not clear (i) what causes that discrepancy (other than that the estimates are very approximate), and (ii) if and why the magnitude of the discrepancy can be transferred to other factors, and (iii) why its necessity does not invalidate the authors' approach.This is an important point and should be made much more explicit than is currently done: The impressive quantitative agreement between theory and observation rests on the assumption that it is appropriate to make this uniform rescaling. In my opinion, the analysis is very interesting – but based on the necessary re-scaling, it is currently not clear if the authors' approach really provides the correct description of reality.

Thank you for raising this important point. We have edited the manuscript by reorganizing and clarifying how we arrived at estimates of association rate constants.

Through these changes, we address: (i) why the physiological association rate constant might be different from the naïve estimate, (ii) the caveats associated with applying the same scaling factor to other factors, and (iii) how the scaling factor may affect our predictions. Modified paragraphs of the manuscript are reproduced below.

In the previous revision, our wording might have unintentionally led to two misconceptions. First, we did not intend to consider the Smoluchowski expression of association rate constants (k_on_=4𝜋DR) as the true diffusion-limited rate constant. The former assumes perfectly absorbing spherical molecules, which isn’t applicable to our case. Second, our wording of ‘rescaling’ was confusing. We did not rescale the Smoluchowski association rate constants. Instead, we used the measured k_on_ between TC-ribosome and estimated the k_on_ for other reactions using a scaling relationship between the association rate constant and the diffusion coefficients. We revised the manuscript accordingly to avoid these misconceptions.

Paragraph starting at line 431 (page 11):

“To obtain the numerical values of association rate constants needed for calculating the optimal tlF stoichiometry (Table 2), we used the measured k^onTC in vivo and estimated all other association rate constants using a biophysically motivated scaling (k^ denotes the raw association rate constant in units µM^−1^s^−1^, which is different from the rescaled k^, see section Conversion between 435 concentration and proteome fraction). […] Nonetheless, we note that the square-root dependence on these parameters (Table 2) for our predictions makes the numerical values less sensitive tlF-specific effects.”

Paragraph starting at line 537 (page 14):

Our coarse-graining approach has several limitations in its connection to detailed biochemical parameters. […] We further note that a number of conclusions from our model, such as the factor of ⟨ℓ⟩ separating the optimal abundances of elongation from initiation/termination tlFs, are generic and do not depend on the specific association rates.”

Paragraphs starting at line 1604 (page 47, Appendix 5):

“To estimate diffusion-limited association rate constants k^on, we scaled the measured in vivo association rate constant for the ternary complex, k^onTC=6.4μM−1s−1 (Dai et al., 2016) by diffusion of the respective components, i.e., k^onAB/k^onTC(DA+DB)/(DTC+Dribo), where *D*, is the diffusion coefficients for the molecular species *i*. […] These could be further refined via e.g., structural modeling (Schlosshauer and Baker, 2004) or upon new in vivo rate constant measurements.”